# From Zero to Hero: Advancing Zero-Shot Foundation Models for Tabular Outlier Detection

Xueying Ding [1]    Haomin Wen [1]    Simon Klütterman [2]    Leman Akoglu [1]

## Abstract

Outlier detection (OD) is widely used in practice; but its effective deployment on new tasks is hindered by lack of labeled outliers, which makes algorithm and hyperparameter selection notoriously hard. Foundation models (FMs) have transformed ML, and OD is no exception: Shen et al. (2025) introduced FoMo-0D, the first FM for OD, achieving remarkable performance against numerous baselines. This work introduces OUTFORMER, which advances FoMo-0D with (1) a mixture of synthetic priors and (2) self-evolving curriculum training. OUTFORMER is pretrained solely on synthetic labeled datasets and infers test labels of a new task by using its training data as in-context input. Inference is fast and zero-shot, requiring merely forward pass and no labeled outliers. Thanks to in-context learning, it requires zero additional work—no OD model training or bespoke model selection—enabling truly plug-and-play deployment. OUTFORMER achieves state-of-the-art performance on the prominent AD-Bench, as well as two new large-scale OD benchmarks that we introduce, comprising over 1,500 datasets, while maintaining speedy inference.

## 1. Introduction

Outlier detection plays a crucial role in many real-world applications in finance, medicine, security, etc. The widespread relevance of OD has driven the development of numerous detection methods (Aggarwal, 2013; Pang et al., 2021). Despite these advances, a fundamental challenge hinders OD's effective practical deployment (Campos et al., 2016; Ding et al., 2022; Ma et al., 2023): model selection, which entails *both algorithm choice and hyperparameter*
tuning. Specifically, the scarcity or complete absence of labeled outliers makes model selection notoriously hard.

Modern era machine learning has been transformed with the surge of large language models (LLMs) (Brown et al., 2020; Minaee et al., 2024) and foundation models (FMs) at large (Bommasani et al., 2021; Liu et al., 2025c; Liang et al., 2024; Van Breugel & Van Der Schaar, 2024). Specifically, the seminal TabPFN (Hollmann et al., 2023) and its successors (Hollmann et al., 2025; Grinsztajn et al., 2025), have shown that pretrained tabular FMs can act as "algorithms-in-a-box" for *supervised* tasks, predicting test samples of a new task through a mere forward pass while ingesting labeled training samples as input context. Through in-context learning (ICL) (Xie et al., 2021; Falck et al., 2024), pretrained FMs achieve remarkable success *without any additional training or hyperparameter (HP) tuning*. Furthermore, their success stands on *synthetic* pretraining datasets rather than massive amounts of labeled real-world datasets.

The implications of FMs and their ICL capability are game-changing for OD. First, ICL requires no additional training/tuning, therefore FMs **obviate the need for model (both algorithm and HP) selection—rendering the key obstacle in OD's effective use obsolete**. Recent unsupervised OD model selection approaches rely on meta-learning on historical OD tasks with labeled outliers (Zhao et al., 2021; 2022; Ding et al., 2024), which transfer surrogate predictors of performance to new tasks without labels. In principle, FMs perform a form of scalable meta-learning on numerous pretraining datasets with labels that effectively generalize to new tasks without labels. Consequently, FMs enable **zero-shot** OD without requiring any labeled outliers at inference. Second, **tabular FMs can thrive on synthetic pretraining data—bypassing the need for numerous real-world tabular OD datasets**, which are minuscule compared to the massive text data that LLMs are pretrained on.

Capitalizing on the premise, Shen et al. (2025) introduced the first foundation model for zero/0-shot OD called FoMo-0D. Despite being trained on synthetic datasets based on Gaussian Mixture Models (GMMs), FoMo-0D has achieved remarkable performance, sharing the **second top** position among 30 prominent OD methods (Shen et al., 2025). In this work, we introduce OUTFORMER; advanc-

[1]Carnegie Mellon University, Pittsburgh, PA, USA [2]Technical University Dortmund, Germany. Correspondence to: Leman Akoglu <lakoglu@andrew.cmu.edu>.

*Proceedings of the 43rd International Conference on Machine Learning*, Seoul, South Korea. PMLR 306, 2026. Copyright 2026 by the author(s).

*Table 1.* OUTFORMER **advances FMs for tabular OD and outperforms** FOMO-0D **(Shen et al., 2025) and other baselines on ADBench.** Best and second-best are highlighted. Reported are five relative performance metrics across baselines, and Win/Lose/Tie is the fraction of datasets where OUTFORMER wins/loses/ties against baseline, while $p$-values $\leq 0.05$ of the paired permutation test between baseline and OUTFORMER indicate OUTFORMER's performance is significantly better. See Appx. Table 22 for similar results w.r.t. AUPRC, and Appx. Tables 38 (AUROC) and 37 (AUPRC) for performances on individual datasets for all methods.

| Model | Avg. Rank (↓) | ELO (↑) | Winrate (↑) | rAUC (↑) | $C_\Delta$ (↓) | Win/Lose/Tie | p-val. |
|---|---|---|---|---|---|---|---|
| DTE-NP | $5.12_{\pm 2.3}$ | 1043 | 0.61 | $0.939_{\pm 0.08}$ | 0.39 | **0.46**/0.39/0.16 | 0.06 |
| kNN | $5.05_{\pm 2.6}$ | 1001 | 0.61 | $0.938_{\pm 0.09}$ | 0.36 | **0.49**/0.32/0.19 | 0.06 |
| LOF | $6.04_{\pm 3.6}$ | 961 | 0.53 | $0.913_{\pm 0.11}$ | 0.43 | **0.56**/0.26/0.18 | **0.00** |
| IForest | $8.46_{\pm 3.1}$ | 794 | 0.32 | $0.879_{\pm 0.12}$ | 0.52 | **0.75**/0.18/0.07 | **0.00** |
| DTE-C | $6.00_{\pm 3.1}$ | 937 | 0.53 | $0.932_{\pm 0.08}$ | 0.42 | **0.63**/0.23/0.14 | **0.00** |
| ICL | $5.63_{\pm 3.7}$ | 1005 | 0.57 | $0.925_{\pm 0.14}$ | 0.35 | **0.54**/0.32/0.14 | **0.02** |
| DDPM | $7.12_{\pm 2.9}$ | 943 | 0.43 | $0.904_{\pm 0.09}$ | 0.48 | **0.72**/0.12/0.16 | **0.00** |
| GOAD | $9.32_{\pm 2.9}$ | 981 | 0.23 | $0.805_{\pm 0.19}$ | 0.58 | **0.81**/0.11/0.09 | **0.00** |
| DeepSVDD | $9.81_{\pm 2.9}$ | 788 | 0.20 | $0.796_{\pm 0.16}$ | 0.63 | **0.84**/0.07/0.09 | **0.00** |
| TabPFN-OD | $4.74_{\pm 3.1}$ | 1227 | 0.65 | $0.945_{\pm 0.07}$ | 0.34 | **0.44**/0.35/0.21 | 0.12 |
| FOMO-0D | $6.00_{\pm 3.4}$ | 1084 | 0.54 | $0.928_{\pm 0.08}$ | 0.41 | **0.54**/0.26/0.19 | **0.01** |
| OUTFORMER | $\mathbf{4.02_{\pm 2.6}}$ | **1235** | **0.71** | $\mathbf{0.956_{\pm 0.06}}$ | **0.32** | – | – |

*Figure 1.* Total time (train&inference) quantiles (q10, q90) over all 1500+ datasets across benchmarks. OUT-FORMER **maintains speedy inference** while achieving competitive performance. green: shallow, blue: deep, red: foundation models. Deeper color: best model in each category.

ing zero-shot tabular OD FMs to SOTA performance with desirable inference latency, as shown in Table 1 and Fig. 1.

At the heart of our advance are two key innovations as highlighted in Fig. 2: (1) mixed data priors; a novel and diverse mixture of synthetic data distributions for both inliers and outliers, and (2) self-evolving curriculum training; a novel multi-armed bandit based curriculum to train over heterogeneous datasets of diverse distribution and dimensionality, without any manual specification of dataset order. Importantly, (1) and (2) are synergistic: Vanilla pretraining on our data mixture that subsumes GMMs can hurt (!) as compared to training only on GMMs (see Table 3). Additional levers that further improve performance are a larger model size to accommodate a wider pretraining data distribution, as well as ensembling with varying context samples and dimensions to alleviate the limited context size. Together, our proposed OUTFORMER **significantly outperforms** FOMO-0D **and reaches state-of-the art performance** not only on the *de facto* OD benchmark ADBench (Han et al., 2022) but also on two new large-scale OD benchmarks that we introduce with over 1400 datasets in total. The following summarizes our main contributions.

- **New FM for Tabular OD:** We introduce OUT-FORMER, a foundation model (FM) enabling zero-shot in-context outlier detection (OD). Provided with unlabeled in-context samples, OUTFORMER labels test instances through mere forward pass, removing the need for not only model training but also the bespoke unsupervised model selection (both method and hyperparameters) without any labeled outliers.
- **Mixed Synthetic Priors for Tabular OD:** We construct novel families of prior distributions for tabular data to synthesize diverse inliers and outliers. Our mix-

ture consists of Gaussian Mixture Models (GMMs), Structural Causal Models (SCMs) and Copulas, yielding diverse outlier archetypes.

- **Self-evolving Curriculum for Mixed Prior Training:** Based on the intriguing finding that vanilla-trained models on our mixed prior that subsumes GMMs underperform GMM-only models, we employ a new curriculum training strategy based on a multi-armed bandit, where heterogeneous data categories w.r.t. the generating prior and dimensionality depict separate arms. Notably, our curriculum is self-evolving, requiring no manual intervention or any apriori specification of dataset order by difficulty.
- **SOTA Performance w/ Low Latency:** We evaluate OUTFORMER extensively against established baselines; shallow, deep and foundation models including the prior FOMO-0D (Shen et al., 2025) and their reported SOTA DTE-NP (Livernoche et al., 2024), on three real-world OD benchmarks. OUTFORMER significantly outperforms all baselines on ADBench (Table 1) and reaches SOTA across 1500+ datasets from all benchmarks (Table 4), establishing the first OD foundation model with SOTA performance. Further, OUTFORMER offers speedy inference by completely bypassing any model training or selection (Fig. 1).

**Reproducibility.** We open-source (1) OUTFORMER check-points, (2) code for our synthetic prior generators, (3) our two new large OD benchmarks (1446 datasets), and (4) individual dataset performances at `https://github.com/psorus/Outformer.git` and `https://huggingface.co/MacrOData-CMU`.

**Conflict of Interest.** We declare that we have no competing financial interests related to the research in this paper.

# 2. Preliminaries

## 2.1. Problem

We consider the outlier detection problem in the inductive setting with a train/test data split, where training set comprises only inliers. Let $\mathcal{D}_{\text{train}} = \{(\mathbf{x}_1, y_1), \ldots, (\mathbf{x}_n, y_n)\}$ denote the "clean" training set with input points $\mathbf{x}_i \in \mathbb{R}^d$ and $y_i = -1$, $\forall i \in [n]$. Given test set $\mathcal{D}_{\text{test}} = \{\mathbf{x}_i\}_{i=1}^{n'}$, containing both inliers and outliers, the task is to assign labels $y_i \in \{+1, -1\}$, $\forall i \in [n']$. Note that $\mathcal{D}_{\text{test}} \cap \mathcal{D}_{\text{train}} = \emptyset$.

## 2.2. Prior-Fitted Networks (PFNs)

**Posterior Predictive Distribution (PPD):** In Bayesian supervised learning, a prior defines a hypothesis space $\Phi$, where each $\phi \in \Phi$ specifies a data-generating mechanism. The posterior predictive distribution, $p(\cdot \mid \mathbf{x}_{\text{test}}, \mathcal{D}_{\text{train}})$, predicts new data $\mathbf{x}_{\text{test}}$ conditioned on the training data $\mathcal{D}_{\text{train}} = \{(\mathbf{x}_1, y_1), \ldots, (\mathbf{x}_n, y_n)\}$. By Bayes' theorem, PPD is obtained by integration over all hypotheses in $\Phi$:

$$p(y_{\text{test}}|\mathbf{x}_{\text{test}}, \mathcal{D}_{\text{train}}) = \int_{\Phi} p(y_{\text{test}}|\mathbf{x}_{\text{test}}, \phi)p(\mathcal{D}_{\text{train}}|\phi)p(\phi)d\phi,$$

where $p(\phi)$ denotes the prior probability and $p(\mathcal{D}|\phi)$ is the likelihood of data $\mathcal{D}$ under generating mechanism $\phi$.

**PFNs for PPD approximation:** Computing the exact PPD is generally intractable, therefore, Prior-data Fitted Networks (PFNs) have been proposed to approximate it (Müller et al., 2022). Unlike conventional models trained directly on a given dataset, PFNs are pre-trained on simulated datasets sampled from a specified prior. This framework consists of two stages: pre-training and inference.

During **pre-training**, datasets are sampled $\mathcal{D} \sim p(\mathcal{D}|\phi)$ from a sampled hypothesis $\phi \sim p(\phi)$, and split into $\mathcal{D}_{\text{test}} \subset \mathcal{D}$ and $\mathcal{D}_{\text{train}} = \mathcal{D} \setminus \mathcal{D}_{\text{test}}$. The PFN $q_{\boldsymbol{\theta}}$, parameterized by $\boldsymbol{\theta}$, is trained to minimize the expected loss for a test point $(\mathbf{x}_{\text{test}}, y_{\text{test}}) \in \mathcal{D}_{\text{test}}$ given $\mathcal{D}_{\text{train}}$, i.e.

$$\mathcal{L} = \mathbb{E}_{(\{(\mathbf{x}_{\text{test}}, y_{\text{test}})\} \cup \mathcal{D}_{\text{train}}) \sim p(\mathcal{D})}[-\log q_{\boldsymbol{\theta}}(y_{\text{test}}|\mathbf{x}_{\text{test}}, \mathcal{D}_{\text{train}})],$$

which can be interpreted as minimizing the expected KL divergence between $p(\cdot|\mathbf{x}, \mathcal{D})$ and $q_{\boldsymbol{\theta}}(\cdot|\mathbf{x}, \mathcal{D})$ (Müller et al., 2022). Typically, a PFN model $q_{\boldsymbol{\theta}}$ is built on a Transformer backbone (Vaswani et al., 2017). It takes as input the labeled $\mathcal{D}_{\text{train}}$ as *context* points and the unlabeled test set $\mathcal{D}_{\text{test}}$ as *query* points, and outputs the conditional class probabilities for the test points. Importantly, test points attend only to the labeled training points via cross-attention.

During **inference**, a pre-trained (frozen) PFN acts like an "algorithm in a box", ingesting incoming training data as context and inferring the PPD $q_{\boldsymbol{\theta}}(\cdot|\mathbf{x}_{\text{test}}, \mathcal{D}_{\text{train}})$ of test data in a single forward pass, without any gradient-based parameter updates on new datasets (Hollmann et al., 2023) through in-context learning (Xie et al., 2021; Falck et al., 2024).

## 2.3. Proposed OUTFORMER: Overview

The core idea behind our foundation model for outlier detection (OD) is to pre-train a PFN built on a Transformer backbone on numerous OD tasks in a supervised manner. OUTFORMER's supervised pre-training consumes synthetically generated tabular datasets, each containing labeled inliers and outliers. To reflect the wide range of generative mechanisms observed in real world data, we construct a carefully curated library of **mixed tabular data priors** that span diverse multivariate distributions and outlier archetypes.

Importantly, naïvely training OUTFORMER on datasets with widely varying distributions, dimensionality, and outlier severity leads to subpar performance and demands careful handling of the training dynamics. The difficulty mirrors that of training reasoning-focused language models (LMs) on coding or mathematics tasks, where effective learning requires building foundational skills before advancing to more complex tasks; much like one cannot solve olympiad-level math without first learning calculus. Unlike math curricula for LMs where task difficulty naturally follows grade levels, our OD tasks lack a predetermined ordering of difficulty. To overcome this, we introduce a **self-evolving curriculum** based on a multi-armed bandit setup, where each arm corresponds to a task category. A carefully designed reward function guides the bandit toward selecting tasks that are neither too easy nor too difficult at any point in training.

Finally, OUTFORMER employs two ensembling strategies at inference, subsampling and feature bagging (Aggarwal & Sathe, 2017), not only to accommodate large-$n$/large-$d$ datasets that exceed its context size, but also on smaller datasets for the prowess of ensembling. Ensembling FMs is quite lightweight at inference; relying on mere forward pass, and not training multiple, potentially large OD models.

In short, OUTFORMER is built on two core innovations: an effective training dataset design and an effective training algorithm. The following introduces our (1) mixed synthetic tabular data priors and (2) self-evolving curriculum training strategy, in §3 and §4, respectively. Fig. 2 shows the conceptual design of our proposed framework.

# 3. OUTFORMER: Mixed Synthetic Priors

The data priors used to synthesize training datasets for PFNs should reflect the broad range of distributions that appear in the real world to support downstream generalization. As the *first key building block* of our OUTFORMER, we propose a mixture of tabular data priors to represent a diverse spectrum of potential data-generating functions.

Our mixture contains three distinct priors for inliers—Gaussian Mixture Models (GMMs), Structural Causal Models (SCMs), and Copulas—which lend themselves to five dif-

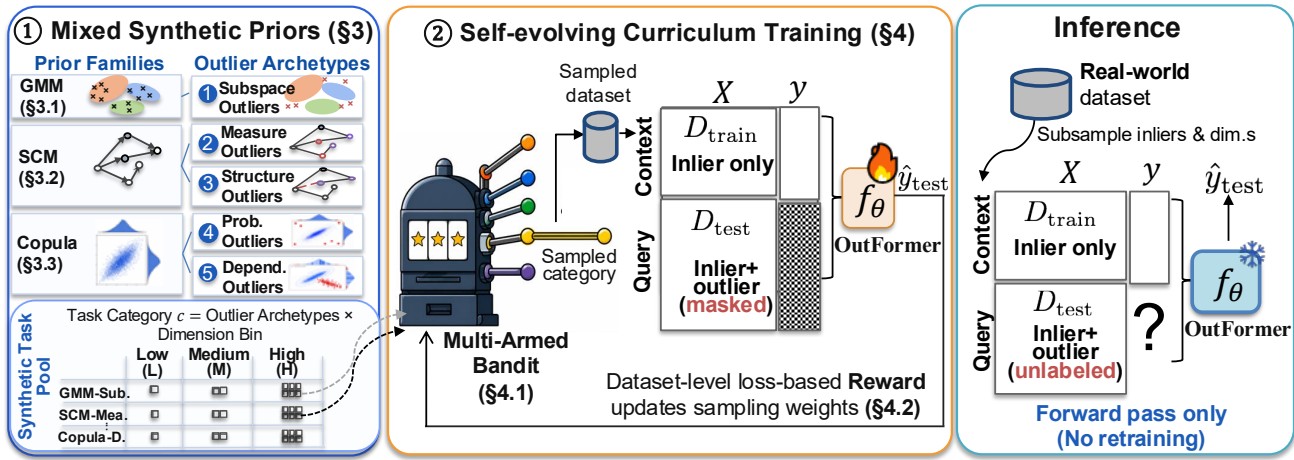

*Figure 2.* Proposed OUTFORMER framework for tabular outlier detection. Pretraining capitalizes on diverse labeled synthetic datasets from a **mixture of data priors**, and employs **self-evolving curriculum** training based on multi-armed bandits to estimate the masked test labels. At inference, (frozen) OUTFORMER estimates the test labels of a real-world task via forward pass only while ingesting training data as context. Speedy inference enables ensembling OUTFORMER over different subsampled examples and dimensions in the context.

ferent outlier archetypes. We describe these individual data priors and their unique advantages as follows. Table 2 shows that our priors are **complementary** (training on a single prior yields strong within-prior generalization but weaker cross-prior transfer), as well as **nontrivial** (well-established baselines do not readily achieve top performance). Appx. C provides all the implementation details.

### 3.1. Gaussian Mixtures

**Inliers:** Following Shen et al. (2025) we simulate inliers from multivariate GMMs, with varying number of components and dimensions, and designate the points within $90th$ percentile of GMM likelihood as inliers.

**Outliers:** We generate *contextual subspace outliers* by randomly selecting a component and "inflating" variances along a subset of dimensions, designating points outside the $90th$ percentile as outliers. We vary the severity of outliers based on the subspace size and inflation scale.

**Advantages:** GMMs reflect multi-modality that naturally arises from multiple underlying subpopulations in real world datasets. Varying this multi-modality, they enable capturing simple to highly expressive, complex distributions. In fact, the GMM family is a universal approximator for continuous densities (Carreira-Perpinan, 2002). GMMs also offer efficient sampling of large synthetic datasets via simple parameterization and analytical expressions.

### 3.2. Structural Causal Models

**Inliers:** SCMs model the generative relationships among variables via causal graphs and structural equations. An SCM is specified by a directed acyclic graph $G=(V, E)$

where each node $j \in V$ corresponds to a variable. Structural equations (or mechanisms) $X_j = f_j(X_{Pa(X_j;G)}, \epsilon_j)$ specify causal relationships, where $Pa(X_j; G)$ is the set of parent variables (i.e., direct causes) of $j$ in the causal graph $G$, and $\epsilon_j$ represents exogenous noise.

Building on Hollmann et al. (2023) we materialize $G$ by an MLP from which we randomly drop a fraction of the edges. Edges are directed from the input layer toward the output. From each sampled SCM, we pick a subset of nodes $\mathcal{X} = \{X_1, \ldots, X_d\} \subset V$ and $Y \in V$ to serve as the features and the target, respectively. We sample the inputs, edge weights, and noise variables from standard Normal distributions. We set the structural equations $f_j$'s as the weighted sum followed by a random activation function.

Given $G$, sampled weights and activations, we sample an input and the noise variables i.i.d. and propagate these through $f_j$'s according to $G$ (i.e. through the MLP). Reading out the values at the selected variables $(\mathcal{X}, Y)$ yields a synthetic point $(\mathbf{x}, y)$. We repeat to obtain a set of inliers.

**Outliers:** Using SCMs, we simulate two outlier archetypes: *measurement outliers* and *structural outliers*. The former capture exogenous measurement errors, while the latter reflect endogenous structural changes.

To create each measurement outlier, we pick a variable $X_j \in \mathcal{X}$ at random, sample its exogenous noise $\epsilon_j \sim \mathcal{N}(0, s)$ with inflated variance, and let the resulting perturbation propagate to descendant variables in a manner consistent with the underlying structural dependencies—producing a multivariate outlier. This process models exogenous shocks while preserving the causal structure and mechanisms. In contrast, structural outliers represent changes in the underlying causal relationships. We either "break" a causal edge (by setting its

*Table 2.* **Our proposed priors are complementary and non-trivial.** We train the original FOMO-0D model from Shen et al. (2025), while replacing its GMM prior with alternative priors and report avg. AUROC performance across synthetic datasets from individual priors. Best and Second are highlighted. The prominent diagonal shows that models trained on individual priors generalize better within than across prior distributions. Even well-established baselines –kNN (averaged over $k \in \{5, 10, 20, 50\}$) and Isolation Forest (IForest) with default hyperparameters – do not readily achieve top performance.

| Train on:/ Test on: | Copula-Depend. | Copula-Prob. | SCM-Struct. | SCM-Measure. | GMM |
|---|---|---|---|---|---|
| Copula-Depend. | $0.994_{\pm 0.03}$ | $0.813_{\pm 0.05}$ | $0.759_{\pm 0.15}$ | $0.765_{\pm 0.12}$ | $0.748_{\pm 0.10}$ |
| Copula-Prob. | $0.861_{\pm 0.08}$ | $0.923_{\pm 0.04}$ | $0.646_{\pm 0.17}$ | $0.701_{\pm 0.15}$ | $0.831_{\pm 0.07}$ |
| SCM-Struct. | $0.785_{\pm 0.10}$ | $0.803_{\pm 0.07}$ | $0.981_{\pm 0.06}$ | $0.962_{\pm 0.05}$ | $0.760_{\pm 0.12}$ |
| SCM-Measure. | $0.767_{\pm 0.08}$ | $0.835_{\pm 0.04}$ | $0.982_{\pm 0.06}$ | $0.976_{\pm 0.06}$ | $0.753_{\pm 0.12}$ |
| GMM | $0.862_{\pm 0.04}$ | $0.902_{\pm 0.04}$ | $0.979_{\pm 0.06}$ | $0.965_{\pm 0.05}$ | $0.941_{\pm 0.06}$ |
| kNN | $0.849_{\pm 0.04}$ | $0.890_{\pm 0.04}$ | $0.980_{\pm 0.06}$ | $0.965_{\pm 0.06}$ | $0.864_{\pm 0.08}$ |
| iForest | $0.677_{\pm 0.04}$ | $0.776_{\pm 0.04}$ | $0.964_{\pm 0.06}$ | $0.947_{\pm 0.05}$ | $0.742_{\pm 0.10}$ |

*Table 3.* **Our proposed self-evolving curriculum (SEC) empowers mixed-prior training.** Naïve mixed-prior training underperforms GMM-only training on GMM datasets and ADBench. Training on mixed-priors using SEC boosts performance on all priors as well as AD-Bench as compared to no curriculum.

| Test on: Train on: | GMM (Prior) | Mixed (Priors) | ADBench (Real) |
|---|---|---|---|
| GMM (w/o SEC) | **0.941** | 0.935 | 0.920 |
| Mixed (w/o SEC) | 0.873 | 0.937 | 0.898 |
| Mixed (w/ SEC) | 0.930 | **0.968** | **0.926** |

weight to zero) or reverse its direction.[1] These interventions simulate inherent changes in parent–child relationships, altering the data-generating mechanisms.

While measurement outliers tend to exhibit extreme values, structural outliers may or may not be extreme-value observations. Instead, their defining feature is a deviation from the expected causal dependencies, which can change the joint distribution. The magnitude of both outliers depends on the depth at which they occur, as the effect propagates downstream. We also vary outlier severity via the noise inflation factor for measurement outliers and the fraction of edges to break or reverse for structural outliers.

**Advantages:** Nonparametric SCMs, under mild assumptions, can universally represent a wide class of data-generating processes (Bongers et al., 2021), making them a flexible framework for tabular data from the real world, such as biology, economics and social sciences (Peters et al., 2017; Schölkopf, 2022). Different structural functions, graph topologies, and noise distributions produce highly diverse distributions, with multi-modality, nonlinearity, non-Gaussian marginals, and heteroscedastic noise—adding to the diversity of our prior mixture. Finally, they offer efficient sampling with a single topological pass through an MLP, which is linear time in the number of edges.

### 3.3. Copulas

**Inliers:** Real world univariate distributions have been shown to follow skewed, power-law-like distributions across numerous domains (Clauset et al., 2009). While SCMs produce non-Gaussian marginals, they do not explicitly allow specifying the feature marginals, for which we use copulas (Nelsen, 2006; Houssou et al., 2022).

What makes copula models extremely useful is Sklar (1959)'s theorem, which states that for any joint cumulative distribution function (CDF) of continuous random variables

$\{X_j\}_{j=1}^{d}$ with marginal CDFs $F_j(x_j) = P(X_j \leq x_j)$, there exists a unique copula function $C$ such that

$$F(x_1, \ldots, x_d) = C(F_1(x_1), \ldots, F_d(x_d)) .$$

This decomposition allows us to model the marginals and dependence structure separately, with each feature's marginal specified independently, offering full control over their type and diversity. With this flexibility at hand, we draw each feature from a pool of parametric probability functions including Gaussian, Beta, Exponential, Student's t, Power law, and Log-logistic distributions. For the copula, we choose either a Gaussian or (bivariate) vine copula. Given copula $C$ and arbitrary marginal CDFs $F_1, \ldots, F_d$ specified, we generate each point by drawing $\mathbf{u} \sim C$ on the unit cube $[0, 1]^d$, and transforming $x_j = F_j^{-1}(u_j); \ \forall j \in [d]$ (in parallel). The resulting $\mathbf{x} = (x_1, \ldots, x_d)$ is a synthetic inlier. We widelu vary the marginal distributions, their parameters, and the copula to obtain a variety of tabular datasets.

**Outliers:** Leveraging the decomposition of a copula into marginal distributions and dependence structure, we construct both *probabilistic outliers* and *dependence outliers*.

For each probabilistic outlier, we randomly select a subset of features and perturb their copula coordinates by pushing them toward the boundaries, replacing $u_j$ with very small or large values. Applied in copula space and subsequently mapped back through the inverse marginal transformations, these perturbations generate multivariate extremes that mimic exogenous measurement errors while approximately preserving the underlying dependence structure. For each dependence outlier, we overwrite its copula vector with one that violates the original dependence; either by inverting a subset of dimensions by overwriting $u_j := 1 - u_j$, or by randomly permuting coordinates. These manipulations disrupt the joint dependencies (thereby the data-generating mechanism) without necessarily inducing extreme values. We vary the outlier severity through the magnitude of $u_j$'s perturbed as well as the fraction of dimensions perturbed, inverted, or permuted.

---

[1]MLP edges connect only consecutive layers; without any skip or within-layer edges, thus, edge reversing does not create cycles.

**Advantages:** Similar to GMMs and SCMs, copulas offer universality thanks to Sklar's theorem that any multivariate joint distribution can be represented as a copula plus marginal distributions. This decomposition further allows (*i*) specifying arbitrary marginals for each individual feature—the primary reason we employed Copulas, but also (*ii*) easy, parallelizable sampling from complex high-dimensional multivariate distributions. Further, copulas model highly flexible dependencies, accommodating non-linear relationships, asymmetries, and tail dependence; where basic correlation/covariance-based models fall short.

# 4. OUTFORMER: **Self-Evolving Curriculum**

We aim to pre-train OUTFORMER on a large collection of datasets from multiple different priors to capture the diversity of real world OD tasks and thereby improve downstream generalization. However, this creates a challenging multi-task training scenario in which the datasets vary widely. Furthermore, this diversity is layered; due to heterogeneity in (*i*) priors, (*ii*) datasets within each prior (e.g., in dimensionality and marginal distributions), and (*iii*) individual points within a dataset (e.g., in outlier severity). Pretraining dataset diversity suggests that ordering the tasks carefully may be more effective than learning them jointly, therefore the need for a curriculum (Pentina et al., 2015), such that harder tasks with large gradients do not dominate, and the model retains the opportunity to learn from easier tasks first.

However, the difficulty order of tasks in our setting is unknown apriori. As the *second key building block* of OUTFORMER, we introduce a self-evolving curriculum (SEC) that treats distinct task categories as arms in a multi-armed bandit (MAB), enabling the model to adaptively select tasks of suitable difficulty during training. As the MAB setup promotes sample-efficient[2] learning through a carefully crafted implicit reward, it simultaneously prevents wasted sampling by shifting focus away from overly hard tasks toward learnable categories, conserving on-the-fly sampling effort.

Table 3 shows that SEC plays a key role in mixed prior training. Counterintuitively, naïve training on mixed priors that *subsume* the GMM prior underperforms GMM-only training on ADBench. A deep-dive shows that larger gradients from other priors dominate GMM, leading to a "missed opportunity" to learn the GMM distributions, as shown with inferior performance on GMM datasets.[3] This negatively transfers to ADBench that exhibits Gaussian-noise-like outliers (Livernoche et al., 2024). SEC boosts learning not only GMM but all priors, improving generalization to ADBench.

## 4.1. Designing the Curriculum

A simple curriculum on a random batch of datasets could sort all points by their individual loss and discard those with high loss (treating them as hard or noisy) and backpropagate gradients only through the remaining points (Jiang et al., 2015). This approach requires no explicit notion of task difficulty, however, it treats all points uniformly, regardless of the datasets or priors from which they originate. As a result, this simple design is costly: As datasets are sampled *on-the-fly* during training, discarding numerous points from certain (hard) datasets for not meeting the loss threshold results in wasted data. Conversely, (easy) datasets with many low-loss points contribute little to learning. To adaptively shift focus away from such low utility tasks, a bookkeeping mechanism is required to track which priors and type of datasets are most beneficial during the course of training.

To that end, we frame our curriculum learning objective as a non-stationary[4] multi-armed bandit (MAB) problem (Besbes et al., 2014), where the "arms" represent different task categories, defined by data priors and dataset dimensionality. We then dynamically allocate sampling resources to the categories that yield the highest *reward*, which we design to capture "learning utility". The details of our reward function are deferred to the next section. We outline our curriculum strategy below, with detailed pseudocode in Appx. Algo. 1.

First, we define $P \cdot K$ task categories based on $P$ distinct priors and $K$ dimensionality bins, assigning each category $c$ an initial equal weight $Q_0(c)$. At each training step $t$, a category (pair of prior and dimension bin) is sampled with probability proportional to its temperature-scaled[5] current weight. A dataset is generated from the selected prior with dimensionality sampled uniformly from the selected bin's range. This process is repeated until the desired batch size is reached. Point-wise losses are computed based on current model parameters. The algorithm then calculates a *reward* for each dataset category, which measures its utility based on the dispersion of its point-wise losses. These rewards are used to update the category weights, ensuring that the model focuses more on categories with higher utility over time. Finally, gradients are back-propagated from those points that meet a time-varying loss threshold, dictated by a pace scheduler. Appx. D provides further details.

## 4.2. Modeling Reward

Consider $n_c$ training points belonging to category $c$ in a batch, with point-wise rewards $r_1, \ldots, r_{n_c}$ at any given epoch. What reward should one use for individual points?

---

[2]Note that in our context *sample* refers to a dataset, while *point* refers to individual examples within a dataset.

[3]Appx. Fig. 10 further shows that GMM loss is higher under naïve mixed prior training than GMM-only training.

[4]Our MAB setup is non-stationary: expected reward distribution of arms shifts as model parameters are updated during training.

[5]Temperature scaling is critical; as overly sharp distributions limit exploration and may lead to forgetting. See Appx. G.3.1.

How should one utilize point-wise rewards into a category-level reward to guide dataset sampling for the next batch?

We use the centered cross-entropy loss $l_i$ as point-wise reward $r_i$, and define category-level reward $r(c)$ as

$$r(c) = \frac{1}{n_c} r_i^2 = \frac{1}{n_c}(l_i - \text{mean}(\{l_1, \ldots, l_{n_c}\}))^2 , \quad (1)$$

which is effectively the variance of point-wise rewards. It is easy to see that $r(c)$ is maximized when $r_i = 0$ for half of the points (i.e., highly certain but wrong predictions) and $r_i = -\log(\approx 0)$ for the other half (i.e., highly certain and accurate predictions). In essence, it prioritizes datasets with highly (1) dispersed point-wise rewards and (2) confident prediction probabilities. Datasets where most samples receive consistently low rewards (overly difficult), consistently high rewards (overly easy), or non-extreme rewards due to uncertainty in probability estimates are downweighted.

In fact, category-level reward $r(c)$ becomes zero when the class probabilities across all points are $0.5$. While avoiding high-uncertainty datasets during training may initially appear counterproductive, we note that this arises from data uncertainty rather than model uncertainty. Since inherent data ambiguity or noise cannot be mitigated by more training data, our reward function effectively downweights datasets with high data uncertainty. This highlights why continuous point-wise loss is preferable to binary loss, as the latter lacks probabilistic information. (See Appx. G.3.2.)

## 5. Experiments

**Existing and New OD Benchmarks.** We extensively evaluate OUTFORMER on *hundreds of datasets from 3 real-world benchmarks*: (1) ADBench (Han et al., 2022) is the *de facto* OD benchmark with 57 datasets. We introduce two new OD benchmarks, (2) OddBench and (3) OvRBench, with 690 and 756 datasets respectively, enabling a more comprehensive evaluation with greater statistical power. In a nutshell, OddBench curates real-world tables from Tablib (Eggert et al., 2023) with metadata indicating semantic anomalies ("fraud", "failure", "defect", etc.). OvRBench re-purposes classification benchmarks (TabArena (Erickson et al., 2025), TabRepo (Salinas & Erickson, 2024), TabZilla (McElfresh et al., 2023), etc.) by selecting one class as inliers and subsampling the rest as outliers. Together, they provide a **large-scale testbed** comprising diverse real-world domains. See Appx. E.1 for details. We also report results on (4) SynBench with 800 in-distribution datasets from our priors.

**Baselines and Hyperparameters (HPs).** We compare OUTFORMER with a long list of OD baselines (Appx. Table 12), comprising classical shallow, modern deep, and the latest foundation models (FM). Appx. Table 13 lists all HP settings. While FOMO-0D (Shen et al., 2025) is the only FM for tabular OD, we also re-purpose TabPFN (Hollmann

et al., 2023) for OD, by designating each feature as a prediction target and averaging prediction errors across features to derive an outlier score. See Appx. E.2 for details.

OUTFORMER **Architecture, Training and Inference.** We use a 10-layer Transformer (Vaswani et al., 2017), 512 hidden dimensions, 8 attention heads, and a linear input embedding layer; comprising $45.1M$ parameters in total. It is trained on 4 NVIDIA RTX A6000 GPUs for 1,500 batches of 1K synthetic datasets each, sampled according to our curriculum. Each dataset contains up to 5K inliers as context, and 10K query points with an equal number of inliers and outliers with up to 100 dimensions. At inference, we ensemble outlier scores over 50 randomly subsampled context points and dimensions. We validate SEC HPs on hold-out synthetic data. See Appx. E.3 for further details.

**Metrics and Tests.** We evaluate methods w.r.t. both AUROC and AUPRC on each dataset. As aggregating performance is not meaningful across hundreds of datasets that vary in difficulty, we report five metrics that better capture relative performance across datasets: average rank, ELO (Elo, 1967), Winrate, rescaled AUROC (rAUC), and Champion delta ($C_\Delta$) (Zhang et al., 2025a). All results w.r.t. AUPRC are in Appx. F.5. We also perform pairwise tests to statistically compare two methods and report p-value based on the permutation test. See Appx. E.4 for details.

*Table 4.* **Compared on ALL 1500+ datasets from all 3 OD benchmarks,** OUTFORMER **performs on par with the SOTA.** We show the overall merged results and give separate benchmark results in Appx. F.2: Table 15 (ADBench), Table 16 (OddBench), Table 17 (OvRBench) due to space limits. **Best** and second-best are highlighted. Reported are five performance metrics across all baselines on all datasets w.r.t. AUROC; Win/Tie is the fraction of datasets where OUTFORMER wins/ties against baseline. $p$-values $\leq 0.05$ of the paired permutation test (Appx. F.3) between the baseline and OUTFORMER indicate OUTFORMER's performance is significantly better. Notably, corresponding results in **Appx. Table 25 w.r.t. AUPRC show that** OUTFORMER **significantly outperforms ALL baselines** ($p \leq 0.00$).

| Model | Avg. Rank ($\downarrow$) | ELO ($\uparrow$) | Winrate ($\uparrow$) | rAUC ($\uparrow$) | $C_\Delta$ ($\downarrow$) | Win/Tie | p-val. |
|---|---|---|---|---|---|---|---|
| DTE-NP | $5.04_{\pm 2.9}$ | 1100 | **0.61** | $\mathbf{0.905_{\pm 0.13}}$ | 0.35 | **0.43**/0.16 | 0.63 |
| kNN | $5.41_{\pm 2.9}$ | 1129 | 0.57 | $0.899_{\pm 0.13}$ | 0.35 | **0.45**/0.14 | 0.09 |
| LOF | $6.17_{\pm 3.4}$ | 864 | 0.51 | $0.869_{\pm 0.15}$ | 0.42 | **0.53**/0.12 | **0.00** |
| IForest | $6.57_{\pm 3.6}$ | 917 | 0.49 | $0.883_{\pm 0.12}$ | 0.43 | **0.55**/0.07 | **0.00** |
| DTE-C | $5.94_{\pm 3.2}$ | 1167 | 0.38 | $0.885_{\pm 0.14}$ | 0.37 | **0.51**/0.14 | **0.00** |
| ICL | $6.40_{\pm 3.3}$ | 1000 | 0.49 | $0.871_{\pm 0.14}$ | 0.39 | **0.54**/0.13 | **0.00** |
| DDPM | $9.18_{\pm 2.6}$ | 770 | 0.24 | $0.796_{\pm 0.14}$ | 0.53 | **0.77**/0.08 | **0.00** |
| GOAD | $8.72_{\pm 3.6}$ | 696 | 0.28 | $0.752_{\pm 0.23}$ | 0.51 | **0.71**/0.09 | **0.00** |
| DeepSVDD | $6.34_{\pm 3.2}$ | 994 | 0.49 | $0.869_{\pm 0.15}$ | 0.40 | **0.55**/0.15 | **0.00** |
| TabPFN-OD | $5.39_{\pm 3.4}$ | 1141 | 0.58 | $0.901_{\pm 0.13}$ | 0.34 | **0.45**/0.15 | 0.21 |
| FOMO-0D | $6.45_{\pm 3.9}$ | 1012 | 0.49 | $0.869_{\pm 0.15}$ | 0.39 | **0.55**/0.13 | **0.00** |
| OUTFORMER | $5.06_{\pm 3.3}$ | **1209** | 0.60 | $0.903_{\pm 0.12}$ | **0.32** | – | – |

## 5.1. Main Results

Table 1 shows that OUTFORMER achieves the highest performance on the prominent ADBench, outperforming all baselines, including FOMO-0D and the previous SOTA DTE-NP. Simultaneously, it offers desirable inference speed as Fig. 1 shows. Repurposed TabPFN-OD ranks second, underscoring the prowess of FMs for OD, however it requires considerable time to iterate over feature predictions.

Table 4 shows similar results over 1500+ datasets across all three real-world benchmarks: OUTFORMER achieves state-of-the-art performance, significantly outperforming most baselines (including FOMO-0D) and tying with DTE-NP, KNN and TabPFN-OD at the top rank (see Appx. Fig. 11). Even more notably, **corresponding results w.r.t. AUPRC in Appx. Table 25 show that** OUTFORMER **outperforms all baselines significantly** ($p \leq 0.05$), including DTE-NP and TabPFN-OD. (Results on separate benchmarks w.r.t. AUPRC: Table 22 (ADBench), Table 23 (OddBench), Table 24 (OvRBench); also see Appx. Fig. 13.)

*Table 6.* Comparison of OUTFORMER variants across three benchmarks. OUTFORMER **surpasses all** FOMO-0D **variants, underscoring each component's importance.** Best highlighted.

| Model | Avg. (↓) Rank | (↑) ELO | Win (↑) rate | (↑) rAUC | (↓) $C_\Delta$ | p-val. |
|---|---|---|---|---|---|---|
| FOMO-0D (L4, GMM) | $5.01_{\pm 2.4}$ | 946 | 0.40 | $0.876_{\pm 0.16}$ | 0.35 | **0.00** |
| FOMO-0D (L4, Mix.) | $5.38_{\pm 2.5}$ | 746 | 0.35 | $0.864_{\pm 0.16}$ | 0.37 | **0.00** |
| FOMO-0D (L4, Mix. w. SEC) | $4.78_{\pm 2.3}$ | 1034 | 0.43 | $0.891_{\pm 0.14}$ | 0.33 | **0.00** |
| OUTFORMER | $3.55_{\pm 2.2}$ | 1122 | 0.59 | $0.935_{\pm 0.10}$ | 0.23 | - |
| OUTFORMER w.o. Ens | $4.19_{\pm 2.1}$ | 1065 | 0.50 | $0.917_{\pm 0.11}$ | 0.29 | **0.00** |
| OUTFORMER w.o. SEC | $3.65_{\pm 2.0}$ | 954 | 0.56 | $0.922_{\pm 0.12}$ | 0.24 | **0.00** |
| OUTFORMER w.o SEC&Ens | $4.23_{\pm 2.2}$ | 921 | 0.49 | $0.908_{\pm 0.12}$ | 0.29 | **0.00** |
| OUTFORMER w.o Mix.&SEC | $4.16_{\pm 2.5}$ | 1113 | 0.52 | $0.913_{\pm 0.12}$ | 0.26 | **0.00** |

Table 6 shows the improvements we achieved over the original FOMO-0D, leading to OUTFORMER. Training with mixed priors (Mix.) alone initially harms performance, but training on mixed priors with our curriculum (SEC) results in a significant boost. Further scaling up the model size and incorporating context ensembling (Ens.) yield OUTFORMER with superior performance. Ablating either ensembling, SEC, or both leads to a performance drop, yet OUTFORMER still outperforms all FOMO-0D variants. Ablating both mixed priors (Mix.) and SEC, i.e. simply scaling up GMM-based FOMO-0D plus ensembling, significantly underperforms OUTFORMER, underscoring our key contributions. Appx. Fig. 12 shows paired tests, where OUTFORMER outperforms all variants significantly ($p \leq 0.00$).

## 5.2. Ablation Analyses

**Ablation on Curriculum Strategies:** Our work employs a carefully-designed curriculum for training OUTFORMER. We compare it to the following alternatives: **1. Naïve** samples all priors in the mixture uniformly at random per batch;

*Table 7.* **All priors contribute to** OUTFORMER**'s performance.**

| | ADBench | | | | | SynBench |
|---|---|---|---|---|---|---|
| Model | Avg. (↓) Rank | (↑) ELO | Win (↑) rate | (↑) rAUC | (↓) $C_\Delta$ | Avg. (↑) AUROC |
| OUTFORMER | $2.26_{\pm 1.7}$ | 1083 | 0.73 | $0.986_{\pm 0.03}$ | 0.13 | $0.973_{\pm 0.04}$ |
| – Ens. | $3.47_{\pm 1.6}$ | 1080 | 0.53 | $0.954_{\pm 0.06}$ | 0.29 | $0.971_{\pm 0.05}$ |
| – Ens. & SCM-Struct. | $4.04_{\pm 2.0}$ | 981 | 0.47 | $0.951_{\pm 0.06}$ | 0.30 | $0.972_{\pm 0.04}$ |
| – Ens. & SCM-Meas. | $4.60_{\pm 1.8}$ | 976 | 0.39 | $0.940_{\pm 0.07}$ | 0.34 | $0.971_{\pm 0.04}$ |
| – Ens. & Copula-Prob. | $4.28_{\pm 2.0}$ | 1041 | 0.44 | $0.942_{\pm 0.07}$ | 0.30 | $0.950_{\pm 0.07}$ |
| – Ens. & Copula-Dep. | $4.34_{\pm 2.1}$ | 941 | 0.42 | $0.941_{\pm 0.07}$ | 0.32 | $0.949_{\pm 0.07}$ |
| – Ens. & GMM | $4.79_{\pm 1.9}$ | 974 | 0.36 | $0.929_{\pm 0.09}$ | 0.34 | $0.940_{\pm 0.06}$ |

**2. Anti-Curriculum (AC)** goes after hard cases and prioritizes priors with larger gradients to reduce loss fast (Li et al., 2025); **3. Self-paced Learning (SPL)** (Jiang et al., 2015) considers point-wise loss to sort all points across datasets in a batch, and propagates gradients from points with lower loss irrespective of their prior or dimensionality; **4. Manual** dictates hand-crafted sampling probabilities or order on tasks. We implement two versions: **Manual1** samples high-dimensional GMMs with high probability and the rest uniformly per batch. **Manual2** orders the priors by diagonal values in Table 2 from high to low (easy-to-hard), pairs them with low-to-high dimensions to dictate a partial order, and assigns high sampling probability in a round-robin fashion as training progresses. See details in Appx. G.1.

Table 5 shows that OUTFORMER performs subpar when trained using alternative curriculum strategies. While SPL supports comparable in-distribution learning, it does not generalize to real-world data as well as SEC. Manual curricula provide only slight progress over Naïve, while AC even underperforms Naïve as simple yet promising priors like GMM get dominated by others with large gradients.

**Ablation of Priors:** Our results demonstrate the benefit of pretraining on our extended mixture of priors as a whole. We conduct an additional ablation that isolates the effect of each prior by excluding it from training while retaining all others. Table 7 shows the effect of removing each prior independently. Notice that removing GMM has the largest impact on OUTFORMER's performance on ADBench as well as on SynBench. Although SCM based priors cause little performance degradation on SynBench when removed, they are nonetheless critical for improving OUTFORMER's performance on ADBench. Overall, the analyses demonstrate that diversity in pretraining data distributions improves real-world generalization, motivating the exploration of additional synthetic priors in future work.

**Ablation of Hyperparameters in SEC:** Table 30 shows the ablation of different bin sizes in SEC training, where the baseline is No bins (No SEC). Introducing SEC bins improves OUTFORMER 's generalization. Increasing the number of bins from 1 to 5 yields the best overall performance, suggesting that moderate binning provides finer curriculum control and better separates task difficulty. The

*Table 5.* Comparison of different curriculum strategies. OUTFORMER **trained using SEC(Ours) strategy outperforms its variants trained with alternative curricula.** Metrics are evaluated on ADBench (real-world), and the last column reports overall AUROC on SynBench (in-distribution). Best results are highlighted.

| Model | ADBench | | | | | SynBench |
| | Avg. Rank ($\downarrow$) | ELO ($\uparrow$) | Winrate ($\uparrow$) | rAUC ($\uparrow$) | $C_\Delta$ ($\downarrow$) | Avg. AUROC ($\uparrow$) |
|---|---|---|---|---|---|---|
| SEC (Ours) | $2.30_{\pm 1.4}$ | 1115 | 0.73 | $0.967_{\pm 0.07}$ | 0.13 | $0.967_{\pm 0.06}$ |
| SPL | $2.72_{\pm 1.3}$ | 1094 | 0.64 | $0.963_{\pm 0.05}$ | 0.24 | $0.963_{\pm 0.05}$ |
| Manual1 | $3.49_{\pm 1.4}$ | 996 | 0.49 | $0.942_{\pm 0.08}$ | 0.29 | $0.941_{\pm 0.05}$ |
| Manual2 | $3.37_{\pm 1.3}$ | 1026 | 0.52 | $0.945_{\pm 0.07}$ | 0.28 | $0.945_{\pm 0.05}$ |
| AC | $5.44_{\pm 1.4}$ | 705 | 0.11 | $0.737_{\pm 0.21}$ | 0.57 | $0.912_{\pm 0.14}$ |
| Naïve | $3.53_{\pm 1.4}$ | 1062 | 0.48 | $0.943_{\pm 0.07}$ | 0.27 | $0.930_{\pm 0.05}$ |

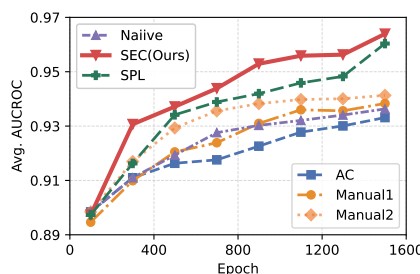

*Figure 3.* Comparison of curriculum strategies w.r.t. in-distribution generalization on SynBench. Our SEC provides best results during the course of training across epochs.

10-bin setting remains competitive, but performance starts to degrade when the number of bins becomes too large. With only 1,000 datasets per epoch, large bin counts produce too few samples per task category, leading to noisy reward estimates and weaker learning signals.

**Ablation of Inference Latency vs. Num. of Ensembles:** We evaluate the inference latency and performance trade-off under different ensemble sizes on ADBench. Table 31 and Table 32 show that latency increases sub-linearly with ensemble size due to batching and GPU parallelization, while rAUC improves substantially on medium and large datasets because ensembling provides more diverse context samples. Overall, ensembles offer a favorable balance between accuracy and inference cost.

### 5.3. Generalization to Categorical Data

For categorical anomaly detection, we evaluate OUT-FORMER under two practical input strategies: (i) one-hot with subsampling, where categorical variables are one-hot encoded, treated as numerical features, and subsampled to match OUTFORMER 's feature limit; and (ii) feature projection, where we apply the XStream (Manzoor et al., 2018) projection method to compress categorical representations into 100 numeric features. We utilize IEEE-CIS-Fraud (Howard et al., 2019) as a real-world mixed fraud dataset and re-purpose classification datasets from TALENT (Liu et al., 2025b) as OD tasks. See details in Appx. H.

Table 34 shows the comparison between OUTFORMER and IForest. OUTFORMER without projection outperforms IForest on 5 out of 7 datasets, suggesting that OUTFORMER generalizes reasonably well to categorical-heavy settings despite lacking such priors in pre-training. While OUT-FORMER exhibits a promising transfer to categorical OD, the result also motivates dedicated categorical priors and model scaling as important future directions.

## 6. Conclusion

Foundation models (FMs), as in other areas of ML, have been transformative for outlier detection (OD), offering: (1) zero effort, zero-shot inference on new OD tasks—*requiring zero additional training, tuning or labeled outliers*—enabling plug-and-play deployment in practice; and (2) speedy inference through forward pass only. This work introduced OUTFORMER, advancing FOMO-0D; the latest FM for OD (Shen et al., 2025). OUTFORMER is trained on an extended mixture of inlier/outlier priors, using a carefully-designed self-evolving curriculum that may be of more general independent interest. Extensive experiments on hundreds of real-world datasets showed that OUTFORMER achieves state-of-the-art performance, while simultaneously maintaining speedy inference. Our work establishes FMs as a fertile and promising frontier for the future of OD.

## Impact Statement

This paper aims to advance the field of outlier detection (OD) and tabular foundation models, targeting applications in diverse real-world domains such as finance, medicine, manufacturing, and environmental monitoring, as well as societal applications including policing, content filtering, and account blocking. Outlier detection is inherently unsupervised, as outliers are rare and constantly evolving, particularly in adversarial settings. This makes algorithm selection and hyperparameter tuning especially challenging. Foundation models for OD promise zero-effort zero-shot inference, eliminating the need for model training, hyperparameter tuning, or access to labeled outliers. This greatly reduces the primary barrier to practical deployment, making OD systems more accessible for real-world applications. Although attractive from a practical point of view, our proposed OUT-FORMER primarily focuses on detection performance and inference latency, without accounting for important considerations such as fairness, bias, or other potential blind spots. In sensitive real-world scenarios, the societal and ethical costs of incorrect detections must be carefully considered.

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

# Appendix

# A. Extended Related Work

**Outlier Detection (OD):** As outliers may indicate real-world anomalies such as faults, failures, errors, defects, attacks, etc., OD finds numerous practical applications across many domains (Aggarwal, 2013). OD is characteristically unsupervised, in two possible scenarios. First, any labeled outliers are absent as they are rare and/or unknown apriori. Second, we aim not to fit to or learn from limited (again, anomalies are rare) *known* outliers but to detect novel, emerging outliers where outliers continually change especially in adversarial settings as the adversaries switch tactics. As such, related areas include novelty and out-of-distribution detection (Yang et al., 2024) with differences highlighted by Salehi et al. (2022).

Unsupervised outlier model selection, or UOMS as coined by Zhao et al. (2021), is a bottleneck for effective practical use of OD: Given a new task, there exists a vast body of methods to choose from, and most are sensitive to the choice of their hyperparameter (HP) values (Campos et al., 2016; Ding et al., 2022; Yoo et al., 2023). This is especially true for modern deep learning based OD models with many more HPs (architectural, regularization, and optimization HPs) than their classical shallow counterparts (e.g. $k$ in kNN (Ramaswamy et al., 2000)). On the other hand, UOMS remains notoriously hard; Ma et al. (2023) provide a meta-analysis of various intrinsic heuristic measures proposed for UOMS and find that they are not significantly different from random selection. They also find that while consensus based approaches do better, they cannot significantly outperform the classical, speedy Isolation Forest (Liu et al., 2008) algorithm with *default* HPs despite being computationally more demanding.

An effective alternative to UOMS is outlier ensembles (Zimek et al., 2014; Aggarwal & Sathe, 2017). For example, the popular Isolation Forest (Liu et al., 2008) algorithm is a bagged randomized tree ensemble, which is a prominent practical choice thanks to its robustness to HP choices (Ma et al., 2023). Other ensembling techniques for OD include subsampling (Zimek et al., 2013), feature bagging (Lazarevic & Kumar, 2005), selective ensembles (Rayana & Akoglu, 2016), as well as sequential ensembles (Rayana et al., 2016). In this work, we leverage the former two, subsampling and feature bagging, for ensembling scores over randomized contexts thanks to their simplicity and low latency.

**Tabular Foundation Models (FMs):** Inspired by the growing success of large language models, tabular foundation models have made their first appearance for classification and regression, i.e. supervised learning tasks (Hollmann et al., 2023). Due to their remarkable success, several follow-up works introduced further advances in architecture, pretraining data, and scalability (Hollmann et al., 2025; Qu et al., 2025; Grinsztajn et al., 2025; Ma et al., 2025; Zhang et al., 2025a; Bühler et al.; Feuer et al., 2024; Ma et al., 2024; Wang et al., 2025).

Tabular FMs have subsequently been extended to other data modalities, including time series forecasting (Dooley et al., 2023; Taga et al., 2025), subgraph classification (Fey et al., 2025), causal inference (Robertson et al., 2025b; Balazadeh et al., 2025), counterfactual fairness (Robertson et al., 2025a), as well as outlier detection (Shen et al., 2025). Recent work also explore FMs for anomaly detection in images (Gu et al., 2024), graphs (Qiao et al., 2025), as well as time series (Lan et al., 2025), while FoMo-0D (Shen et al., 2025) is the only FM for OD on tabular data. In this work, we build on FoMo-0D, advancing the state-of-the-art on zero-shot tabular OD performance with FMs.

**FMs for OD:** The power of FMs for OD extends beyond zero-shot inference without any labeled outliers: Although OD is widely needed in real-world domains, it is inherently unsupervised as outliers are rare and ever-changing—making both algorithm choice and hyperparameter tuning particularly challenging. FMs for OD, if designed and trained successfully, would entirely eliminate both challenges, lifting the primary barrier to practical deployment of OD.

Shen et al. (2025) introduced the *first and only foundation model for tabular OD* called FoMo-0D, which is a vanilla Transformer architecture (Vaswani et al., 2017) pretrained on synthetic datasets from a simple prior based on mixtures of Gaussians. Despite its simplicity, FoMo-0D achieved remarkable performance on real-world datasets from the prominents ADBench OD benchmark (Han et al., 2022). This work advances FoMo-0D with an extended mixture of data priors; capturing an enhanced diversity of inlier/outlier distributions, as well as a self-evolving curriculum strategy for pretraining; enabling effective learning of the priors and better generalization to real-world datasets.

**Curriculum Learning.** Curriculum learning is inspired by human pedagogy: models are first exposed to simpler examples before progressively harder ones, with the goal of facilitating optimization and generalization (Bengio et al., 2009). Extending curriculum learning beyond single task paradigms, Pentina et al. (2015) demonstrated that sequentially training on multiple tasks, ordered from easier to harder, can yield improved performance compared to joint training. Their results highlighted that task order matters: solving easier tasks first establishes inductive biases that bootstrap learning on subsequent tasks.

Work in multi-task learning has shown that conflicting gradients between tasks that point in opposing directions can hinder joint optimization, thereby degrading both convergence speed and final performance (Standley et al., 2020). To address this, several approaches proposed ways to reconcile or re-weight gradients. To this end, PCGrad projects conflicting gradients to avoid interference (Yu et al., 2020), while GradNorm adaptively balances task losses based on their relative difficulty and learning rates (Chen et al., 2018).

Anti-curriculum strategies, in contrast, prioritize hard examples from the start. Hard example mining (Shrivastava et al., 2016) identifies and up-weights high-loss samples. Li et al. (2025) dynamically adjust sampling probabilities to emphasize challenging samples with high loss or large gradients, aiming to accelerate convergence under the assumption of low gradient conflicts. However, under gradient interference, anti-curriculum strategies may impede performance, as our worked empirically showed. Similarly, recent work in this vein by Kotha et al. (2025) showed that reducing the diversity of pretraining data decreases gradient interference.

Unlike many prior curriculum frameworks, our setting imposes a layered heterogeneity across datasets that differ along multiple axes: data priors and thus distributional biases, input dimensionality, and point-wise differences in sample difficulty. Our multi-armed bandit approach addresses aspects of dataset heterogeneity, by modeling variations in dataset priors and dimensionality with different arms representing distinct dataset categories, and allowing the training process to adaptively prioritize sources that maximize learning progress.

# B. Limitations and Future Directions

Our work sets the stage for future foundation models for OD, showcasing their outstanding potential and practical implications, while leaving ample room for future breakthroughs. We discuss limitations with respect to three levers: (1) pretraining data, (2) training and architecture, and (3) context optimization.

While our inlier/outlier priors are from diverse families, we foresee that additional synthetic priors (especially those for categorical data) as well as generative models learned from real-world data could continue to improve performance. On the other hand, a deeper investigation of training dynamics can reveal further insights toward more effective (pre)training. Our model architecture employs only sample-to-sample (1D) attention coupled with a projection layer of features into row-wise embeddings, whereas dual (2D) attention, i.e. sample-to-sample as well as feature-to-feature, coupled with cell-wise representations could further improve performance; as with TabPFN v1 vs. v2 (Hollmann et al., 2023; 2025). Our model adopts a vanilla Transformer backbone (Vaswani et al., 2017), whereas architectural innovations resembling mixture-of-experts (Jacobs et al., 1991; Fedus et al., 2022) or conditional generation (Keskar et al., 2019) may better cater to training with mixture of priors. Finally, our model is limited by its context size to datasets of certain size and dimensionality, due to which we employ bagged ensembling. Together with scalable alternatives (Ma et al., 2024; Feuer et al., 2024) that help extend context length, additional innovations on context optimization are likely to boost performance, although any form of optimization faces two challenges: the lack of labeled outliers, and an increased inference-time latency.

# C. Details on Data Priors

Our work introduces synthetic inlier/outlier data priors based on three different families, yielding five different outlier archetypes. We provide additional details on our priors as follows. All code for our synthetic prior generators are open sourced at `https://github.com/psorus/Outformer.git`.

### C.1. Gaussian Mixtures

**Inliers:** We simulate inliers by drawing from multivariate GMMs with varying number of components and dimensions. $m$-components in $d$ dimensions; with centroids $\boldsymbol{\mu}_j^{(k)} \in [-5, 5]$ and diagonal covariance $\boldsymbol{\Sigma}_{jj}^{(k)} \in (0, 5]$ for $k \in [m]$ and $j \in [d]$. We vary $m \in [M]$ and $d \in [D]$ uniformly as we draw new datasets, with $M = 5$ and $D = 100$.

Linear transformation of a GMM dataset follows $T(\mathbf{x}) = \mathbf{W}\mathbf{x} + \mathbf{b}$, where $\mathbf{W} \in \mathbb{R}^{d \times d}$ and $\mathbf{b} \in \mathbb{R}^d$, and have entries sampled independently from the uniform distribution on $[-1, 1]$. This simple and efficient transformation creates a new GMM dataset as if drawn with centers $T(\boldsymbol{\mu}_j) = \mathbf{W}\boldsymbol{\mu}^{(k)} + \mathbf{b}$ and *non-diagonal* covariance $T(\boldsymbol{\Sigma}_j) = \mathbf{W}\boldsymbol{\Sigma}^{(k)}\mathbf{W}^T, \forall k \in [m]$. Importantly, we do not materialize the transformed mixture parameters; we only apply the transformation to the sampled data points. These transformations preserve the Mahalanobis distances as well as the percentiles of the original GMM (Shen et al., 2025).

**Outliers:** We generate *contextual subspace outliers* by randomly selecting a GMM component $k$ and a subset of dimensions $\mathcal{S}$ and "inflating" the variances as $s \times \Sigma_{jj}^{(k)}$ for $j \in \mathcal{S}$ while keeping the centroids as before. We then sample points from the inflated GMM and designate outliers as those points outside the $90th$ percentile of the original GMM based on their Mahalanobis distance. We vary the severity of outliers based on $|\mathcal{S}| = \alpha d$ and $s$, the subspace size and the inflation scale, respectively.

Table 8 provides the range of hyperparameter configurations for sampling datasets from the GMM prior.

*Table 8.* Hyperparameters for Gaussian Mixture Model (GMM) Data Prior

| Hyperparameter | Values | Description |
|---|---|---|
| $m$ | $[1, 5]$ | Number of mixture components |
| $d$ | $[2, 100]$ | Dimensionality of data |
| $\boldsymbol{\mu}_j^{(k)}$ | $[-5, 5]$ | Component means |
| $\boldsymbol{\Sigma}_{jj}^{(k)}$ | $(0, 5]$ | Diagonal variances |
| $\alpha$ | $[\frac{1}{d}, 1]$ | Fraction of dimensions for variance inflation |
| $s$ | $[5, 10]$ | Variance inflation factor for subspace outliers |
| $r$ | $[0.02, 0.2]$ | Outlier rate of contamination |

## C.2. Structural Causal Models

**Inliers**: Structural causal models (SCMs) represent the generative dependencies among variables through causal graphs and structural equations. An SCM is defined by a directed acyclic graph $G = (V, E)$, where each node $j \in V$ corresponds to a variable, and causal mechanisms are specified as

$$X_j = f_j\big(X_{\mathrm{Pa}(X_j;G)}, \epsilon_j\big),$$

where $\mathrm{Pa}(X_j; G)$ denotes the set of parent (direct cause) variables of $X_j$ in $G$, and $\epsilon_j$ represents exogenous noise. For SCM inliers, We first instantiate the causal graph $G$ using a multilayer perceptron (MLP), and then apply a weight mask with a ratio in the range $(0.4, 0.6)$ to induce sparsity, selecting a subset of $d$ nodes and thereby producing $d$-dimensional vectors. Nonlinear causal relationships between nodes are modeled using an activation function $a(\cdot)$. Once a MLP is created, the weights are The SCM structure is frozen to simulate a fixed causal mechanism. Inliers are generated by propagating inputs through the MLP and are subsequently flattened into multi-dimensional vectors.

**Measurement Outliers:** To generate each measurement outlier, we randomly select a variable $X_j \in \mathcal{X}$, sample its exogenous noise $\epsilon_j \sim \mathcal{N}(0, s)$ with inflated variance (inflating variance by $s$), and allow the resulting perturbation to propagate to its descendant variables according to the underlying structural dependencies, yielding a multivariate outlier. This procedure models exogenous shocks while preserving the underlying causal structure and mechanisms. For each SCM, we randomly generate outliers by inflating a fraction $r$ of the total nodes.

**Structural Outliers:** Structural outliers correspond to changes in the underlying causal relationships. We introduce such outliers masking the weights on the MLP, either by *breaking* a causal edge (i.e., setting its weight to zero) or reversing its direction (times the weight with -1) . These interventions simulate intrinsic changes in parent–child relationships, thereby altering the data-generating mechanisms. We set the total edge-perturbation probability to 0.2 (0.1 for *breaking* and 0.1 for *reversing*), and impose a filtering criterion to ensure that at least one leaf node is affected by the intervention.

Fig. 4 provides an example of generated SCMs, and Table 9 provides the hyperparameter settings of the SCM priors.

## C.3. Copulas

**Inliers:** Real-world univariate features often exhibit skewed, heavy-tailed behavior (Clauset et al., 2009). While SCMs induce non-Gaussian marginals, they do not provide explicit control over feature-wise distributions; we therefore adopt copula models (Nelsen, 2006; Houssou et al., 2022). By Sklar's theorem (Sklar, 1959), any joint CDF of continuous variables $\{X_j\}_{j=1}^d$ with marginals $F_j$ can be expressed as

$$F(x_1, \ldots, x_d) = C(F_1(x_1), \ldots, F_d(x_d)),$$

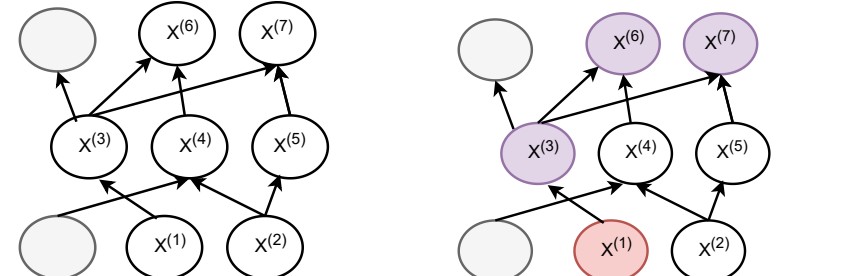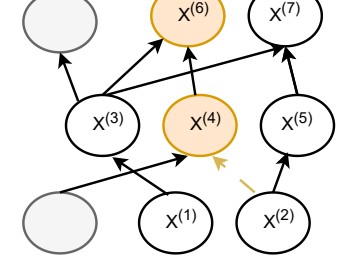

*Figure 4.* **Left:** We construct an SCM and select a subset of seven nodes, from which 7-dimensional inlier samples are generated under a frozen causal structure. **Middle:** Measurement outliers are produced by inflating the variance of node $x^{(1)}$, causing the perturbation to propagate to $x^{(3)}$, $x^{(6)}$, and $x^{(7)}$. **Right:** Structural outliers are generated by breaking the causal link between $x^{(2)}$ and $x^{(4)}$, thereby altering the dependency node $x^{(6)}$.

*Table 9.* Hyperparameters for Structural Causal Model (SCM) based Data Priors

| Hyperparameter | Values | Description |
|---|---|---|
| MLP depth | $[3, 5]$ | Number of layers used to generate the DAG |
| MLP width | $[20, 40]$ | Number of nodes per layer |
| Edge-drop rate | $[0.4, 0.6]$ | Fraction of edges randomly removed to form the DAG |
| $d$ | $[2, 100]$ | Number of selected nodes as features $\mathcal{X}$ |
| $a(\cdot)$ | {ReLU, tanh, sigmoid} | Activation function for structural equations |
| $s$ | $[5, 10]$ | Variance inflation factor for measurement outliers |
| $p_{\text{break}}$ | $[0.1]$ | Probability of removing a causal edge for structural outliers |
| $p_{\text{flip}}$ | $[0.1]$ | Probability of reversing a causal edge for structural outliers |
| $r$ | $[0.02, 0.2]$ | Outlier rate of contamination |

enabling independent specification of marginals and dependence structure. We sample each feature's marginal from a diverse pool of parametric distributions (Gaussian, Beta, Exponential, Student's $t$, Power-law, and Log-logistic), and use either a Gaussian or (bivariate) vine copula for $C$. Given $C$ and $\{F_j\}_{j=1}^d$, inliers are generated by drawing $\mathbf{u} \sim C$ on $[0,1]^d$ and transforming $x_j = F_j^{-1}(u_j)$ for all $j \in [d]$. We widely vary the marginal families, their parameters, and the copula to synthesize diverse tabular inlier datasets.

**Probabilistic Outliers:** For each probabilistic outlier, we randomly select a subset of features and perturb their copula coordinates by pushing them toward the boundaries, replacing $u_j$ with very small ($u_{\text{perturb}} \in [0.1, 0.3]$) or large values ($u_{\text{perturb}} \in [0.7, 0.9]$). We keep perturbing $\gamma_{\text{perturb}} \in [0.02, 0.2]$ fraction of dimensions, while keeping at least one of the dimensions perturbed.

**Dependence Outliers:** For each dependence outlier, we modify the copula representation to violate the original dependence structure, either by inverting a subset of dimensions (i.e., setting $u_j \leftarrow 1 - u_j$) or by randomly permuting selected coordinates. We disturb $k_{\text{invcorr}} \in \left[ \left\lfloor 1 + \frac{d}{3} \right\rfloor, \min\left( \left\lfloor 1 + \frac{2d}{3} \right\rfloor, d+1 \right) \right)$ dimensions ($\sim 33\%$–$67\%$ of the total $d$).

Table 10 provides the hyperparameter settings of the Copula priors.

# D. Details on Curriculum Design

### D.1. Pseudo-code

Algorithm 1 presents the pseudocode of our proposed self-evolving curriculum (SEC) based on multi-armed bandits.

*Table 10.* Hyperparameters for Copula Data Prior

| Hyperparameter | Values | Description |
|---|---|---|
| $d$ | $[2, 100]$ | Number of features / copula dimension |
| $C$ | {Gaussian, Vine (bivariate)} | Copula parameter family |
| $\alpha_{\text{indp}}$ | $[0.1, 0.3]$ | Fraction of features set independent in copula |
| Bivariate vine | Gauss., Stu., Clayton, Gumbel, Frank, Joe | Bivariate family for each Vine edge |
| Marginal | {Gauss., Beta, Exp., Stu. t, Power-law, Log-logistic} | Marginal family for each feature |
| $\mu/\sigma$ | $(-1, 1)/(0.5, 1.0)$ | mean/variance for Gaussian |
| $a/b$ | $(1, 5)/(1, 5)$ | for Beta |
| $loc, scale$ | -5, 10 | Variables rescaled as $loc + scale \times x_{\text{beta}}$ for effective range $[-5, 5]$ |
| $\lambda / loc$ | $(0.5, 1.0)$, -5 | Exponential distribution scale / shift as $loc + x_{\text{exp}}$ for effective range |
| df | $(3, 10)$ | Degrees of freedom for Student's $t$ |
| $loc, scale$ | $[-1, 1]$, $[0.5, 1.0]$ | Variables rescaled as $loc + scale \times x_{\text{stu}}$ for effective range |
| $a / loc \ \& \ scale$ | $(0.5, 5) / $ -5 & 5 | Power law exponent, location & scale for effective range |
| $c / loc \ \& \ scale$ | $(0.5, 5) / $ -5 & 5 | Log-logistic shape, location & scale for effective range |
| $\gamma_{\text{perturb}}$ | $[0.02, 0.2]$ | Fraction of dimensions to perturb for probabilistic outliers |
| $u_{\text{perturb}}$ | $[0.1, 0.3]$ or $[0.7, 0.9]$ | CDF perturbation for probabilistic outliers |
| Copula perturbation | {inverse_corr, random_permutation} | Types of dependence outliers |
| $k_{\textbf{invcorr}}$ | $[1, 67)$ | Num of dimensions to invert for dependence outliers |

---

**Algorithm 1** Self-evolving Curriculum (SEC) Training of OUTFORMER

---

**Require:** Data priors $\{\phi_1, \ldots, \phi_P\}$; Disjoint dimension bins $\{\mathcal{B}_1, \ldots, \mathcal{B}_K\}$; total train epochs $T$; batch size $B$; sampling temperature $\tau$; pace scheduler $g_{a,b}(t)$; EMA constant $\gamma$; Number of points $n$ per dataset.

1: **Initialize:** Dataset categories $\mathcal{C} \leftarrow \{(\mathcal{B}_b, \phi_p) \mid b \in \{1, \ldots, K\}, \ p \in \{1, \ldots, P\}\}$, s.t. $|C| = K \cdot P$
2: Category weights $Q_0(c) \leftarrow \frac{1}{K \cdot P}, \forall c \in \mathcal{C}$
3: Model parameters $\boldsymbol{\theta}_0 \sim$ init_weight()
4: **for** $t = 1, \ldots, T$ **do**
5: $\quad \mathcal{D}_t \leftarrow \emptyset$
6: $\quad$ **while** $|\mathcal{D}_t| < B$ **do**
7: $\quad\quad$ Sample $c := (\mathcal{B}_b, \phi_p) \sim \text{Softmax}\left(\frac{Q_{t-1}(c)}{\tau}\right)$
8: $\quad\quad$ Sample dimension $d \sim \text{Unif}(\mathcal{B}_b)$
9: $\quad\quad$ Sample $\mathcal{D} = \{(\mathbf{x}_i, y_i)\}_{i=1}^n \sim \phi_p$ s.t. $\mathbf{x}_i \in \mathbb{R}^d$
10: $\quad\quad$ $\mathcal{D}_t \leftarrow \mathcal{D}_t \cup \mathcal{D}$
11: $\quad$ **end while**
12: $\quad$ Loss $l_{i,t} = \mathcal{L}\big(f(\mathbf{x}_i; \boldsymbol{\theta}_{t-1}), y_i\big) \ \forall (\mathbf{x}_i, y_i) \in \mathcal{D}_t$
13: $\quad$ **for** $c \in \mathcal{C}$ **do**
14: $\quad\quad$ $\mathcal{D}_c \leftarrow \{(\mathbf{x}, y) \in \mathcal{D}_t \mid (\mathbf{x}, y) \text{ from category } c\}$
15: $\quad\quad$ $\mathcal{R}_t(c) \leftarrow \{r_{i,t} := (l_{i,t} - \frac{1}{|\mathcal{D}_c|} \sum_{j:\mathbf{x}_j \in \mathcal{D}_c} l_{j,t})\}_{i=1}^{|\mathcal{D}_c|}$
16: $\quad\quad$ Reward $r_t(c) \leftarrow \frac{1}{|\mathcal{D}_c|} \sum_{i:\mathbf{x}_i \in \mathcal{D}_c} (\mathcal{R}_t(c)[i])^2$
17: $\quad\quad$ EMA Update: $Q_t(c) \leftarrow \gamma r_t(c) + (1 - \gamma) Q_{t-1}(c)$
18: $\quad$ **end for**
19: $\quad$ Set $\mathcal{D}_{\text{train}} = \{(\mathbf{x}_1, y_1), \ldots, (\mathbf{x}_{g_{a,b}(t)}, y_{g_{a,b}(t)})\}$ from Sorted($\{l_{i,t}\}$; "increasing")}
20: $\quad$ $\boldsymbol{\theta}_t \leftarrow \text{TRAINONEEPOCH}\big(\mathcal{D}_{\text{train}}; \boldsymbol{\theta}_{t-1}\big)$
21: **end for**
22: **return** Trained model parameters $\boldsymbol{\theta}_T$

---

### D.2. Design Choices

We remark on a few curriculum design choices, namely the pace scheduler (to alleviate label noise), the sampling temperature (to alleviate forgetting), and the non-binary loss based reward (to capture learning utility), as follows.

First, the pace scheduler $g(\cdot)$ (line 19) is employed to filter high-loss points. This follows the simple strategy outlined in main text, limiting large gradients from hard and/or noisy points. This is especially useful against noisy points; since our

datasets may exhibit label noise, containing hard-to-detect noisy outliers. Table 11 presents various pace functions, first proposed in (Wu et al., 2021), with curves illustrated in Fig. 5, which gradually decrease the amount of filtering, allowing harder samples to participate in training.

*Table 11.* Pacing functions $g_{(a,b)}(t)$

| Name | Expression $g_{(a,b)}(t)$ |
|---|---|
| log | $Nb + N(1-b)\big(1 + 0.1\log\big(\frac{t}{aT} + e^{-10}\big)\big)$ |
| exp | $Nb + \frac{N(1-b)}{e^{10}-1}\big(\exp\big(\frac{10t}{aT}\big) - 1\big)$ |
| step | $Nb + N\left\lceil \frac{t}{aT}(1-b)\right\rceil$ |
| linear | $Nb + N\frac{(1-b)}{aT}t$ |
| root | $Nb + N\frac{(1-b)}{(aT)^{1/2}}t^{1/2}$ |
| quadratic | $Nb + N\frac{(1-b)}{(aT)^2}t^2$ |

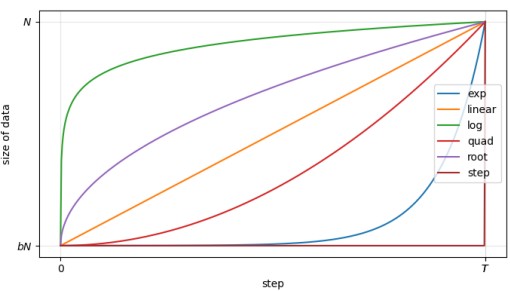

*Figure 5.* Pacing functions gradually reduce the number of filtered high-loss (hard or noisy) samples as training progresses, with varying pace.

Second, the sampling temperature $\tau$ (line 7) that controls the explore-exploit trade-off. While the goal is to prioritize high-utility datasets, sampling continues to explore in two ways: (1) to prevent forgetting of easy, already-learned tasks, and (2) to continually probe the model's performance on more challenging tasks. We validate the choice of the pacing function and sampling temperature on in-distribution validation datasets from the synthetic priors.

The third and particularly crucial modeling choice is the dataset category-level Reward (lines 15–16) that prioritizes tasks with high point-wise cross-entropy loss dispersion. We discuss the advantages of this design against binary reward in main text, specifically regarding its ability to account for data uncertainty through class probability estimates.

We also remark on the relation between our Reward score and the Advantage score used in the Group Relative Policy Optimization (GRPO) algorithm for reinforcement learning (Shao et al., 2024). There exists a key difference: unlike GRPO's standardized reward (a.k.a. Advantage), we do not standardize the point-wise rewards in Eq. (1) by dividing by $\text{std}(\{l_1, \ldots, l_{n_c}\})$. In GRPO, Advantage is used to scale policy gradient updates where standardization prevents large gradients from destabilizing training. In contrast, in our case, Reward is *not* used in model gradient updates; instead, it feeds into updating the expected reward distribution of arms in our multi-armed bandit setup. In fact, standardizing the rewards would hinder our ability to effectively quantify the degree of dispersion.

To illustrate how our SEC reward differs from a binary reward by incorporating prediction probabilities, we conduct a controlled simulation over *fraction of points with high entropy* and *dataset difficulty*. For each pair $(\phi, \chi)$ on a $20 \times 20$ grid, we generate $n = 100$ binary labels in an alternating pattern. *Fraction of points with high entropy* $\chi \in [0, 1]$ is defined as the fraction of samples assigned an uninformative prediction $\hat{p} = 0.5$, while the remaining $1 - \chi$ fraction receive confident predictions $\hat{p} \in \{0.05, 0.95\}$. *Dataset difficulty* $\phi \in [0, 1]$ is defined as the probability that a data is predicted incorrectly with high confidence. For each confident sample, we flip the label with probability $\phi$ and assign $\hat{p} = 0.95$ or $0.05$ accordingly; uncertain samples are always assigned $\hat{p} = 0.5$. We then compute the SEC reward $r([\phi, \chi])$ (Eq. 1) and a *binary reward*. The binary reward is defined as the following: given predicted probability $\hat{p}_i$, we first convert into hard labels at threshold $\tau = 0.5$ and acquire predicted label $\hat{y}_i$, then we define the per-sample binary loss as $l_{binary,i} = \mathbf{1}[\hat{y}_i = y_i]$ and $l_{binary,i} \in \{0, 1\}$, indicating whether the prediction is correct. And the binary reward for a category is defined as:

$$r_{binary}(c) = \sum_{i=1}^{n} \left| l_{binary,i} - \frac{1}{n}\sum_{j=1}^{n} l_{binary,j} \right| \tag{2}$$

which depends only on the overall accuracy and is insensitive to prediction confidence or calibration. Indeed, we visualize both rewards' heatmaps across the grids, and pick three dataset difficulty level (easy $\phi = 0$, medium with $\phi = 0.5$, and hard with $\phi = 1.0$). Fig. 7 and Fig. 6 show the differences between two rewards. The SEC reward assigns higher value to informative regimes, such as moderate entropy and hard data, while penalizing uninformative extremes. In contrast, the binary reward favors easy data under low entropy and fails to distinguish confident errors from uncertain predictions.

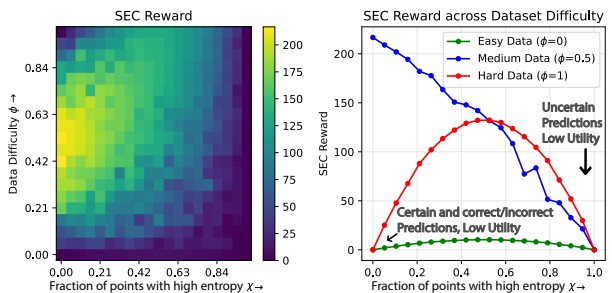

*Figure 6.* The SEC reward operates on cross-entropy loss and captures both prediction confidence and correctness. SEC reward becomes low when the model becomes certain and always provide correct/incorrect predictions, or produces constant uncertain predictions (prediction prob=0.5).

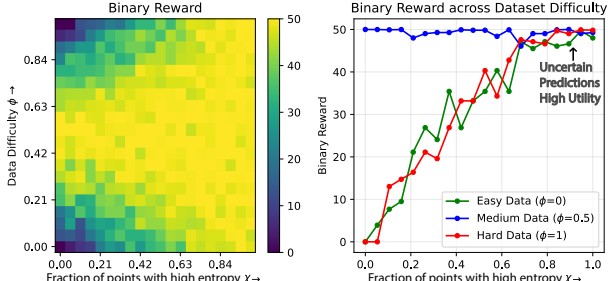

*Figure 7.* Binary rewards depend solely on prediction correctness and ignore prediction entropy. As a result, when the fraction of high-entropy points $\chi$ increases, the reward becomes indiscriminately high across easy, moderately hard, and hard datasets. Although higher data uncertainty leads to higher rewards, such uncertainty cannot be reduced through additional training, making the signal misleading for curriculum learning.

# E. Details on Experiment Setup

## E.1. Our Benchmarks: **OddBench** and **OvRBench**

The prominent *de facto* OD benchmark in the literature is ADBench (Han et al., 2022) which contains only 57 datasets. Its minuscule size limits not only its representativeness but also the statistical power of evaluation. To provide a comprehensive evaluation, we carefully curate and use two new large-scale OD benchmarks, namely OddBench and OvRBench, with hundreds of datasets each. As shown in Figure 8 (best in color), OddBench and OvRBench offer both scale and diversity toward a comprehensive evaluation for OD.

All curated datasets from both benchmarks are open-sourced at `https://github.com/psorus/Outformer.git`.

**OddBench** contains 690 *real-world* datasets associated with various abnormalities, thus containing semantic anomalies, curated from Tablib (Eggert et al., 2023). **OvRBench** contains 756 *real-world* datasets curated from Tablib (Eggert et al., 2023) as well as established classification benchmarks (TabArena (Erickson et al., 2025), TabRepo (Salinas & Erickson, 2024), TabZilla (McElfresh et al., 2023), among others), containing statistical outliers.

TabLib (Eggert et al., 2023) contributes to both benchmarks, which contain 627 million real-world tables with abundant metadata, extracted from numerous file formats, including CSV, HTML, SQLite, PDF, Excel, and others, sourced from GitHub and Common Crawl. Importantly, these tables do not represent any machine learning task, i.e., they usually do not exhibit a designated target column.

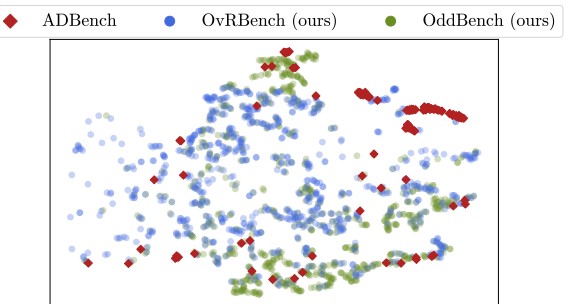

*Figure 8.* t-SNE embedding of (1446) datasets from our proposed benchmarks and (57) from ADBench. OddBench and OvRBench offer both scale and diversity as compared to the de facto ADBench, yielding a comprehensive OD testbed.

**Preliminary Filters:** Tablib only contains a collection of raw Web-sourced tables, therefore in a first step, we remove all clearly unusable tables. For each raw dataset, we first drop the columns with the highest ratio of null or infinity values, then drop any rows that contain null or infinity values. We also drop all monotonous features that might indicate indices. Afterwards, we drop all datasets that do not contain more than 1000 samples, more than 3 numerical features, or any categorical feature that can be used to assign labels.

The preliminary selection criteria filter down Tablib by more than an order of magnitude (623M to 11M datasets). From the remaining datasets, we use half to build OddBench and the remaining half for OvRBench. We elaborate on the curation process of these benchmarks as follows.

### E.1.1. ODDBENCH

**Filtering Criteria**: The characteristic feature of OddBench is that it contains semantic anomalies, where we filter Tablib datasets based on their metadata containing keywords related to real-world abnormalities. Specifically, we search for datasets where exactly one categorical feature value is associated with anomaly detection (e.g., "fraud", "failure", "defect"; see the full list below). We use samples with this feature value as anomalies and drop all other categorical values. Word clouds of these keywords are illustrated in Figure 9, showing that OddBench captures various types of semantic anomalies from a variety of real-world domains.

*Full list of keywords used to filter Tablib tables by abnormality:* [ "fraud", "intrusion", "attack", "malware", "spam", "phishing", "defect", "failure", "fault", "error", "bug", "outlier", "anomaly", "abnormal", "irregular", "rare", "deviation", "exception", "deviation", "irregularity", "abnormality", "flaw", "disturbance", "variance", "misfire", "oddity", "discrepancy", "dissonance", "unusual", "quirk", "oddity", "peculiarity", "nonconformity", "misfit", "aberration", "mistake", "fault", "glitch", "hiccup", "error", "breakdown" ]

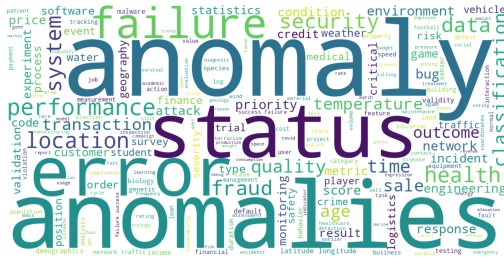
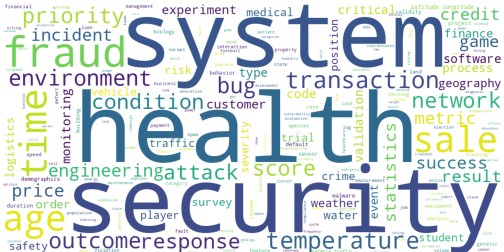

(a) OddBench reflects **semantic** anomalies.  (b) OddBench spans **real-world domains**.

*Figure 9.* Word clouds of keywords across OddBench datasets showing (a) the semantic anomalies and (b) a broad spectrum of domains covered. Full version is shown in (a), while overly frequent keywords including "anomal", "status", "outlier", "classification", "data", "performance", "failure", "quality", "location", or "error" are dropped in (b).

**Target Designation**: We select for OddBench only the datasets in which one categorical feature clearly labels samples as anomalous. A list of such keywords (e.g., "outlier", "fraud", "abnormal") are provided above. Occurrences of this feature value are then considered anomalous. Furthermore, every selected dataset is manually checked to ensure it is semantically meaningful and related to anomaly detection. If, next to an anomalous category value, there are multiple other possible values, we also manually select a suitable group of normal values.

**Hashing for Duplicates and Separability Check:** We calculate a custom hash code (detailed below) for each dataset to remove duplicate datasets and also consider only such datasets where normal samples can be separated from our anomalies. For the latter, we demand that a supervised Random Forest classifier reaches at least $60\%$ ROC-AUC.

To identify duplicate tables in Tablib, we apply a custom hashing function to each feature. When two datasets share the same hash value, we randomly drop one of these datasets. Our hashing function calculates the average of five different simple functions and considers two features to be equivalent when they share all five resulting numbers. The five functions we use are $\sin(x)$, $\cos(e \cdot x)$, $\frac{x}{1+|x|}$, $\arctan(x)$, $\log(|x| + 1)$. We design this hashing procedure in such a way that dropping or shuffling features does not resolve a hash conflict. However, small variations (e.g., NaN imputation, subset selection) still conceal hash conflicts. Thus, we also manually filter for duplicates.

**Manual Quality Control:** All remaining 2907 candidate datasets are inspected manually. For this, we first check that these datasets are really related to anomaly detection (1300/2907) and select the best-matching normal categories for each dataset. Finally, we manually remove all duplicate datasets that our earlier hash function missed (790/1300). We also keep 100 datasets private for use in a future online anomaly detection leaderboard, leaving 690 datasets in OddBench.

### E.1.2. OVRBENCH

A common approach in the literature ((Ruff et al., 2018; Han et al., 2022)) for creating anomaly detection benchmark datasets involves adapting tabular classification datasets by treating one class as normal and all others as anomalies. While we believe that the semantic anomalies of OddBench are more realistic, we still consider this approach valid and useful, e.g., for OOD detection (Yang et al., 2024). Thus, we create OvRBench ("one vs. rest" benchmark) which leverages both

commonly used classification benchmarks and the remaining Tablib datasets. We follow very similar steps to those for OddBench in the previous subsection and thus highlight only the differences in creating OvRBench.

**Data Sources and Filtering Criteria**: We start with the remaining half of the datasets that pass the preliminary filters. Additionally, we take further datasets from established tabular classification benchmarking repositories. These include (dataset count in parentheses) OpenML-CC18 (Bischl et al., 2021) (72), AutoML (Gijsbers et al., 2019) (39), TabZilla (McElfresh et al., 2023) (36), Talent-CLS (Liu et al., 2025a) (180), TabRepo (Salinas & Erickson, 2024) (246), BCCO-CLS (Zhang et al., 2025b) (106), and TabArena (Erickson et al., 2025) (37).

Similar to OddBench, we apply the following filtering steps: (1) In addition to our handling of Tablib tables, we also remove duplicate datasets across different repositories; (2) We select datasets that are related to anomaly detection based on their metadata; (3) We remove duplicate datasets as well as those datasets where anomalies are not separable from normal samples; and (4) We remove duplicate datasets that were missed in earlier steps manually.

**Target Designation and Anomaly Generation**: Datasets from existing classification benchmarks naturally include a label/target column. However, for Tablib datasets, we select only those where one feature is explicitly named "label" or "target". Further, we demand that the cardinality of possible values of this feature is between 2 and 10 and that the largest category contains at least $50\%$ of the values. We designate the largest category in the target feature as the normal class and subsample remaining classes as anomalies at a given anomaly ratio randomly drawn from the range [0.05, 0.2]. This is done to ensure that anomalies are not unrealistically common.

Finally, we also reserve $100$ datasets for a future leaderboard, leaving $758$ for OvRBench.

### E.2. Baselines and Hyperparameters: Details

Table 12 provides the list of OD baselines we have used in the experiments. For shallow models, i.e., kNN, LOF, IForest and DTE-NP, we choose to tune possible HPs. For deep models, due to computation restriction, we will compare to the default hyperparameters (HPs) that are listed either in the PyOD implementation, original paper, or (Livernoche et al., 2024) that provides a comprehensive testbed. Table 13 shows the list of hyperparameters (HPs) for classical shallow and modern deep models. For the fast classical baselines, we report average performance across a grid of hyperparameter (HP) choices, while for slower deep models we use default HPs since our evaluation comprises over 1500 datasets in total.

**TabPFN-OD** repurposes the original TabPFN (Hollmann et al., 2023) for outlier detection. TabPFN-OD predicts the outliers by the following: Given normal training data $\mathbf{x}_{\text{train}}$ in $d$-dimension and test data $\mathbf{x}_{\text{test}}$, we compute anomaly scores via a feature-wise self-prediction scheme. We randomly select a subset of features and, for each feature $j$, train a TabPFN regressor to predict $x_j$ from the remaining features $\mathbf{x}_{-j}$ using only normal training samples. The trained model is then applied to the test data, and the absolute prediction error is recorded. Final anomaly scores are obtained by averaging prediction errors across selected features. This approach relies on the assumption that normal samples exhibit strong inter-feature dependencies, while anomalies violate these relations and yield larger prediction errors. For large dataset, TabPFN-OD randomly selects up to 100 features and fix the context length equal to 5000. We include our repurposed TabPFN for anomaly scoring and evaluation in our code base.

*Table 12.* List of OD baselines comprising classical "shallow" methods, deep learning based models, and foundation models.

| Method | Abbrev. | Year | Type | Codes |
|---|---|---|---|---|
| k-th Nearest Neighbors(Ramaswamy et al., 2000) | kNN | 2000 | Shallow | PyOD (Zhao et al., 2019) |
| Local Outlier Factor (Breunig et al., 2000) | LOF | 2000 | Shallow | PyOD (Zhao et al., 2019) |
| Isolation Forest (Liu et al., 2008) | IForest | 2008 | Shallow | PyOD (Zhao et al., 2019) |
| Deep One-Class Classification(Ruff et al., 2018) | DeepSVDD | 2018 | Deep | PyOD (Zhao et al., 2019) |
| Classification Based AD using General Data (Bergman & Hoshen, 2020) | GOAD | 2020 | Deep | Paper |
| Internal Contrastive Learning (Shenkar & Wolf, 2022) | ICL | 2022 | Deep | Paper |
| Denoising diffusion models for out-of-distribution detection (Graham et al., 2023) | DDPM | 2023 | Deep | Paper |
| Diffusion Time Estimation (Non Parametric) (Livernoche et al., 2024) | DTE-NP | 2024 | Shallow | Paper |
| Diffusion Time Estimation (Categorical) (Livernoche et al., 2024) | DTE-C | 2024 | Deep | Paper |
| Zero-shot Tabular Outlier Detection (Shen et al., 2025) | FoMo-0D | 2025 | Foundation | Paper |
| Prior-data Fitted Network (Hollmann et al., 2023) | TabPFN-OD | 2022 | Foundation | Paper+Ours |

*Table 13.* Hyperparameter (HP) configurations for shallow and deep baselines. Foundations models are not included as they require zero training or tuning, given a new OD task. For shallow models, we **bold** the default setting as suggested in the paper or recent benchmarks. For deep models, we list the default HP.

| Model | Hyperparameter Configuration |
|---|---|
| kNN | $k \in [\mathbf{5}, 10, 20, 50, 100]$ |
| LOF | $N_{\text{neighbors}} \in [10, \mathbf{20}, 50, 100]$ |
| IForest | $N_{\text{estimators}} \in [50, \mathbf{100}, 200]$ 
 max_samples $\in [64, 128, \mathbf{256}]$ 
 max_features $\in [\mathbf{1.0}]$ |
| DTE-NP | $N_{\text{neighbors}} \in [\mathbf{5}, 10, 20, 50, 100]$ |
| DeepSVDD | Representation dim: 32 
 Hidden dim: 64, Activation: ReLU 
 Center $\epsilon = 0.1$ 
 Batch size: 128, LR: $10^{-3}$, Optimizer: Adam 
 Weight decay $w_d = 10^{-6}$, Epochs: 50 |
| GOAD | Embedding dim: 32 
 $m = 1$, $\lambda = 0.1$ (TC loss) 
 Transformations: 256, Feature maps: 8 
 Activation: LeakyReLU(0.2) 
 Batch size: 64, LR: $10^{-3}$, Optimizer: Adam |
| ICL | $k \in \{2, 10, d - 150\}$ (dimension-dependent) 
 Temperature $\tau = 0.1$ 
 F-network: 3 layers, hidden sizes [200, 400,200], BatchNorm 
 G-network: 3 layers, hidden sizes [50, 100,100], BatchNorm 
 Activation: LeakyReLU(0.2), LR: $10^{-3}$ 
 Early stopping when loss $< 10^{-3}$ or $10^{-2}$ 
 Optimizer: Adam |
| DDPM | Number of blocks: 3 
 Main dim: 128, Hidden dim: 256 
 Time embedding dim: 256 
 Dropout: $(0.4, 0.1)$ 
 Batch size: 64, LR: $10^{-4}$, Epochs: 400 
 Maximum timestep $T_{\text{max}} = 1000$, Reconstruction timestep $T_{\text{rec}} = 250$ 
 Optimizer: Adam |
| DTE-C | Hidden layers: [256, 512, 256] 
 Activation: ReLU 
 Dropout: 0.5 
 Batch size: 64, LR: $10^{-4}$, Epochs: 400 
 Maximum timestep $T_{\text{max}} = 400$, Number of bins: 7 
 Optimizer: Adam |

### E.3. OUTFORMER **Architecture, Hardware, Training and Inference**

**Architecture:** OUTFORMER utilizes a 10-layer Transformer structure, with 512 hidden dimensions `h_dim`, 8 attention heads, and a linear embedding layer mapping 100 dimensional input to `h_dim`, comprising 45.1M parameters in total. The output layer of the Transformer is a 2-layer MLP ($R^{\text{h\_dim}} \rightarrow R^2$) for inlier vs. outlier binary classification. For faster training, we initialize the self-attention weights to zero and freeze them, updating only the cross-attention parameters. This encourages cross-attention to capture the inference-time relationship between test outliers/inliers and the in-context inlier-only examples. Prior work (Shen et al., 2025) shows that disabling self-attention in this manner leads to improved performance while simultaneously speeding up training and inference.

**Training with varying dimensions and dataset size:** To handle inputs with a varying number of features $d$, when $d < D$, we rescale the input by a factor of $D/d$ and pad the feature dimension to size $D$ with zeros; when $d > D$, we randomly subsample $D$ features from the original $d$ dimensions.

**Hardware Setup:** For OUTFORMER and ablation models' training, we base our experiments on 2 different machines, first is one with 4 NVIDIA RTX A6000 GPUs with AMD EPYC 7742 64-Core Processors and second is one with 4 L40S GPUs, each with 48GB of GPU memory. For fairly measuring inference time (including training time for non-FM models) over various baselines, we only utilize 6 NVIDIA A6000 GPUs with AMD EPYC 7742 64-Core Processors to evaluate on SynBench, ADBench, OddBench and OvRBench.

**Curriculum Training:** For our SEC, we first use 5 categories for each prior {GMM, Copula-Dependence, Copula-Probability, SCM-Structure, SCM-Measurement } and also 5 equal-size bins across lower-to-high dimensions $\{[2, 21], [21, 40], [40, 59], [59, 78], [78, 101]\}$, consisting of 25 individual bins of different prior and dimension combinations in total. The point-wise pace function, we choose to use Root function in Table 11, with $\alpha = 0.8, \beta = 0.2$ (taken the default HP of two constants from (Wu et al., 2021)). We further impose an upper bound of 0.95 on the filtering ratio to ensure that a non-trivial amount of noise is consistently removed throughout training. For SEC, we set the temperature parameter (Line 7 in Algorithm 1) to $\tau = 0.5$, which strikes a balance between effective curriculum shaping and avoiding overly aggressive data reduction or forgetting across categories. The hyperparameters (HPs) used in curriculum learning are conducted over a small grid search within temperature $\tau \in \{0.3, 0.5, 1\}$ and pace function {Root, Linear, Log}. We select the optimal HPs using an in-distribution validation set from SynBench, consisting of 800 independent samples, uniformly drawn from five priors and spanning both low- and high-dimensional settings.

We trained our final model for 1,500 epochs of 1,000 synthetically generated dataset each, and set batch size per device equal to 2 datasets. Each dataset contains up to 100 dimensions, with up to 5,000 inliers as context points. That is, OUTFORMER is trained on 1,500,000 synthetically generated datasets. Training a 10-layer OUTFORMER on A6000 machine takes ∼6 days while training on L40S machine takes ∼3 days. We use cosine learning rate scheduler with a peak learning rate equals to $1e-4$ and warm up to 100 epochs. The optimizer is set to AdamW. The Fig. 10 shows the training curve for 4-layer OUTFORMER with different curriculum training strategies.

**Inference:** Without ensembling, OUTFORMER takes random subsampling up to 5,000 inliers from the training data as well as subsampling up to 100 dimensions to create a randomized context. For inference-time ensembling, OUTFORMER randomly subsample up to 1,000 inliers from the training data as well as subsample up to 100 dimensions to create a randomized context, and average outlier scores across 50 such random contexts run in parallel. Following Müller et al. (2022), we rescale input with $d<100$ by $\frac{100}{d}$ and pad to size 100 with 0s.

### E.4. Performance Metrics and Statistical Tests

#### E.4.1. RANKING BASED METRICS

For ranking and group based metrics, we adopt 5 metrics following the previous FM literature (Zhang et al., 2025a) and give a brief introduction of the metrics.

**Average Rank.** Let $\mathcal{M}$ denote the set of models and $\mathcal{D}$ the set of evaluation datasets. For a model $m \in \mathcal{M}$ and a dataset $\delta \in \mathcal{D}$, let $\text{rank}_\delta(m)$ denote the relative rank of $m$ on dataset $\delta$, where a smaller value indicates better performance (e.g., rank 1 is best) under a given evaluation metric (in OUTFORMER comparison scenario, we use AUROC). The average rank of model $m$ is defined as:

$$\text{AvgRank}(m) \;=\; \frac{1}{|\mathcal{D}|} \sum_{\delta \in \mathcal{D}} \text{rank}_\delta(m). \tag{3}$$

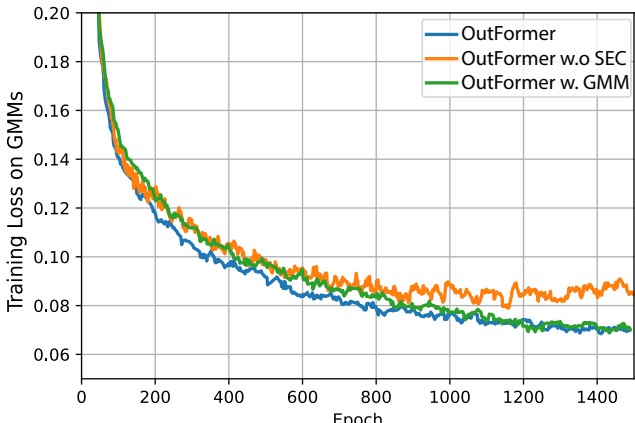

*Figure 10.* Training Losses: visualize the training losses of OUTFORMER, OUTFORMER w.o. SEC (just trained on Mixed Priors) and OUTFORMER trained only on GMM prior (OUTFORMER w. GMM). On 4-layer models, we see that SEC improves the training losses from OUTFORMER w.o. SEC, converging faster than GMM only.

**Elo Rating.** The Elo rating framework aggregates pairwise win–loss outcomes into a global competitive ranking (Elo, 1967). Each model is treated as a player, and its rating is iteratively updated based on pairwise performance comparisons against other models across datasets. For a given dataset, all participating models are compared in a round-robin fashion, where match outcomes are determined by their relative performance scores, with ties allowed within a tolerance. Rating updates follow the standard Elo formulation with an initial rating $R_0(=1000)$, a K-factor $K(=32)$ controlling rating volatility, and a tie threshold $\epsilon(=0.5)$. The final Elo score reflects a model's relative strength over all pairwise matchups and datasets.

**Winrate.** Winrate measures the fraction of pairwise comparisons in which a model outperforms competing models across datasets. Formally, for a model $m \in \mathcal{M}$, the winrate is defined as

$$\text{Winrate}(m) = \frac{1}{|\mathcal{D}|(|\mathcal{M}|-1)} \sum_{\delta \in \mathcal{D}} \sum_{\substack{m' \in \mathcal{M} \\ m' \neq m}} \left( \mathbb{I}[E_\delta(m) < E_\delta(m')] + \frac{1}{2} \mathbb{I}[E_\delta(m) = E_\delta(m')] \right), \tag{4}$$

where $E$ denotes the error metric, defined as $E = 1 - \text{AUROC}$ for anomaly detection. The indicator function $\mathbb{I}[\cdot]$ equals one if the condition holds and zero otherwise; ties contribute half a win. We define the metrics using AUROC, but they also apply to other performance metrics such as AUPRC.

**Rescaled AUC (rAUC).** To reduce the impact of various datasets' difficulties, we normalize model errors within each dataset relative to the best and worst-performing models. For dataset $\delta$, the rescaled AUC of model $m$ is defined as

$$\text{rAuc}_\delta(m) = 1 - \frac{E_\delta(m) - \min_m E_\delta(m)}{\max_m E_\delta(m) - \min_m E_\delta(m)}. \tag{5}$$

The overall Rescaled AUC is the average of $\text{RAcc}_\delta(m)$ over all datasets.

**Champion Delta.** Let $m^* = \arg\min_m E(m)$ denote the champion model with the lowest error. The Champion Delta of model $m$ is defined as

$$\text{C}\Delta(m) = \left( 1 - \frac{E(m^*)}{E(m)} \right) \times 100, \tag{6}$$

which quantifies the relative percentage performance gap between model $m$ and the best-performing model.

E.4.2. STATISTICAL TESTS

**Permutation Test.** Given two models evaluated on the same set of datasets, we compute the per-dataset performance differences $d_i = s_i^{(A)} - s_i^{(B)}$ and discard ties ($d_i = 0$). The test statistic is the sum of differences, $T = \sum_i d_i$. Under the

null hypothesis that both models perform equally well, flipping the signs of $d_i$ randomly generates a similar value of $T$. To test this, we define a distribution of randomly permuted statistics.

$$T^{(b)} = \sum_i \epsilon_i d_i, \quad \epsilon_i \in \{-1, +1\}.$$

The one-sided $p$-value of the permutation test is estimated as $p = \Pr\big(T^{(b)} \geq T\big)$, which measures the likelihood that model A outperforms model B by chance.

We prefer this permutation test over a sign test because it incorporates the magnitude of performance differences rather than only their direction, making it more informative and statistically powerful when effect sizes vary. While the Wilcoxon signed-rank test relies on rank transformation of the performance differences, partially discarding the scale information and implicitly assuming that the distribution of paired differences is symmetric around its median. However, for completeness of the evaluation, we also report the Wilcoxon rank test $p$-values, which provide similar conclusions, in the following section with additional results (in Table 15, Table 16, Table 17 and Table 18).

# F. Additional Experiment Results

### F.1. Mixed-prior Training Synthetic-Prior Results

We provide an extended result from Table 3 with different priors on SynBench.

*Table 14.* Extended results from Table 3. Colored numbers show that naïve mixed-prior training underperforms GMM-only training on GMM datasets but SEC helps to improve the GMM performance.

| Train/Test on | Copula-Depend. | Copula-Prob. | SCM-Struct. | SCM-Measure. | GMM |
|---|---|---|---|---|---|
| Mixed (w/SEC) | 0.980±0.029 | 0.951±0.036 | 0.983±0.058 | 0.972±0.046 | 0.930±0.046 |
| Mixed (w/o SEC) | 0.961±0.036 | 0.878±0.038 | 0.980±0.062 | 0.963±0.054 | 0.873±0.085 |
| GMM (w/o SEC) | 0.862±0.038 | 0.902±0.042 | 0.979±0.060 | 0.965±0.049 | 0.941±0.057 |

### F.2. Individual OD Benchmark Results

We provide extended results of main experiment section (Table 4). Table 15 provides the comprehensive results on ADBench. Table 16 is the comprehensive summary results on OddBench. Table 17 provides the summary results on OvRBench, and Table 18 is the summary statistics on synthetic priors.

*Table 15.* Performance summary and comparison against OUTFORMER on ADBench. **Best** and second-best results are highlighted. Win/Lose/Tie reports the fraction of datasets where OUTFORMER wins, loses, or ties against baseline. Perm./Wil. $p$-value is based on the permutation test/ one-sided Wilcoxon rank test between the baseline and OUTFORMER, where lower values indicate OUTFORMER's performance is significantly better. **On ADBench, OUTFORMER is out-performing other baselines.**

| Model | Avg. Rank (↓) | ELO (↑) | Winrate (↑) | rAUC (↑) | $C_\Delta$ (↓) | Win / Lose / Tie | Perm. p | Wil. p |
|---|---|---|---|---|---|---|---|---|
| DTE-NP | $5.12_{\pm 2.3}$ | 1043 | 0.61 | $0.939_{\pm 0.08}$ | 0.39 | **0.46** / 0.39 / 0.16 | 0.06 | 0.11 |
| kNN | $5.05_{\pm 2.6}$ | 1001 | 0.61 | $0.938_{\pm 0.09}$ | 0.36 | **0.49** / 0.32 / 0.19 | 0.06 | 0.07 |
| LOF | $6.04_{\pm 3.6}$ | 961 | 0.53 | $0.913_{\pm 0.11}$ | 0.43 | **0.56** / 0.26 / 0.18 | **0.00** | **0.00** |
| IForest | $8.46_{\pm 3.1}$ | 794 | 0.32 | $0.879_{\pm 0.12}$ | 0.52 | **0.75** / 0.18 / 0.07 | **0.00** | **0.00** |
| DTE-C | $6.00_{\pm 3.1}$ | 937 | 0.53 | $0.932_{\pm 0.08}$ | 0.42 | **0.63** / 0.23 / 0.14 | **0.00** | **0.00** |
| ICL | $5.63_{\pm 3.7}$ | 1005 | 0.57 | $0.925_{\pm 0.14}$ | 0.35 | **0.54** / 0.32 / 0.14 | **0.02** | **0.03** |
| DDPM | $7.12_{\pm 2.9}$ | 943 | 0.43 | $0.904_{\pm 0.09}$ | 0.48 | **0.72** / 0.12 / 0.16 | **0.00** | **0.00** |
| GOAD | $9.32_{\pm 2.9}$ | 981 | 0.23 | $0.805_{\pm 0.19}$ | 0.58 | **0.81** / 0.11 / 0.09 | **0.00** | **0.00** |
| DeepSVDD | $9.81_{\pm 2.9}$ | 788 | 0.20 | $0.796_{\pm 0.16}$ | 0.63 | **0.84** / 0.07 / 0.09 | **0.00** | **0.00** |
| TabPFN-OD | $\underline{4.74_{\pm 3.1}}$ | $\underline{1227}$ | $\underline{0.65}$ | $\underline{0.945_{\pm 0.07}}$ | $\underline{0.34}$ | **0.44** / 0.35 / 0.21 | 0.12 | 0.17 |
| FOMO-0D | $6.0_{\pm 3.4}$ | 1084 | 0.54 | $0.928_{\pm 0.08}$ | 0.41 | **0.54** / 0.26 / 0.19 | **0.01** | **0.00** |
| OUTFORMER (Ours) | **$4.02_{\pm 2.6}$** | **1235** | **0.71** | **$0.956_{\pm 0.06}$** | **0.32** | – | – | – |

*Table 16.* Overall performance comparison across models on OddBench. **Best** and Second-best results are highlighted. Win/Lose/Tie reports the fraction of datasets where OUTFORMER wins, loses, or ties against baseline. Perm./Wil. *p*-value is based on the permutation test/ one-sided Wilcoxon rank test between the baseline and OUTFORMER, where lower values indicate OUTFORMER's performance is significantly better. **On OddBench,** OUTFORMER **is on par to strongest baseline, DTE-NP.**

| Model | Avg. Rank (↓) | ELO (↑) | Winrate (↑) | rAUC (↑) | $C_\Delta$ (↓) | Win / Lose / Tie | Perm. p | Wil. p |
|---|---|---|---|---|---|---|---|---|
| DTE-NP | **4.92**$_{\pm 2.9}$ | 1083 | **0.62** | **0.896**$_{\pm 0.14}$ | 0.40 | **0.42** / 0.40 / 0.18 | 0.58 | 0.45 |
| kNN | 5.54$_{\pm 2.9}$ | 1094 | 0.55 | 0.884$_{\pm 0.14}$ | 0.42 | **0.48** / 0.37 / 0.15 | **0.03** | **0.01** |
| LOF | 6.31$_{\pm 3.4}$ | 997 | 0.49 | 0.854$_{\pm 0.15}$ | 0.49 | **0.54** / 0.32 / 0.14 | **0.00** | **0.00** |
| IForest | 6.53$_{\pm 3.6}$ | 948 | 0.50 | 0.871$_{\pm 0.13}$ | 0.49 | **0.55** / 0.38 / 0.07 | **0.00** | **0.00** |
| DTE-C | 6.02$_{\pm 3.2}$ | 1079 | 0.51 | 0.871$_{\pm 0.16}$ | 0.44 | **0.51** / 0.33 / 0.16 | **0.00** | **0.00** |
| ICL | 6.35$_{\pm 3.4}$ | 1060 | 0.48 | 0.852$_{\pm 0.16}$ | 0.44 | **0.53** / 0.32 / 0.15 | **0.00** | **0.00** |
| DDPM | 9.13$_{\pm 2.7}$ | 848 | 0.23 | 0.775$_{\pm 0.15}$ | 0.58 | **0.76** / 0.15 / 0.09 | **0.00** | **0.00** |
| GOAD | 8.71$_{\pm 3.7}$ | 735 | 0.27 | 0.724$_{\pm 0.25}$ | 0.56 | **0.70** / 0.20 / 0.10 | **0.00** | **0.00** |
| DeepSVDD | 5.97$_{\pm 3.2}$ | 1124 | 0.51 | 0.862$_{\pm 0.16}$ | 0.43 | **0.52** / 0.31 / 0.17 | **0.00** | **0.00** |
| TabPFN-OD | 5.38$_{\pm 3.5}$ | 897 | 0.57 | 0.892$_{\pm 0.14}$ | 0.38 | **0.44** / 0.38 / 0.18 | 0.30 | 0.35 |
| FOMO-0D | 6.24$_{\pm 3.9}$ | **1100** | 0.50 | 0.863$_{\pm 0.16}$ | 0.41 | **0.54** / 0.33 / 0.13 | **0.00** | **0.00** |
| OUTFORMER (Ours) | 4.97$_{\pm 3.4}$ | 1034 | 0.60 | 0.895$_{\pm 0.13}$ | **0.36** | – | – | – |

*Table 17.* Overall performance comparison across models on OvRBench. **Best** and Second-best results are highlighted. Win/Lose/Tie reports the fraction of datasets where OUTFORMER wins, loses, or ties against baseline. Perm./Wil. *p*-value is based on the permutation test/ one-sided Wilcoxon rank test between the baseline and OUTFORMER, where lower values indicate OUTFORMER's performance is significantly better. **On OvRBench,** OUTFORMER **is on par to strongest baselines, DTE-NP and kNN.**

| Model | Avg. Rank (↓) | ELO (↑) | Winrate (↑) | rAUC (↑) | $C_\Delta$ (↓) | Win / Lose / Tie | Perm. p | Wil. p |
|---|---|---|---|---|---|---|---|---|
| DTE-NP | **5.13**$_{\pm 2.9}$ | 1100 | **0.61** | **0.910**$_{\pm 0.12}$ | 0.31 | **0.44** / 0.42 / 0.14 | 0.75 | 0.81 |
| kNN | 5.32$_{\pm 3.0}$ | 1129 | 0.59 | **0.910**$_{\pm 0.12}$ | 0.29 | **0.42** / 0.45 / 0.13 | 0.73 | 0.86 |
| LOF | 6.06$_{\pm 3.4}$ | 864 | 0.53 | 0.878$_{\pm 0.14}$ | 0.36 | **0.51** / 0.38 / 0.10 | **0.00** | **0.00** |
| IForest | 6.45$_{\pm 3.6}$ | 917 | 0.50 | 0.895$_{\pm 0.11}$ | 0.37 | **0.53** / 0.40 / 0.07 | **0.00** | **0.00** |
| DTE-C | 5.86$_{\pm 3.1}$ | 1167 | 0.54 | 0.895$_{\pm 0.12}$ | 0.32 | **0.50** / 0.38 / 0.12 | **0.00** | **0.00** |
| ICL | 6.49$_{\pm 3.3}$ | 1000 | 0.49 | 0.884$_{\pm 0.11}$ | 0.35 | **0.55** / 0.34 / 0.11 | **0.00** | **0.00** |
| DDPM | 9.39$_{\pm 2.5}$ | 770 | 0.23 | 0.807$_{\pm 0.12}$ | 0.49 | **0.78** / 0.18 / 0.06 | **0.00** | **0.00** |
| GOAD | 8.69$_{\pm 3.6}$ | 696 | 0.29 | 0.773$_{\pm 0.21}$ | 0.47 | **0.71** / 0.21 / 0.08 | **0.00** | **0.00** |
| DeepSVDD | 6.43$_{\pm 3.1}$ | 994 | 0.49 | 0.880$_{\pm 0.13}$ | 0.36 | **0.56** / 0.32 / 0.12 | **0.00** | **0.00** |
| TabPFN-OD | 5.45$_{\pm 3.3}$ | 1141 | 0.58 | 0.907$_{\pm 0.12}$ | 0.29 | **0.45** / 0.42 / 0.13 | 0.33 | 0.32 |
| FOMO-0D | 6.67$_{\pm 3.8}$ | 1012 | 0.47 | 0.871$_{\pm 0.14}$ | 0.36 | **0.57** / 0.30 / 0.13 | **0.00** | **0.00** |
| OUTFORMER (Ours) | 5.21$_{\pm 3.3}$ | **1209** | 0.60 | 0.907$_{\pm 0.11}$ | **0.28** | – | – | – |

*Table 18.* Overall performance comparison across models on SynBench (in-distribution training for OUTFORMER). **Best** and Second-best results are highlighted. Win/Lose/Tie reports the fraction of datasets where OUTFORMER wins, loses, or ties against baseline. Perm./Wil. *p*-value is based on the permutation test/ one-sided Wilcoxon rank test between the baseline and OUTFORMER, where lower values indicate OUTFORMER's performance is significantly better. OUTFORMER **is outperforming all benchmarks due to pretrained on same family of in-distribution data.**

| Model | Avg. Rank (↓) | ELO (↑) | Winrate (↑) | rAUC (↑) | $C_\Delta$ (↓) | Win / Lose / Tie | Perm. p | Wil. p |
|---|---|---|---|---|---|---|---|---|
| DTE-NP | 4.89$_{\pm 2.6}$ | 1050 | 0.40 | 0.938$_{\pm 0.07}$ | 0.57 | **0.71** / 0.04 / 0.26 | **0.00** | **0.00** |
| kNN | 5.47$_{\pm 3.1}$ | 1042 | 0.31 | 0.934$_{\pm 0.06}$ | 0.58 | **0.71** / 0.04 / 0.25 | **0.00** | **0.00** |
| LOF | 8.40$_{\pm 1.5}$ | 841 | 0.11 | 0.835$_{\pm 0.06}$ | 0.69 | **0.86** / 0.04 / 0.10 | **0.00** | **0.00** |
| IForest | 9.40$_{\pm 3.3}$ | 762 | 0.10 | 0.840$_{\pm 0.13}$ | 0.77 | **0.86** / 0.02 / 0.12 | **0.00** | **0.00** |
| DTE-C | 2.95$_{\pm 1.9}$ | 1142 | 0.56 | 0.965$_{\pm 0.04}$ | 0.50 | **0.60** / 0.08 / 0.33 | **0.00** | **0.00** |
| ICL | 4.96$_{\pm 3.3}$ | 1060 | 0.37 | 0.924$_{\pm 0.09}$ | 0.58 | **0.68** / 0.05 / 0.27 | **0.00** | **0.00** |
| DDPM | 8.89$_{\pm 4.2}$ | 804 | 0.07 | 0.803$_{\pm 0.16}$ | 0.70 | **0.78** / 0.02 / 0.20 | **0.00** | **0.00** |
| GOAD | 7.98$_{\pm 4.2}$ | 801 | 0.14 | 0.837$_{\pm 0.15}$ | 0.66 | **0.78** / 0.02 / 0.20 | **0.00** | **0.00** |
| DeepSVDD | 4.18$_{\pm 3.1}$ | 1048 | 0.43 | 0.937$_{\pm 0.08}$ | 0.54 | **0.63** / 0.08 / 0.29 | **0.00** | **0.00** |
| TabPFN-OD | 2.79$_{\pm 3.0}$ | 1180 | 0.58 | 0.969$_{\pm 0.07}$ | **0.20** | **0.39** / 0.31 / 0.31 | **0.00** | **0.00** |
| FOMO-0D | 4.27$_{\pm 3.1}$ | 948 | 0.43 | 0.951$_{\pm 0.06}$ | 0.48 | **0.62** / 0.05 / 0.33 | **0.00** | **0.00** |
| OUTFORMER | **1.67**$_{\pm 1.5}$ | **1219** | **0.67** | **0.994**$_{\pm 0.01}$ | 0.23 | – | – | – |

## F.3. Pairwise Comparison Results

Figure 11 provides pairwise scatter plot and histogram distribution of performance across four dataset benchmarks (ADBench,OddBench,OvRBench, Synthetic Priors), for OUTFORMER and its strongest competitor, DTE-NP. We see that in general, OUTFORMER either outperforms or performs comparably to DTE-NP.

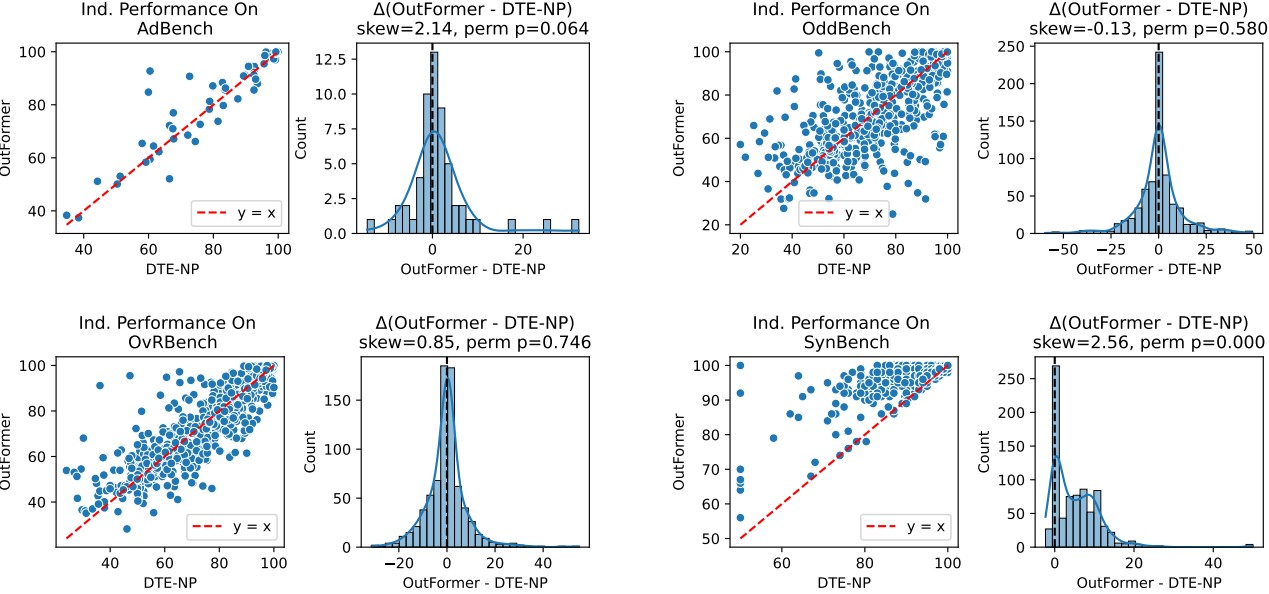

*Figure 11.* Left panel: Scatter plot of OUTFORMER vs DTE-NP performance across all datasets, w.r.t. **AUROC** as performance metric. Each point represents one dataset. The red dashed diagonal line (y=x) shows where equal performance would be; points above it mean OUTFORMER performs better, points below mean DTE-NP performs better. Right panel: Histogram showing the distribution of performance differences (OUTFORMER- DTE-NP). Positive values mean OUTFORMER wins, negative means DTE-NP wins. The vertical dashed line at 0 marks equal performance.

## F.4. Comparison Across Versions of OUTFORMER

We provide extended results of Table 5 in experiment section. Table 19 provides the comparison across OUTFORMER versions on ADBench. Table 20 provides the comparison across OUTFORMER versions on OddBench. Table 19 provides the comparison across OUTFORMER versions on OvRBench. Figure 12 provides the permutation test results across different versions of OUTFORMER, demonstrating that each component of OUTFORMER is essential for its success.

*Table 19.* **Comparison on ADBench for different verions of** OUTFORMER. **Best** result is highlighted.

| Model | Avg. Rank (↓) | ELO (↑) | Winrate (↑) | rAUC (↑) | $C_\Delta$ (↓) |
|---|---|---|---|---|---|
| FOMO-0D (L4, GMM) | $5.09_{\pm 2.4}$ | 913 | 0.41 | $0.935_{\pm 0.07}$ | 0.37 |
| FOMO-0D (L4, Mix.) | $5.93_{\pm 2.4}$ | 965 | 0.28 | $0.913_{\pm 0.09}$ | 0.42 |
| FOMO-0D (L4, Mix. w. SEC) | $4.67_{\pm 2.0}$ | 954 | 0.47 | $0.942_{\pm 0.08}$ | 0.35 |
| OUTFORMER | $3.23_{\pm 2.1}$ | **1095** | **0.63** | $\mathbf{0.971}_{\pm 0.05}$ | **0.22** |
| OUTFORMER w.o. Ens | $4.11_{\pm 2.2}$ | 928 | 0.47 | $0.939_{\pm 0.07}$ | 0.30 |
| OUTFORMER w.o. SEC | $\mathbf{3.21}_{\pm 1.8}$ | 1061 | **0.63** | $0.968_{\pm 0.05}$ | 0.24 |
| OUTFORMER w.o SEC&Ens | $4.79_{\pm 1.9}$ | 1046 | 0.45 | $0.941_{\pm 0.07}$ | 0.34 |
| OUTFORMER w.o Mix.&SEC | $3.86_{\pm 2.4}$ | 1038 | 0.58 | $0.958_{\pm 0.05}$ | 0.25 |

## F.5. Corresponding Results based on AUPRC

Instead of relying on $E = 1 - \text{AUROC}$ as the error metric and calculate win-rates, ranking, and Elo based on AUROC, we provide additional results on using AUPRC (Area Under the Precision–Recall Curve). Table 22 shows the performance

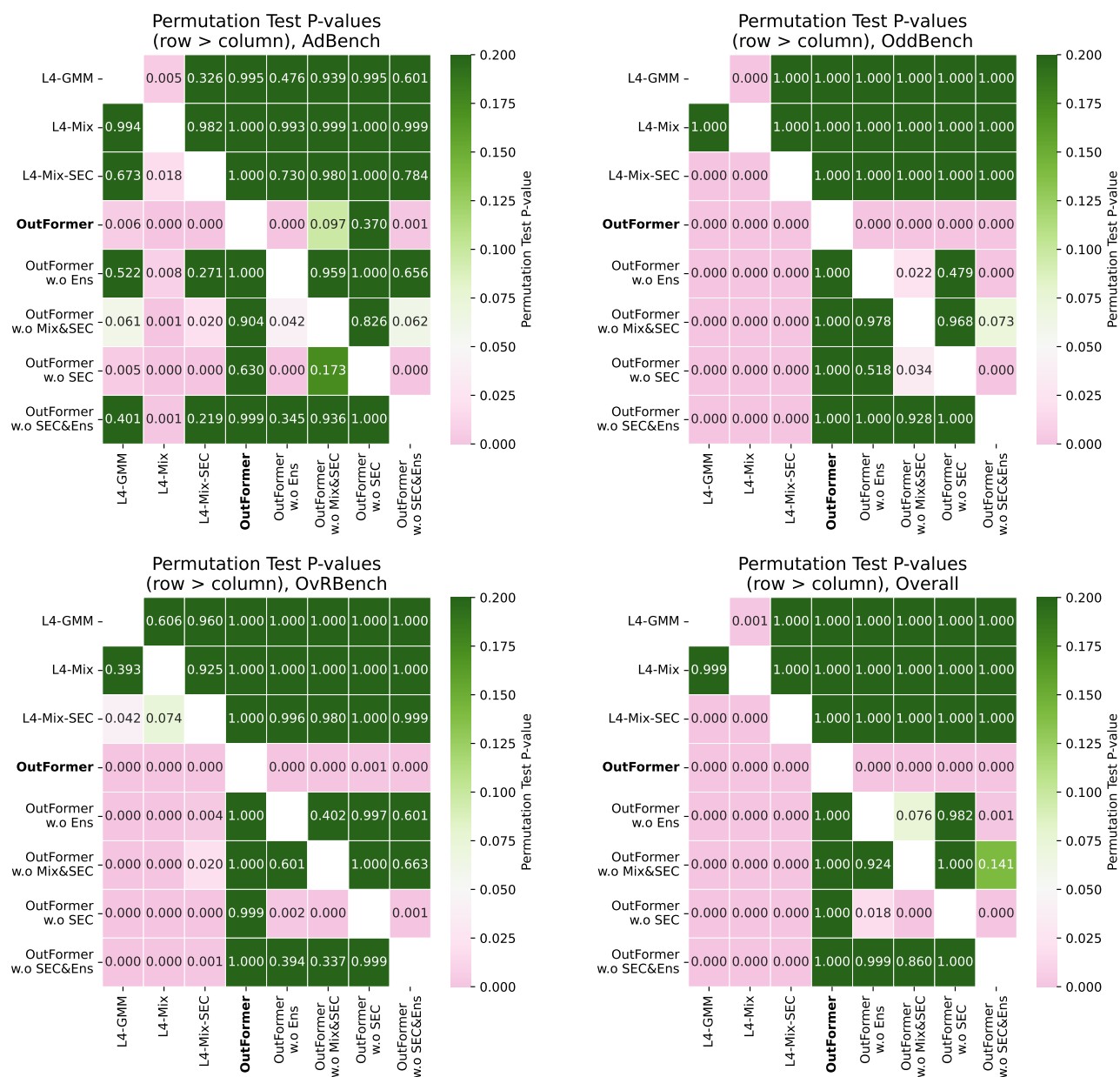

*Figure 12.* Heatmap of permutation test's signed-rank p-values comparing each model (row) against each other model (column). Lighter cells (lower $p$) indicate the row model significantly outperforms the column model; darker cells indicate little or no evidence of superiority.

*Table 20.* **Comparison on OddBench for different versions of** OUTFORMER**. Best** highlighted.

| Model | Avg. Rank ($\downarrow$) | ELO ($\uparrow$) | Winrate ($\uparrow$) | rAUC ($\uparrow$) | $C_\Delta$ ($\downarrow$) |
|---|---|---|---|---|---|
| FOMO-0D (L4, GMM) | $5.05_{\pm 2.5}$ | 950 | 0.38 | $0.849_{\pm 0.19}$ | 0.39 |
| FOMO-0D (L4, Mix.) | $5.51_{\pm 2.6}$ | 745 | 0.32 | $0.826_{\pm 0.20}$ | 0.42 |
| FOMO-0D (L4, Mix. w. SEC) | $4.81_{\pm 2.3}$ | 1035 | 0.41 | $0.873_{\pm 0.16}$ | 0.37 |
| OUTFORMER | $\mathbf{3.54_{\pm 2.1}}$ | **1124** | **0.59** | $\mathbf{0.932_{\pm 0.11}}$ | **0.25** |
| OUTFORMER w.o. Ens | $4.04_{\pm 2.1}$ | 1062 | 0.50 | $0.914_{\pm 0.13}$ | 0.30 |
| OUTFORMER w.o. SEC | $3.56_{\pm 2.2}$ | 953 | 0.56 | $0.913_{\pm 0.14}$ | 0.26 |
| OUTFORMER w.o SEC&Ens | $4.11_{\pm 2.2}$ | 919 | 0.49 | $0.896_{\pm 0.15}$ | 0.31 |
| OUTFORMER w.o Mix.&SEC | $3.97_{\pm 2.4}$ | 1112 | 0.53 | $0.905_{\pm 0.14}$ | 0.28 |

*Table 21.* **Comparison on OvRBench for different verions of** OUTFORMER**. Best** highlighted.

| Model | Avg. Rank ($\downarrow$) | ELO ($\uparrow$) | Winrate ($\uparrow$) | rAUC ($\uparrow$) | $C_\Delta$ ($\downarrow$) |
|---|---|---|---|---|---|
| FOMO-0D (L4, GMM) | $4.97_{\pm 2.3}$ | 933 | 0.42 | $0.898_{\pm 0.12}$ | 0.30 |
| FOMO-0D (L4, Mix.) | $5.20_{\pm 2.4}$ | 763 | 0.39 | $0.900_{\pm 0.11}$ | 0.31 |
| FOMO-0D (L4, Mix. w. SEC) | $4.76_{\pm 2.2}$ | 978 | 0.45 | $0.908_{\pm 0.12}$ | 0.29 |
| OUTFORMER | $\mathbf{3.58_{\pm 2.3}}$ | **1147** | **0.59** | $\mathbf{0.937_{\pm 0.09}}$ | **0.20** |
| OUTFORMER w.o. Ens | $4.31_{\pm 2.2}$ | 1021 | 0.49 | $0.918_{\pm 0.10}$ | 0.27 |
| OUTFORMER w.o. SEC | $3.80_{\pm 2.1}$ | 1095 | 0.56 | $0.928_{\pm 0.09}$ | 0.22 |
| OUTFORMER w.o SEC&Ens | $4.32_{\pm 2.0}$ | 1021 | 0.49 | $0.918_{\pm 0.09}$ | 0.27 |
| OUTFORMER w.o Mix.&SEC | $4.38_{\pm 2.5}$ | 1039 | 0.50 | $0.917_{\pm 0.11}$ | 0.24 |

across OUTFORMER and baselines evaluated on AUPRC, for ADBench datasets. Table 23 shows the performance across OUTFORMER and baselines evaluated on AUPRC, for OddBench datasets. Table 24 shows the performance across OUTFORMER and baselines evaluated on AUPRC, for OvRBench datasets. Finally, Table 25 shows the **overall** results on all 1500+ datasets combined across all these three benchmarks. Fig. 13 provides scatter plots and histogram of OUTFORMER's performances w.r.t. AUPRC compared to its strongest competitor, DTE-NP. The $p$-value and skewness of the distribution show OUTFORMER wins over DTE-NP by a large margin.

Comparing across multiple benchmarks, OUTFORMER shows superior performance in AUPRC performance, achieving the best across several ranking metrics. Notably, OUTFORMER outperforms all baselines significantly ($p$-val $< 0.05$), including DTE-NP and TabPFN-OD, on the largest two benchmarks OddBench and OvRBench.

*Table 22.* Overall performance comparison across models on ADBench with **AUPRC** as the comparison metric. **Best** and Second-best results are highlighted. Win/Lose/Tie reports the fraction of datasets where OUTFORMER wins, loses, or ties against baseline. Perm./Wil. $p$-value is based on the permutation test/ one-sided Wilcoxon rank test between the baseline and OUTFORMER, where lower values indicate OUTFORMER's performance is significantly better. **Evaluated w.r.t. AUPRC for ADBench,** OUTFORMER **achieves the best performance in three metrics and is comparable to DTE-NP and TabPFN-0D for statistical testing.**

| Model | Avg. Rank ($\downarrow$) | ELO ($\uparrow$) | Winrate ($\uparrow$) | rAUC ($\uparrow$) | $C_\Delta$ ($\downarrow$) | Win / Lose / Tie | Perm. p | Wil. p |
|---|---|---|---|---|---|---|---|---|
| DTE-NP | $4.75_{\pm 2.3}$ | **1209** | 0.65 | $0.854_{\pm 0.18}$ | 0.32 | **0.40** / 0.37 / 0.23 | 0.09 | 0.36 |
| kNN | $5.60_{\pm 2.7}$ | 1178 | 0.57 | $0.839_{\pm 0.19}$ | 0.31 | **0.44** / 0.37 / 0.19 | **0.04** | 0.14 |
| LOF | $6.09_{\pm 3.6}$ | 895 | 0.15 | $0.801_{\pm 0.21}$ | 0.38 | **0.63** / 0.21 / 0.16 | **0.00** | **0.00** |
| IForest | $8.89_{\pm 3.0}$ | 858 | 0.28 | $0.664_{\pm 0.27}$ | 0.45 | **0.74** / 0.16 / 0.11 | **0.00** | **0.00** |
| DTE-C | $6.39_{\pm 3.2}$ | 1021 | 0.50 | $0.789_{\pm 0.19}$ | 0.38 | **0.65** / 0.18 / 0.18 | **0.00** | **0.00** |
| ICL | $5.33_{\pm 3.8}$ | 1022 | 0.60 | $0.823_{\pm 0.22}$ | **0.27** | **0.51** / 0.40 / 0.09 | **0.05** | 0.11 |
| DDPM | $6.89_{\pm 3.2}$ | 878 | 0.45 | $0.784_{\pm 0.22}$ | 0.40 | **0.68** / 0.19 / 0.12 | **0.00** | **0.00** |
| GOAD | $9.25_{\pm 2.9}$ | 747 | 0.24 | $0.660_{\pm 0.29}$ | 0.44 | **0.74** / 0.12 / 0.14 | **0.00** | **0.00** |
| DeepSVDD | $9.07_{\pm 3.0}$ | 742 | 0.26 | $0.666_{\pm 0.26}$ | 0.49 | **0.77** / 0.09 / 0.14 | **0.00** | **0.00** |
| TabPFN-OD | $4.82_{\pm 3.2}$ | 1195 | 0.64 | $0.829_{\pm 0.21}$ | 0.28 | **0.42** / 0.40 / 0.18 | **0.02** | 0.21 |
| FOMO-0D | $5.86_{\pm 3.1}$ | 1109 | 0.56 | $0.832_{\pm 0.17}$ | 0.33 | **0.56** / 0.33 / 0.11 | 0.06 | **0.02** |
| OUTFORMER (**Ours**) | $\mathbf{4.46_{\pm 3.0}}$ | 1142 | **0.67** | $\mathbf{0.875_{\pm 0.17}}$ | 0.30 | – | – | – |

*Table 23.* Overall performance comparison across models on OddBench with **AUPRC** as the comparison metric. **Best** and Second-best results are highlighted. Win/Lose/Tie reports the fraction of datasets where OUTFORMER wins, loses, or ties against baseline. Perm./Wil. *p*-value is based on the permutation test/ one-sided Wilcoxon rank test between the baseline and OUTFORMER, where lower values indicate OUTFORMER's performance is significantly better. **Evaluated w.r.t. AUPRC for OddBench,** OUTFORMER **achieves the best performance.**

| Model | Avg. Rank ($\downarrow$) | ELO ($\uparrow$) | Winrate ($\uparrow$) | rAUC ($\uparrow$) | $C_\Delta$ ($\downarrow$) | Win / Lose / Tie | Perm. p | Wil. p |
|---|---|---|---|---|---|---|---|---|
| DTE-NP | $5.23_{\pm 2.8}$ | 1144 | 0.59 | $0.721_{\pm 0.27}$ | 0.29 | **0.50** / 0.33 / 0.18 | **0.00** | **0.00** |
| kNN | $5.98_{\pm 2.8}$ | 1040 | 0.51 | $0.694_{\pm 0.27}$ | 0.31 | **0.53** / 0.30 / 0.17 | **0.00** | **0.00** |
| LOF | $6.01_{\pm 3.5}$ | 970 | 0.52 | $0.689_{\pm 0.28}$ | 0.32 | **0.54** / 0.29 / 0.17 | **0.00** | **0.00** |
| IForest | $7.25_{\pm 3.8}$ | 989 | 0.43 | $0.599_{\pm 0.31}$ | 0.38 | **0.61** / 0.27 / 0.12 | **0.00** | **0.00** |
| DTE-C | $6.51_{\pm 3.2}$ | 924 | 0.47 | $0.667_{\pm 0.29}$ | 0.31 | **0.55** / 0.24 / 0.21 | **0.00** | **0.00** |
| ICL | $6.05_{\pm 3.3}$ | 1097 | 0.52 | $0.690_{\pm 0.28}$ | 0.28 | **0.50** / 0.31 / 0.19 | **0.00** | **0.00** |
| DDPM | $9.08_{\pm 2.8}$ | 792 | 0.24 | $0.535_{\pm 0.29}$ | 0.38 | **0.72** / 0.11 / 0.17 | **0.00** | **0.00** |
| GOAD | $8.65_{\pm 3.5}$ | 713 | 0.28 | $0.547_{\pm 0.32}$ | 0.36 | **0.68** / 0.17 / 0.15 | **0.00** | **0.00** |
| DeepSVDD | $5.78_{\pm 3.2}$ | **1224** | 0.53 | $0.692_{\pm 0.29}$ | 0.27 | **0.51** / 0.28 / 0.21 | **0.00** | **0.00** |
| TabPFN-OD | $5.35_{\pm 3.4}$ | 962 | 0.57 | $0.729_{\pm 0.29}$ | 0.25 | **0.45** / 0.36 / 0.19 | **0.01** | **0.00** |
| FOMO-0D | $5.66_{\pm 3.8}$ | 1158 | 0.55 | $0.732_{\pm 0.28}$ | 0.25 | **0.45** / 0.35 / 0.20 | **0.05** | **0.00** |
| OUTFORMER (**Ours**) | **$4.73_{\pm 3.3}$** | 987 | **0.63** | **$0.750_{\pm 0.27}$** | **0.22** | – | – | – |

*Table 24.* Overall performance comparison across models on OvRBench with **AUPRC** as the comparison metric. **Best** and Second-best results are highlighted. Win/Lose/Tie reports the fraction of datasets where OUTFORMER wins, loses, or ties against baseline. Perm./Wil. *p*-value is based on the permutation test/ one-sided Wilcoxon rank test between the baseline and OUTFORMER, where lower values indicate OUTFORMER's performance is significantly better. **Evaluated w.r.t. AUPRC on OvRBench,** OUTFORMER **is achieving the best performance.**

| Model | Avg. Rank ($\downarrow$) | ELO ($\uparrow$) | Winrate ($\uparrow$) | rAUC ($\uparrow$) | $C_\Delta$ ($\downarrow$) | Win / Lose / Tie | Perm. p | Wil. p |
|---|---|---|---|---|---|---|---|---|
| DTE-NP | $5.20_{\pm 2.8}$ | 1049 | 0.60 | $0.825_{\pm 0.19}$ | 0.24 | **0.46** / 0.40 / 0.14 | **0.02** | **0.04** |
| kNN | $5.75_{\pm 3.0}$ | 997 | 0.55 | $0.813_{\pm 0.19}$ | 0.25 | **0.48** / 0.39 / 0.13 | **0.00** | **0.00** |
| LOF | $6.02_{\pm 3.4}$ | 871 | 0.53 | $0.784_{\pm 0.21}$ | 0.28 | **0.54** / 0.33 / 0.13 | **0.00** | **0.00** |
| IForest | $7.04_{\pm 3.9}$ | 883 | 0.45 | $0.744_{\pm 0.23}$ | 0.33 | **0.59** / 0.34 / 0.07 | **0.00** | **0.00** |
| DTE-C | $6.32_{\pm 3.1}$ | 1024 | 0.50 | $0.786_{\pm 0.2}$ | 0.27 | **0.57** / 0.32 / 0.11 | **0.00** | **0.00** |
| ICL | $6.22_{\pm 3.2}$ | 985 | 0.51 | $0.789_{\pm 0.19}$ | 0.26 | **0.56** / 0.32 / 0.12 | **0.00** | **0.00** |
| DDPM | $9.31_{\pm 2.5}$ | 752 | 0.23 | $0.653_{\pm 0.22}$ | 0.38 | **0.78** / 0.14 / 0.09 | **0.00** | **0.00** |
| GOAD | $8.68_{\pm 3.5}$ | 752 | 0.29 | $0.661_{\pm 0.26}$ | 0.35 | **0.72** / 0.20 / 0.08 | **0.00** | **0.00** |
| DeepSVDD | $6.23_{\pm 3.0}$ | 1044 | 0.51 | $0.784_{\pm 0.2}$ | 0.27 | **0.58** / 0.30 / 0.12 | **0.00** | **0.00** |
| TabPFN-OD | $5.29_{\pm 3.4}$ | **1285** | 0.60 | $0.824_{\pm 0.2}$ | 0.23 | **0.46** / 0.36 / 0.12 | **0.02** | 0.08 |
| FOMO-0D | $6.19_{\pm 3.8}$ | 1136 | 0.52 | $0.791_{\pm 0.21}$ | 0.25 | **0.54** / 0.35 / 0.11 | **0.00** | **0.00** |
| OUTFORMER (**Ours**) | **$4.99_{\pm 3.2}$** | 1220 | **0.62** | **$0.830_{\pm 0.19}$** | **0.20** | – | – | – |

*Table 25.* Overall performance comparison across models on ALL 1500+ real-world datasets with **AUPRC** as the comparison metric. **Best** and Second-best results are highlighted. Win/Lose/Tie reports the fraction of datasets where OUTFORMER wins, loses, or ties against baseline. Perm./Wil. *p*-value is based on the permutation test/ one-sided Wilcoxon rank test between the baseline and OUTFORMER, where lower values indicate OUTFORMER's performance is significantly better. **Evaluated w.r.t. AUPRC on hundreds of datasets,** OUTFORMER **achieves the best performance, significantly outperforming all baselines.**

| Model | Avg. Rank ($\downarrow$) | ELO ($\uparrow$) | Winrate ($\uparrow$) | rAUC ($\uparrow$) | $C_\Delta$ ($\downarrow$) | Win / Lose / Tie | Perm. p | Wil. p |
|---|---|---|---|---|---|---|---|---|
| DTE-NP | $5.20_{\pm 2.8}$ | 1048 | 0.60 | $0.778_{\pm 0.23}$ | 0.27 | **0.47** / 0.36 / 0.16 | **0.00** | **0.00** |
| kNN | $5.85_{\pm 2.9}$ | 994 | 0.64 | $0.759_{\pm 0.24}$ | 0.28 | **0.50** / 0.35 / 0.15 | **0.00** | **0.00** |
| LOF | $6.02_{\pm 3.4}$ | 859 | 0.53 | $0.741_{\pm 0.25}$ | 0.30 | **0.55** / 0.31 / 0.14 | **0.00** | **0.00** |
| IForest | $7.20_{\pm 3.8}$ | 890 | 0.43 | $0.674_{\pm 0.28}$ | 0.36 | **0.60** / 0.31 / 0.09 | **0.00** | **0.00** |
| DTE-C | $6.41_{\pm 3.2}$ | 1017 | 0.49 | $0.732_{\pm 0.25}$ | 0.29 | **0.56** / 0.28 / 0.16 | **0.00** | **0.00** |
| ICL | $6.11_{\pm 3.3}$ | 976 | 0.52 | $0.745_{\pm 0.24}$ | 0.27 | **0.53** / 0.32 / 0.15 | **0.00** | **0.00** |
| DDPM | $9.11_{\pm 2.7}$ | 758 | 0.24 | $0.604_{\pm 0.27}$ | 0.38 | **0.75** / 0.12 / 0.13 | **0.00** | **0.00** |
| GOAD | $8.69_{\pm 3.5}$ | 759 | 0.28 | $0.609_{\pm 0.29}$ | 0.36 | **0.71** / 0.18 / 0.11 | **0.00** | **0.00** |
| DeepSVDD | $6.13_{\pm 3.2}$ | 1050 | 0.51 | $0.737_{\pm 0.25}$ | 0.28 | **0.55** / 0.29 / 0.16 | **0.00** | **0.00** |
| TabPFN-OD | $5.30_{\pm 3.4}$ | **1287** | 0.59 | $0.781_{\pm 0.25}$ | 0.24 | **0.46** / 0.39 / 0.15 | **0.00** | **0.00** |
| FOMO-0D | $5.93_{\pm 3.8}$ | 1144 | 0.54 | $0.766_{\pm 0.25}$ | 0.25 | **0.50** / 0.35 / 0.15 | **0.00** | **0.00** |
| OUTFORMER (**Ours**) | $4.85_{\pm 3.2}$ | 1217 | **0.63** | $0.795_{\pm 0.23}$ | **0.22** | – | – | – |

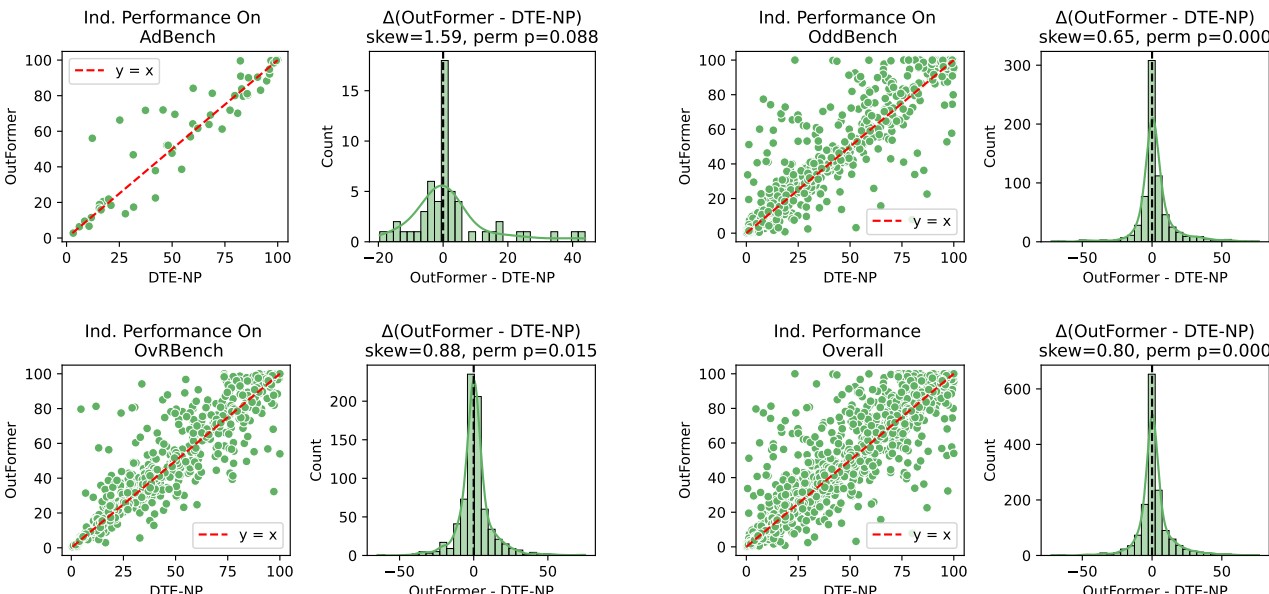

*Figure 13.* Left panel: Scatter plot of OUTFORMER vs DTE-NP performance across all datasets, w.r.t. **AUPRC** as performance metric. Each point represents one dataset. The red dashed diagonal line (y=x) shows where equal performance would be; points above it mean OUTFORMER performs better, points below mean DTE-NP performs better. Right panel: Histogram showing the distribution of performance differences (OUTFORMER- DTE-NP). Positive values mean OUTFORMER wins, negative means DTE-NP wins. The vertical dashed line at 0 marks equal performance.

## F.6. Performance Results on Individual **ADBench** Datasets

We provide individual performance results on each ADBench dataset for all methods compared in Table 37, Table 38 and Table 39 w.r.t AUPRC, AUROC and F1, respectively.

**Disclaimer:** Individual dataset performances on our two large-scale OD benchmarks are provided **online-only**; at `https://github.com/psorus/Outformer.git`, since each contains several hundreds of datasets and thus infeasible for print.

# G. Additional Ablation Results

## G.1. Alternative Curriculum Strategies

First, we describe the alternative curriculum strategies in detail:

**SPL:** Following (Jiang et al., 2015), we implement this curriculum learning strategy as the following: First, SPL considers only point-wise loss to sort all points across datasets in a batch. For SPL implementation, we use the same pace scheduler $g(\cdot)$ as the SEC (root scheduler, $\alpha = 0.8$, $\beta = 0.2$). Then, for all the points within a batch, SPL only propagates gradients from easier points with lower loss irrespective of their priors or dimensionality.

**AC:** Following (Li et al., 2025), we implement an anti-curriculum strategy as follows. At each epoch, we compute the gradient norm $\{\|\nabla\mathcal{L}_p(\theta)\|^2\}_{p=1}^P$ and the gradient variance $\{\sigma_p\}_{p=1}^P$ for each data prior. Then we update the data mixing weights of the each prior $Q_q \leftarrow Q_q \exp(\zeta_1\|\mathcal{L}_p(\theta)\|^2 - \frac{\zeta_2}{2b}\sigma_p^2)$. We keep $\zeta_1 = 1e-1, \zeta_2 = 1e-2, b = 1$, which are one of the possible HP settings as suggested in the paper. The basic idea behind anti-curriculum (AC) strategy is to favor data priors that have contributed to large gradients with smaller loss variances, we refer to the Section 3.2 of (Li et al., 2025) for detailed discussions.

**Manual1:** We provide a simplified variant of (Zhang et al., 2025a), but without constructing a matrix that jointly models performance, diversity, and distinctiveness across data priors. We partition categories and data priors in the same manner as OUTFORMER, using the same number of bins. Guided by the heuristic that (i) the GMM prior is both difficult to learn and particularly beneficial for ADBench, and (ii) high-dimensional data are generally harder to model, we assign a sampling probability of 0.3 to the {GMM prior, data dimension $[78, 101]$} bin, while allocating smaller probabilities to all remaining.

**Manual2:** We divide the priors and dimensionalities in the same manner as OUTFORMER, using the same number of bins. Guided by Table 3, we order the priors by their diagonal values from high to low (easy to hard), pair them with dimensions from low to high to define a partial order, and assign high sampling probabilities (0.3) in a round-robin manner as training progresses.

We train all the different curriculum learning strategies on a L4-OUTFORMER with Mixed priors, keeping the same categories and dimensional bins for the Manual2 Curriculum. We keep the same learning rate, optimizer and scheduler, trained on 1,500 epochs and report the final-epoch checkpoint's performance, as shown in Table 5. In addition, we provide the permutation test and SynBench results by separating the data priors. Figure 14 shows the pairwise permutation test between different curriculum learning strategies on ADBench. Table 26 shows each model's results on each synthetic priors. Overall, **SEC (Ours)** consistently outperforms other curriculum learning variants.

## G.2. Removing Data Priors

For the data-prior ablation, we train all models using an L10-OUTFORMER configuration with mixed priors, while keeping the dimensional bins, learning rate, optimizer, and scheduler fixed. All models are trained for 1,500 epochs, and we report performance from the final-epoch checkpoint. We employ L10 models with SEC throughout this study to mitigate performance degradation arising from conflicts among data priors. Results are shown in Table 7. We next analyze the contribution of individual priors by progressively removing each from pretraining, eventually stripping down to GMM-only pretraining. We begin by eliminating SCM priors, as Table 2 suggests their redundancy, showing that GMM-trained models generalize well to SCM-based data. As shown in Table 27, removing SCMs preserves in-distribution performance but degrades performance on real-world data (ADBench). Removing the Copula priors further degrades performance.

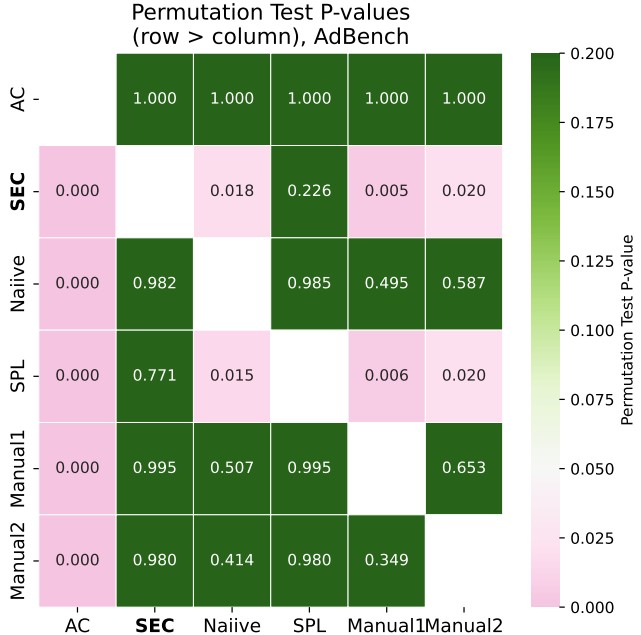

*Figure 14.* Heatmap of permutation test signed-rank *p*-values comparing each model (row) against each other model (column) for different CL strategies, on ADBench. Lighter cells (lower *p*) indicate the row model significantly outperforms the column model. Overall, **SEC (Ours)** consistently outperforms other CL variants.

*Table 26.* Average AUROC across different synthetic datasets (on SynBench) for different curriculum learning (CL) strategies (extended SynBench result from Table 5).

| Model | Copula-Depend. | Copula-Prob. | SCM-Struct. | SCM-Measure. | GMM |
|---|---|---|---|---|---|
| SEC(Ours) | $0.980_{\pm 0.029}$ | $0.951_{\pm 0.036}$ | $0.983_{\pm 0.058}$ | $0.972_{\pm 0.046}$ | $0.930_{\pm 0.046}$ |
| SPL | $0.978_{\pm 0.032}$ | $0.955_{\pm 0.035}$ | $0.982_{\pm 0.032}$ | $0.971_{\pm 0.046}$ | $0.918_{\pm 0.062}$ |
| Manual1 | $0.963_{\pm 0.029}$ | $0.928_{\pm 0.030}$ | $0.976_{\pm 0.030}$ | $0.956_{\pm 0.046}$ | $0.922_{\pm 0.070}$ |
| Manual2 | $0.972_{\pm 0.026}$ | $0.943_{\pm 0.026}$ | $0.968_{\pm 0.057}$ | $0.953_{\pm 0.026}$ | $0.918_{\pm 0.069}$ |
| AC | $0.894_{\pm 0.054}$ | $0.901_{\pm 0.026}$ | $0.895_{\pm 0.094}$ | $0.926_{\pm 0.062}$ | $0.881_{\pm 0.071}$ |
| Naïve | $0.961_{\pm 0.036}$ | $0.878_{\pm 0.038}$ | $0.980_{\pm 0.062}$ | $0.963_{\pm 0.054}$ | $0.873_{\pm 0.085}$ |

*Table 27.* **All priors contribute to** OUTFORMER**'s performance.** The first five metrics are evaluated on ADBench, while the last column reports overall avg. AUROC on SynBench.

| Model | ADBench | | | | | SynBench |
|---|---|---|---|---|---|---|
| | Avg. Rank (↓) | ELO (↑) | Winrate (↑) | rAUC (↑) | $C_\Delta$ (↓) | Avg. AUROC (↑) |
| OUTFORMER | $2.04_{\pm 1.3}$ | 1117 | 0.72 | $0.982_{\pm 0.04}$ | 0.15 | $0.973_{\pm 0.04}$ |
| – Ens. | $3.11_{\pm 1.2}$ | 1046 | 0.50 | $0.950_{\pm 0.04}$ | 0.31 | $0.971_{\pm 0.05}$ |
| – SCM-Struct. | $3.46_{\pm 1.5}$ | 961 | 0.50 | $0.947_{\pm 0.06}$ | 0.32 | $0.972_{\pm 0.04}$ |
| – SCM-Meas. | $3.68_{\pm 1.7}$ | 953 | 0.44 | $0.929_{\pm 0.09}$ | 0.32 | $0.970_{\pm 0.04}$ |
| – Copula-Prob. | $4.26_{\pm 1.5}$ | 964 | 0.37 | $0.921_{\pm 0.10}$ | 0.34 | $0.950_{\pm 0.07}$ |
| – Copula-Dep. | $3.96_{\pm 2.1}$ | 913 | 0.37 | $0.923_{\pm 0.11}$ | 0.36 | $0.936_{\pm 0.07}$ |

## G.3. Ablations on SEC Design Choices

### G.3.1. ROLE OF SAMPLING TEMPERATURE

We conduct additional ablation studies on the temperature parameter $\tau$ in our SEC framework. Specifically, we fix the SEC strategy and vary the softmax temperature $\tau$ using an L4-OUTFORMER configuration with mixed priors, while keeping the category definitions and dimensional bins unchanged. All models are trained for 1,500 epochs using the same learning rate, optimizer, and scheduler, and performance is reported using the final-epoch checkpoint. This study isolates the effect of temperature on balancing exploration and exploitation in SEC, as results are shown in Table 28. When the temperature $\tau$ is too low, the softmax distribution becomes overly sharp, leading to near-hard selection that suppresses exploration of other arms, especially leading to forgetting in later stages of training. Conversely, an excessively high $\tau$ over-smooths the distribution, reducing discrimination between arms, and producing worse generalization on ADBench and SynBench.

*Table 28.* **On ADBench, we see that temperature $\tau \in \{0.3, 0.5\}$ achieves the best performance.** The first five metrics are evaluated on ADBench, while the last column reports overall avg. AUROC on SynBench.

| Temperature | ADBench | | | | | SynBench |
| --- | --- | --- | --- | --- | --- | --- |
| | Avg. Rank ($\downarrow$) | ELO ($\uparrow$) | Winrate ($\uparrow$) | rAUC ($\uparrow$) | $C_\Delta$ ($\downarrow$) | Avg. AUROC ($\uparrow$) |
| $\tau = 0.1$ | $3.40_{\pm 1.4}$ | 906 | 0.35 | $0.966_{\pm 0.05}$ | 0.19 | $0.962_{\pm 0.05}$ |
| $\tau = 0.3$ | $2.51_{\pm 1.4}$ | 1032 | 0.57 | $0.976_{\pm 0.05}$ | 0.12 | $0.965_{\pm 0.06}$ |
| $\tau = 0.5$ | $2.53_{\pm 1.4}$ | 1134 | 0.58 | $0.982_{\pm 0.03}$ | 0.11 | $0.968_{\pm 0.05}$ |
| $\tau = 1.0$ | $3.00_{\pm 1.4}$ | 1008 | 0.47 | $0.972_{\pm 0.04}$ | 0.18 | $0.961_{\pm 0.07}$ |
| $\tau = 5.0$ | $3.14_{\pm 1.4}$ | 919 | 0.43 | $0.965_{\pm 0.06}$ | 0.19 | $0.950_{\pm 0.05}$ |

### G.3.2. REWARD BASED ON BINARY LOSS VS. CONTINUOUS (CROSS-ENTROPY) LOSS

Finally, we perform an ablation study by replacing the continuous loss–based reward with a binary reward (see Eq. 2). We fix the SEC strategy with $\tau = 0.5$ and use a root pacing function, training an L4-OUTFORMER model with mixed priors while keeping the category definitions and dimensional bins unchanged. All models are trained for 1,500 epochs using the same learning rate, optimizer, and scheduler, and results are reported from the final-epoch checkpoint. Table 29 shows the comparison of two reward functions with performance on ADBench. Note that we do not provide ELO and Avg. Rank, as we compare only two models. Our ablation shows that training a continuous-loss based reward wins over binary, while both Permutation test and Wilcoxon rank test show $p < 0.05$ (statistically significant).

*Table 29.* **Reward based on CE loss shows better performance than Reward based on Binary Loss** on ADBench. The Wilcoxon/ Permutation p-val are below $p < 0.05$, showing statistical significance.

| Reward w.r.t. | Eval. Metric: AUROC | | | | | Eval. Metric: AUPRC | | | | |
| --- | --- | --- | --- | --- | --- | --- | --- | --- | --- | --- |
| | Win Rate ($\downarrow$) | rAUC ($\uparrow$) | $C_\Delta$ ($\downarrow$) | Perm. p | Wil. p | Win Rate ($\downarrow$) | rAUC ($\uparrow$) | $C_\Delta$ ($\downarrow$) | Perm. p | Wil. p |
| Binary Loss | 0.42 | $0.973_{\pm 0.04}$ | 0.14 | **0.008** | **0.016** | 0.37 | $0.938_{\pm 0.03}$ | 0.11 | **0.001** | **0.001** |
| Continuous (CE) Loss | 0.58 | $0.986_{\pm 0.03}$ | 0.06 | - | - | 0.63 | $0.989_{\pm 0.03}$ | 0.05 | - | - |

### G.3.3. ROLE OF NUMBER OF BINS IN SEC

We conduct an additional ablation by varying the number of bins per prior, and compare against a model without SEC (No Bins). All models are trained with a smaller OUTFORMER (4 layers), keeping other hyperparameters fixed while only changing the number of bins. All models are trained on 4GPUs for 1,500 epochs until convergence.

Table 30 shows the ADBench performances of OUTFORMER trained on different bin sizes, with five major metrics. Increasing the number of bins ($1 \to 5$) improves generalization performance by enabling finer-grained curriculum control and better separation of task difficulty. The model trained with 10 bins also shows competitive performance. However, with a limited number of datasets per epoch (only 1,000 datasets per epoch), large bin counts (e.g., $20 \times 5$ priors equals 100 in total) result in only 10 samples per task category for estimating and updating the rewards. This leads to noisy reward estimates and insufficient learning signals, degrading stability and generalization. Consequently, performance drops at larger bin sizes (20˘30), only slightly better than the no-SEC baseline. We note that this limitation can be mitigated by increasing the number of samples per epoch, though at the cost of longer pretraining time.

*Table 30.* Ablation on number of bins. **Best** and second-best are highlighted. Bins = 5 is where OUTFORMER's performance peaks.

| Setting | Avg. Rank ($\downarrow$) | ELO ($\uparrow$) | Win Rate ($\uparrow$) | rAUC ($\uparrow$) | $C_\Delta$ ($\downarrow$) |
|---------|---------|---------|---------|---------|---------|
| No Bins | $4.14_{\pm 1.9}$ | 939 | 0.36 | $0.948_{\pm 0.07}$ | 0.27 |
| Bins = 1 ($\times 5$ priors) | $\underline{2.98}_{\pm 1.6}$ | 995 | 0.47 | $0.960_{\pm 0.05}$ | 0.19 |
| Bins = 5 ($\times 5$ priors) | $\mathbf{2.65}_{\pm \mathbf{1.7}}$ | 990 | **0.65** | $\mathbf{0.972}_{\pm \mathbf{0.05}}$ | **0.13** |
| Bins = 10 ($\times 5$ priors) | $3.03_{\pm 1.5}$ | **1048** | 0.58 | $0.969_{\pm 0.06}$ | 0.20 |
| Bins = 20 ($\times 5$ priors) | $3.75_{\pm 1.7}$ | 957 | 0.42 | $0.950_{\pm 0.06}$ | 0.28 |
| Bins = 30 ($\times 5$ priors) | $3.60_{\pm 1.7}$ | 972 | 0.45 | $0.954_{\pm 0.08}$ | 0.24 |

## G.4. Ablations on Ensemble Components and Time Analysis

We conduct an additional experiment on ADBench to quantify the trade-off between inference latency and ensemble size. Specifically, we partition datasets by an effective size measure:

$$S = \frac{\#\text{context points} \times \#\text{features}}{5{,}000 \times 100},$$

where 5,000 and 100 denote the maximum number of inlier context points and feature dimensions used in each forward pass, respectively. Datasets with fewer than 5,000 clean inlier context points and fewer than 100 features do not benefit from ensembling, and are categorized as *No Ens. Required*. The remaining datasets are grouped as *Small* ($1 < S < 5$), *Medium* ($5 < S < 30$), and *Large* ($S > 30$). The average inference time across different ensemble sizes is shown in Table 31.

*Table 31.* Average inference time, in seconds, across different ensemble sizes.

| Dataset Size | No Ens. | Ens. (5) | Ens. (10) | Ens. (20) | Ens. (50) | Ens. (100) |
|---------|---------|---------|---------|---------|---------|---------|
| No Ens. Required | 0.09 | 0.09 | 0.09 | 0.09 | 0.09 | 0.09 |
| Small | 0.44 | 0.88 | 0.90 | 1.67 | 3.23 | 6.42 |
| Medium | 1.09 | 3.26 | 4.03 | 6.08 | 12.23 | 20.89 |
| Large | 1.23 | 3.92 | 4.43 | 7.35 | 12.87 | 20.50 |
| **Total Avg.** | **0.84** | **1.95** | **2.61** | **3.86** | **5.27** | **12.20** |

Latency scales sub-linearly with the number of ensemble components because the framework is highly parallelizable: (i) ensemble components can be evaluated in batches, and (ii) evaluation can be distributed across multiple GPUs. Since OUTFORMER requires only a single forward pass per context and does not require per-dataset training, it is naturally suitable for ensembling while maintaining modest and controllable inference cost.

In addition, we report the mean rescaled AUROC (rAUC) under different ensemble sizes in Table 32.

*Table 32.* Mean rAUC across different ensemble sizes.

| Dataset Size | No Ens. | Ens. (5) | Ens. (10) | Ens. (20) | Ens. (50) | Ens. (100) |
|---------|---------|---------|---------|---------|---------|---------|
| Small | 0.963 | 0.965 | 0.965 | 0.964 | 0.965 | 0.964 |
| Medium | 0.958 | 0.974 | 0.978 | 0.977 | 0.983 | 0.978 |
| Large | 0.924 | 0.973 | 0.984 | 0.986 | 0.987 | 0.990 |

Ensembling improves performance by averaging over diverse context samplings. The gains saturate on small datasets, where a single context already captures most of the relevant information, but continue to scale on medium and large datasets due to increased context diversity and improved estimation of the decision boundary. Overall, ensembling provides a favorable performance–latency trade-off. Even without parallelization, Figure 8 shows that OUTFORMER remains faster than most competing methods, including deep OD models and TabPFN-OD.

*Table 33.* Categorical Testbed Data statistics.

| Dataset | # Samples | # Features |
|---|---|---|
| Customer Satisfaction | 20K | 88 |
| Mfeat_pixel | 128 | 1648 |
| MIC | 890 | 247 |
| PhishingWebsites | 3941 | 68 |
| Student Dropout | 1413 | 366 |
| Android Permissions | 9408 | 172 |
| IEEE-CIS-Fraud | 455K | 186 |

## H. Results on Categorical Data

While OUTFORMER is not designed to handle high-cardinality categorical data due to (1) lack of categorical-data based priors and (2) limit of feature sizes in model training, we provide a preliminary study on categorical data to show OUTFORMER's generalization in such settings.

**Dataset Selection**   Most of public anomaly detection benchmarks contain primarily numerical features, and even when categorical features are present, they are often low-cardinality or do not meaningfully drive the anomalous behavior. One relevant exception is IEEE-CIS-Fraud (Howard et al., 2019), which is derived from Vesta's real-world e-commerce fraud detection setting and contains mixed numerical and categorical transaction/user/device features. Following the commonly used basic preprocessing solution, we select features using exploratory data analysis, apply mean imputation, and obtain 186 mixed features, consisting of one-hot encoded categorical variables and numerical variables. We then use the inlier-only training subset as the context for OutFormer and evaluate on the remaining data.

We also construct additional categorical-heavy outlier detection tasks from TALENT (Liu et al., 2025b). Specifically, we select classification datasets with more than 15 categorical features, retain only the categorical variables, and transform each classification problem into an outlier detection task by treating the majority class as inliers and downsampling the remaining classes to form outliers, with an outlier ratio of approximately 15–20%. We retain the datasets where IForest achieves > 0.650 AUROC. Table 33 provides an overview of the number of test samples as well as the number of features.

**Experiment Setup**   OUTFORMER can be applied to categorical anomaly detection in two practical ways. First, OUT-FORMER treats one-hot encoded categorical variables as numerical inputs. Since OUTFORMER is pretrained with a maximum feature dimension of 100, we subsample up to 100 one-hot/numerical dimensions when the feature dimension exceeds this limit. This approach preserves the original sparse categorical indicators and therefore retains category-level information, including rare values and unusual co-occurrence patterns. Second, feature projection applies the XStream projection strategy described in Sec. 3.2 of (Manzoor et al., 2018) to map the original categorical representation into a fixed set of 100 numerical features. This provides a compact continuous representation and avoids direct subsampling when the one-hot feature space is large.

**Results**   Table 34 shows comparison of OUTFORMER to a strong baseline method that can handle categorical features, IForest. The results suggest that OUTFORMER can already transfer reasonably well to categorical-heavy settings, even though its current synthetic priors were not explicitly designed for categorical variables. In particular, OUTFORMER without projection outperforms IForest on **5 out of 7** datasets, showing that the learned priors capture useful distributional and dependency structures beyond purely numerical settings. We also observe that projection-based features are less effective in this setting. A likely reason is that projection compresses or transforms sparse/high-cardinality categorical representations into a lower-dimensional continuous space, which may discard rare but informative category-level signals. Overall, these findings indicate that OutFormer's existing priors have nontrivial generalization ability to mixed and categorical datasets.

## I. Additional Results on FOMO-0D vs OUTFORMER

To showcase cases where OUTFORMER benefits from additional priors beyond FOMO-0D's GMM prior, we perform a goodness-of-fit test (Huber-Carol et al., 2012) on ADBench datasets. For each dataset $\mathcal{D}$, we fit a Gaussian mixture model (GMM) with up to five components, select the best GMM according to the Bayesian information criterion (BIC), sample

*Table 34.* AUROC comparison between IForest and versions of OUTFORMER across categorical datasets. **Bold** highlights the best result.

| Dataset | IForest (default HP) | OUTFORMER (w/o proj.) | OUTFORMER (w/ proj.) |
|---|---|---|---|
| Customer Satisfaction | 0.873 | **0.912** | 0.755 |
| Mfeat_pixel | 0.901 | **0.944** | 0.894 |
| MIC | **0.789** | **0.789** | 0.765 |
| PhishingWebsites | 0.701 | **0.887** | 0.770 |
| Student Dropout | **0.685** | 0.650 | 0.660 |
| Android Permissions | 0.693 | **0.793** | 0.751 |
| IEEE-CIS-Fraud | **0.741** | 0.738 | 0.739 |

synthetic data $\tilde{\mathcal{D}}$ from the fitted GMM, and conduct a two-sample test between $\mathcal{D}$ and $\tilde{\mathcal{D}}$. A small $p$-value ($p \leq 0.05$) indicates that the GMM prior fails to capture the data distribution. When GMM is a poor fit ($p \leq 0.05$), OUTFORMER shows substantial gains over FOMO-0D in terms of average rank (5.13 vs. 6.93). Table 35 shows representative cases where OUTFORMER greatly improves over FOMO-0D, demonstrating the benefit of heterogeneous priors over a single GMM prior. In addition, Table 36 shows how scaling up models improves over FOMO-0D on large and high-dimensional datasets. While improvements arise from multiple factors, including heterogeneous priors, training dynamics, and ensembling, the results consistently support the effectiveness of expanding the prior space and model sizes.

*Table 35.* Representative cases of ADBench with Poor GMM goodness-of-fit. Avg. rank are shown in parenthesis. OUTFORMER's avg. rank is lower than FOMO-0D.

| Dataset | FOMO-0D | OUTFORMER |
|---|---|---|
| Skin ($p = 0.001$) | 0.706 (9) | 0.970 (3) |
| Fraud ($p = 0.0002$) | 0.939 (6) | 0.955 (2) |
| Satimage-2 ($p = 0.005$) | 0.984 (10) | 0.998 (1) |
| Mammography ($p = 0.003$) | 0.691 (12) | 0.822 (7) |
| Pageblocks ($p = 0.006$) | 0.866 (10) | 0.910 (3) |
| Campaign ($p = 0.002$) | 0.653 (10) | 0.784 (4) |
| Census ($p = 0.009$) | 0.605 (9) | 0.686 (6) |
| Backdoor ($p = 0.015$) | 0.786 (10) | 0.922 (5) |

*Table 36.* Representative cases of ADBench with large-scale / high-dimensional datasets. Avg. Rank in parenthesis. OUTFORMER performs better than FOMO-0D in such setting.

| Dataset | FOMO-0D | OUTFORMER |
|---|---|---|
| InternetAds ($N = 1,966$, $D = 1555$) | 0.556 (11) | 0.770 (2) |
| CIFAR10 ($N = 5,263$, $D = 512$) | 0.645 (9) | 0.710 (1) |
| MNIST-C ($N = 10,000$, $D = 512$) | 0.773 (10) | 0.857 (4) |
| 20News ($N = 11,905$, $D = 768$) | 0.569 (8) | 0.654 (2) |
| AGNews ($N = 10,000$, $D = 768$) | 0.610 (8) | 0.722 (3) |

*Table 37.* Individual dataset performances on ADBench w.r.t. AUPRC (± standard dev. over five seeds). **Best** and Second-best results are highlighted.

| Dataset | kNN | DTE-NP | LOF | IForest | DTE-C | ICL | DDPM | GOAD | DeepSVDD | Tabpfn-OD | FoMo-OD | OUTFORMER |
|---|---|---|---|---|---|---|---|---|---|---|---|---|
| aloi | 6.02±0.0 | 6.05±0.06 | 6.54±0.0 | 5.82±0.1 | 5.76±0.04 | 5.5±0.13 | 5.97±0.06 | 5.7±0.24 | 6.23±0.35 | 5.89±0.02 | 6.93±0.05 | 6.35±0.02 |
| amazon | 11.69±0.0 | 11.73±0.0 | 11.04±0.0 | 11.07±0.19 | 11.15±0.65 | 10.19±0.08 | 10.76±0.02 | 10.94±0.21 | 10.24±1.27 | 12.06±0.13 | 11.10±1.46 | 11.48±0.11 |
| annthyroid | 68.07±0.0 | 68.15±0.38 | 53.53±0.0 | 59.02±5.39 | 82.88±0.59 | 45.83±2.16 | 62.88±2.59 | 58.74±5.0 | 27.83±5.93 | 68.43±0.08 | 48.81±0.00 | 69.12±0.34 |
| backdoor | 46.54±1.41 | 45.7±12.5 | 53.47±2.6 | 9.37±1.48 | 62.44±2.39 | 89.18±1.04 | 14.2±0.63 | 6.31±1.94 | 84.77±2.79 | 20.84±3.98 | 43.86±14.97 | 71.97±1.32 |
| breastw | 98.92±0.32 | 99.19±0.14 | 80.01±10.09 | 99.49±0.12 | 88.25±1.06 | 96.79±1.61 | 98.6±0.51 | 98.77±0.37 | 96.01±1.28 | 99.46±0.20 | 99.03±0.34 | 99.23±0.23 |
| campaign | 49.04±0.0 | 49.95±0.67 | 40.24±0.0 | 45.73±1.85 | 46.9±0.71 | 48.9±0.98 | 48.87±0.29 | 23.09±7.18 | 36.95±12.7 | 43.23±0.90 | 34.24±0.47 | 47.67±0.17 |
| cardio | 77.22±0.0 | 77.41±0.85 | 70.15±0.0 | 78.63±2.69 | 69.29±1.27 | 47.91±11.47 | 69.28±0.91 | 84.79±0.57 | 38.89±5.52 | 55.14±1.88 | 78.92±0.00 | 71.85±0.00 |
| cardiotocography | 57.43±0.0 | 58.68±1.14 | 57.32±0.0 | 62.85±3.42 | 53.34±1.26 | 48.66±3.84 | 51.31±1.98 | 67.52±0.82 | 45.78±5.1 | 47.04±0.33 | 63.55±0.00 | 56.89±0.00 |
| celeba | 11.92±0.5 | 10.65±0.49 | 3.61±0.15 | 11.7±1.35 | 14.19±2.31 | 9.74±0.68 | 18.03±2.6 | 4.01±1.21 | 7.09±4.32 | 13.80±0.73 | 8.39±0.38 | 6.64±0.26 |
| census | 21.68±0.63 | 21.05±0.67 | 13.71±0.42 | 14.19±0.73 | 17.94±1.09 | 21.2±0.61 | 19.67±0.6 | 8.69±0.99 | 15.35±1.07 | 17.52±2.53 | 16.01±2.97 | 18.40±1.21 |
| cover | 55.79±3.74 | 59.97±10.57 | 82.92±2.19 | 8.66±1.53 | 63.73±12.33 | 34.48±16.39 | 73.27±3.36 | 1.09±0.18 | 2.69±1.53 | 2.69±0.64 | 72.57±8.38 | 84.15±3.00 |
| donors | 89.09±0.94 | 85.55±4.56 | 63.39±1.89 | 40.51±3.59 | 71.33±3.89 | 98.35±0.87 | 26.66±2.91 | 9.0±1.93 | 42.75±27.48 | 55.82±5.08 | 98.43±0.48 | 81.12±0.95 |
| fault | 61.98±0.0 | 62.17±0.14 | 50.44±0.0 | 59.19±2.02 | 63.93±0.72 | 63.18±0.7 | 64.75±0.69 | 62.14±0.82 | 55.46±1.48 | 62.93±0.36 | 63.08±0.00 | 61.64±0.00 |
| fraud | 38.68±7.19 | 42.1±7.63 | 55.09±8.19 | 18.22±3.66 | 62.14±10.92 | 53.88±8.78 | 69.21±3.46 | 29.44±24.66 | 48.33±17.13 | 44.63±7.73 | 55.13±13.77 | 22.52±2.33 |
| glass | 42.32±8.47 | 37.38±15.51 | 38.12±9.91 | 21.37±3.58 | 41.51±5.91 | 92.35±8.32 | 31.21±11.3 | 18.33±7.23 | 52.35±21.04 | 84.13±7.57 | 83.60±4.63 | 71.76±11.13 |
| hepatitis | 90.31±4.36 | 82.32±0.26 | 43.67±10.79 | 55.36±6.09 | 95.82±3.85 | 99.83±0.38 | 95.14±1.34 | 65.77±5.43 | 98.73±1.05 | 99.88±0.23 | 99.45±1.10 | 99.57±0.86 |
| http | 100.0±0.0 | 97.1±4.04 | 97.12±3.92 | 53.43±12.08 | 55.45±5.61 | 70.82±42.06 | 99.96±0.06 | 68.38±8.01 | 36.09±31.85 | 80.48±7.38 | 90.58±5.01 | 98.93±0.81 |
| imdb | 8.92±0.0 | 8.97±0.0 | 9.03±0.0 | 8.97±0.17 | 8.9±0.52 | 10.24±0.13 | 8.7±0.02 | 8.8±0.11 | 9.68±1.47 | 9.25±0.32 | 10.53±2.36 | 9.03±0.17 |
| internetads | 49.22±0.0 | 51.3±2.02 | 50.43±0.0 | 29.2±1.74 | 55.22±3.65 | 60.03±1.39 | 47.7±0.16 | 47.43±0.86 | 51.56±4.75 | 45.90±3.80 | 38.29±5.22 | 69.52±1.70 |
| ionosphere | 97.95±0.69 | 98.22±1.02 | 94.58±1.57 | 91.7±1.89 | 96.83±0.38 | 99.06±0.32 | 96.43±0.5 | 93.17±2.64 | 98.09±0.81 | 99.15±0.22 | 97.75±0.76 | 98.46±0.43 |
| landsat | 54.85±0.0 | 54.52±4.05 | 61.37±0.0 | 47.31±3.5 | 36.75±1.23 | 53.12±1.57 | 34.83±0.91 | 31.21±0.8 | 49.43±2.4 | 42.18±0.13 | 58.24±0.00 | 38.62±0.07 |
| letter | 8.7±0.0 | 8.57±0.13 | 11.26±0.0 | 8.22±0.25 | 8.95±0.13 | 12.8±1.17 | 9.53±0.51 | 8.13±0.05 | 8.93±0.52 | 9.44±0.02 | 9.33±0.00 | 9.45±0.00 |
| lymphography | 99.17±0.94 | 99.34±0.93 | 84.16±4.27 | 84.38±3.28 | 86.77±9.24 | 100.0±0.0 | 99.3±1.02 | 98.76±0.88 | 96.82±3.69 | 99.35±1.30 | 99.15±1.26 | 99.76±0.49 |
| magic.gamma | 85.86±0.0 | 86.15±0.74 | 86.36±0.0 | 80.27±1.07 | 89.68±0.5 | 81.33±0.57 | 87.97±0.84 | 76.13±2.42 | 69.54±0.73 | 88.54±0.05 | 87.54±0.18 | 90.07±0.07 |
| mammography | 41.27±0.0 | 42.09±0.86 | 34.07±0.0 | 37.94±3.21 | 39.8±4.26 | 17.11±3.75 | 19.93±3.92 | 27.82±3.84 | 27.54±11.4 | 47.43±0.32 | 36.28±1.26 | 37.86±0.63 |
| mnist | 72.72±0.0 | 73.68±1.31 | 70.97±0.0 | 54.15±6.53 | 56.26±3.26 | 68.45±1.63 | 62.42±4.39 | 65.09±0.57 | 46.0±9.55 | 76.90±1.42 | 57.97±0.00 | 61.23±0.34 |
| musk | 100.0±0.0 | 100.0±0.0 | 100.0±0.0 | 40.39±26.1 | 100.0±0.0 | 92.21±6.32 | 100.0±0.0 | 100.0±0.0 | 99.91±0.17 | 100.00±0.00 | 97.00±2.27 | 100.00±0.00 |
| optdigits | 29.11±0.0 | 31.75±4.86 | 43.63±0.0 | 15.41±3.21 | 15.34±2.17 | 50.94±8.59 | 25.56±4.25 | 7.76±1.15 | 4.53±1.04 | 28.43±1.80 | 31.88±0.00 | 17.37±0.09 |
| pageblocks | 67.6±0.0 | 67.45±0.1 | 71.07±0.0 | 43.42±2.0 | 66.42±1.23 | 68.11±2.3 | 62.1±0.95 | 63.5±1.17 | 52.05±3.89 | 70.10±0.62 | 62.67±0.00 | 63.81±0.43 |
| pendigits | 96.99±0.0 | 91.89±3.3 | 78.55±0.0 | 58.79±5.21 | 48.44±5.92 | 66.41±7.58 | 61.14±4.73 | 33.35±2.85 | 9.34±7.78 | 61.58±0.87 | 66.33±0.00 | 83.07±0.64 |
| pima | 75.37±2.99 | 79.7±2.44 | 68.4±3.79 | 73.65±2.07 | 67.98±2.86 | 78.63±1.94 | 71.18±2.06 | 65.15±8.8 | 59.75±1.75 | 80.54±1.67 | 79.27±1.74 | 80.05±3.02 |
| satellite | 86.01±0.0 | 85.83±0.72 | 85.86±0.0 | 82.35±0.88 | 84.79±0.35 | 87.62±0.24 | 85.09±0.18 | 78.96±0.46 | 81.1±1.97 | 83.72±0.05 | 88.05±0.00 | 80.41±0.05 |
| satimage-2 | 96.69±0.0 | 96.16±0.0 | 88.46±0.0 | 94.53±0.55 | 68.21±3.15 | 94.7±1.19 | 88.05±5.1 | 95.89±0.1 | 76.28±8.21 | 55.24±1.54 | 92.78±0.00 | 92.18±0.17 |
| shuttle | 97.86±0.0 | 98.14±0.48 | 99.75±0.0 | 98.61±0.34 | 94.03±0.11 | 99.72±0.14 | 97.91±0.26 | 60.16±26.92 | 98.03±0.13 | 98.34±0.06 | 99.36±0.17 | 98.06±0.33 |
| skin | 98.24±0.41 | 94.78±2.33 | 61.68±1.85 | 64.58±1.09 | 69.08±0.5 | 32.46±1.0 | 76.37±5.79 | 42.18±1.84 | 42.99±3.28 | 68.80±9.13 | 51.04±1.72 | 88.31±0.43 |
| smtp | 50.53±5.92 | 50.2±6.39 | 48.09±7.38 | 1.1±0.12 | 50.37±6.15 | 3.81±3.83 | 40.81±13.36 | 32.4±8.48 | 30.73±22.82 | 36.83±4.84 | 38.18±9.90 | 50.01±5.79 |
| spambase | 83.32±0.0 | 83.65±0.58 | 72.71±0.0 | 88.26±1.32 | 83.8±0.52 | 86.78±0.59 | 72.89±0.42 | 82.09±0.21 | 75.26±2.44 | 85.92±0.06 | 80.99±0.00 | 79.68±0.07 |
| speech | 2.8±0.0 | 3.17±0.0 | 3.15±0.0 | 3.25±1.0 | 2.85±0.12 | 3.38±0.5 | 3.0±0.29 | 2.81±0.31 | 3.38±0.38 | 2.81±0.32 | 2.94±0.31 | 2.75±0.03 |
| stamps | 71.68±8.35 | 82.47±4.08 | 64.84±8.17 | 58.84±6.84 | 57.65±8.4 | 79.54±5.44 | 64.74±12.87 | 49.57±17.72 | 42.62±9.94 | 84.60±4.54 | 89.37±3.50 | 90.95±5.10 |
| thyroid | 80.94±0.0 | 81.03±0.31 | 60.57±0.0 | 79.66±5.62 | 81.67±0.97 | 51.51±12.75 | 82.22±0.87 | 80.09±0.89 | 69.06±8.1 | 81.60±0.14 | 67.00±0.00 | 70.13±0.33 |
| vertebral | 26.11±2.49 | 25.21±8.9 | 33.87±4.3 | 20.75±1.98 | 35.14±5.39 | 58.75±7.3 | 35.84±9.27 | 21.39±4.98 | 23.42±3.17 | 77.13±8.35 | 69.39±3.97 | 66.28±3.89 |
| vowels | 30.21±0.0 | 31.59±1.59 | 33.09±0.0 | 11.97±1.05 | 38.1±4.89 | 27.39±5.75 | 42.72±4.8 | 20.94±2.43 | 16.88±1.93 | 19.06±0.89 | 24.10±0.00 | 46.78±0.00 |
| waveform | 27.0±0.0 | 27.87±0.0 | 30.66±0.0 | 10.53±0.76 | 9.99±1.13 | 18.63±4.08 | 9.31±1.05 | 8.86±0.64 | 11.52±3.39 | 20.82±0.39 | 9.95±0.00 | 13.65±0.05 |
| wbc | 92.01±3.96 | 96.1±2.17 | 24.89±3.49 | 94.24±3.95 | 29.97±3.1 | 95.12±5.14 | 93.84±2.45 | 91.95±3.26 | 56.51±11.12 | 96.98±2.71 | 96.46±2.49 | 94.87±5.44 |
| wdbc | 82.03±3.28 | 90.47±7.91 | 93.64±3.06 | 71.98±8.57 | 68.82±12.42 | 95.6±6.12 | 84.3±4.44 | 78.77±4.26 | 84.32±8.89 | 85.01±10.18 | 93.82±8.30 | 90.40±8.16 |
| wilt | 12.25±0.0 | 12.2±1.6 | 15.74±0.0 | 8.81±0.51 | 25.41±1.36 | 28.94±3.31 | 17.24±0.46 | 10.86±1.37 | 7.08±0.17 | 24.03±0.09 | 77.46±0.00 | 56.06±0.53 |
| wine | 95.11±1.81 | 96.8±5.58 | 89.95±2.64 | 67.12±7.62 | 99.85±0.23 | 98.26±3.89 | 97.65±0.72 | 70.1±6.29 | 78.56±9.59 | 99.87±0.27 | 99.62±0.77 | 99.71±0.58 |
| wpbc | 46.11±2.74 | 69.02±13.91 | 41.2±2.62 | 40.73±3.15 | 60.35±4.88 | 89.31±5.37 | 54.61±3.75 | 38.88±3.82 | 74.88±5.58 | 89.20±4.96 | 67.36±4.34 | 81.34±4.06 |
| yeast | 48.26±0.0 | 48.12±0.49 | 48.94±0.0 | 46.78±0.37 | 49.74±0.72 | 49.55±1.36 | 51.05±1.65 | 50.77±2.18 | 49.21±3.88 | 49.73±0.08 | 51.00±0.00 | 52.10±0.00 |
| yelp | 16.03±0.0 | 16.35±0.0 | 16.14±0.0 | 13.15±0.29 | 13.0±1.57 | 10.4±0.09 | 12.8±0.02 | 13.13±0.39 | 10.02±1.01 | 16.66±0.20 | 13.90±3.00 | 15.81±0.22 |
| MNIST-C | 46.2±0.0 | 47.42±0.0 | 51.89±0.0 | 32.8±2.43 | 47.16±1.36 | 51.47±1.1 | 41.78±0.12 | 41.23±0.35 | 31.44±3.04 | 51.26±1.33 | 38.40±3.34 | 52.19±0.47 |
| FashionMNIST | 59.15±0.0 | 59.79±0.0 | 63.61±0.0 | 44.73±1.88 | 55.01±1.04 | 63.08±0.96 | 57.14±0.12 | 56.59±0.4 | 45.1±2.04 | 58.41±1.51 | 52.19±2.78 | 64.17±0.31 |
| CIFAR10 | 19.62±0.0 | 19.91±0.0 | 22.17±0.0 | 16.46±0.64 | 19.68±0.86 | 17.39±0.59 | 19.55±0.08 | 19.4±0.4 | 14.03±1.08 | 18.58±0.55 | 17.45±0.52 | 21.88±0.11 |
| SVHN | 15.34±0.0 | 15.53±0.0 | 15.97±0.0 | 13.85±0.49 | 15.47±0.47 | 15.6±0.31 | 15.08±0.04 | 14.93±0.18 | 12.4±0.95 | 15.67±0.50 | 14.67±0.33 | 16.71±0.07 |
| MVTec-AD | 75.76±2.71 | 82.94±2.68 | 75.79±2.95 | 70.0±3.07 | 85.11±2.96 | 89.46±2.23 | 73.66±2.72 | 72.62±2.74 | 83.78±3.32 | 88.91±1.02 | 73.25±1.79 | 83.78±0.54 |
| 20news | 13.47±0.52 | 15.61±1.35 | 15.04±0.67 | 11.56±0.43 | 17.33±2.36 | 14.62±0.88 | 11.48±0.44 | 11.49±0.51 | 12.9±1.9 | 20.96±0.69 | 13.51±0.84 | 18.86±1.48 |
| agnews | 16.68±0.0 | 17.35±0.0 | 25.86±0.0 | 11.94±0.32 | 19.22±2.97 | 15.42±0.38 | 11.85±0.03 | 12.42±0.31 | 10.22±1.85 | 23.97±0.89 | 15.16±3.17 | 20.50±0.22 |

Table 38. Individual dataset performances on ADBench w.r.t. AUROC (± standard dev. over five seeds). **Best** and Second-best results are highlighted.

| Dataset | kNN | DTE-NP | LOF | IForest | DTE-C | ICL | DDPM | GOAD | DeepSVDD | Tabpfn-OD | FoMo-0D | OUTFORMER |
|---|---|---|---|---|---|---|---|---|---|---|---|---|
| aloi | $51.04_{\pm 0.0}$ | $51.19_{\pm 0.46}$ | $48.76_{\pm 0.0}$ | $50.74_{\pm 0.68}$ | $50.44_{\pm 0.19}$ | $47.5_{\pm 0.98}$ | $49.91_{\pm 0.35}$ | $48.01_{\pm 0.92}$ | $50.89_{\pm 2.05}$ | $49.87_{\pm 0.10}$ | **$54.05_{\pm 0.30}$** | $53.04_{\pm 0.04}$ |
| amazon | $60.58_{\pm 0.0}$ | $60.82_{\pm 0.0}$ | $57.88_{\pm 0.0}$ | $56.4_{\pm 0.95}$ | $56.71_{\pm 2.15}$ | $54.21_{\pm 0.22}$ | $55.09_{\pm 0.1}$ | $56.07_{\pm 0.9}$ | $51.2_{\pm 4.42}$ | **$61.52_{\pm 0.49}$** | $54.74_{\pm 4.18}$ | $59.45_{\pm 0.34}$ |
| annthyroid | $92.81_{\pm 0.0}$ | $92.9_{\pm 0.25}$ | $88.63_{\pm 0.0}$ | $90.28_{\pm 1.52}$ | **$97.52_{\pm 0.15}$** | $81.11_{\pm 1.07}$ | $88.82_{\pm 1.34}$ | $81.01_{\pm 5.16}$ | $55.01_{\pm 3.62}$ | $91.74_{\pm 0.03}$ | $85.17_{\pm 0.00}$ | $94.45_{\pm 0.13}$ |
| backdoor | $93.75_{\pm 0.53}$ | $93.31_{\pm 1.7}$ | **$95.33_{\pm 0.25}$** | $74.89_{\pm 2.67}$ | $91.65_{\pm 1.73}$ | $93.62_{\pm 0.69}$ | $80.93_{\pm 0.56}$ | $52.9_{\pm 14.48}$ | $91.14_{\pm 2.62}$ | $84.79_{\pm 4.09}$ | $78.59_{\pm 17.11}$ | $92.22_{\pm 0.56}$ |
| breastw | $99.05_{\pm 0.26}$ | $99.28_{\pm 0.12}$ | $88.91_{\pm 6.8}$ | $99.5_{\pm 0.08}$ | $92.78_{\pm 1.75}$ | $98.28_{\pm 0.45}$ | $98.7_{\pm 0.43}$ | $98.86_{\pm 0.33}$ | $96.96_{\pm 0.92}$ | **$99.51_{\pm 0.13}$** | $99.15_{\pm 0.23}$ | $99.39_{\pm 0.12}$ |
| campaign | $78.48_{\pm 0.0}$ | $78.79_{\pm 0.25}$ | $70.55_{\pm 0.0}$ | $73.64_{\pm 1.48}$ | $77.95_{\pm 1.11}$ | **$80.92_{\pm 0.79}$** | $74.51_{\pm 0.42}$ | $47.89_{\pm 12.32}$ | $62.21_{\pm 12.88}$ | $72.99_{\pm 0.59}$ | $65.30_{\pm 0.70}$ | $78.37_{\pm 0.21}$ |
| cardio | $92.0_{\pm 0.0}$ | $91.8_{\pm 0.59}$ | $92.21_{\pm 0.0}$ | $93.32_{\pm 1.43}$ | $87.26_{\pm 1.0}$ | $80.01_{\pm 2.12}$ | $86.94_{\pm 1.96}$ | **$96.01_{\pm 0.27}$** | $65.43_{\pm 4.37}$ | $76.26_{\pm 0.34}$ | $94.27_{\pm 0.00}$ | $91.26_{\pm 0.00}$ |
| cardiotocography | $62.11_{\pm 0.0}$ | $63.76_{\pm 1.88}$ | $64.49_{\pm 0.0}$ | $74.24_{\pm 2.88}$ | $60.13_{\pm 2.59}$ | $54.2_{\pm 1.8}$ | $54.54_{\pm 2.96}$ | **$76.06_{\pm 1.4}$** | $47.75_{\pm 8.59}$ | $47.38_{\pm 0.56}$ | $72.15_{\pm 0.00}$ | $62.29_{\pm 0.00}$ |
| celeba | $73.14_{\pm 0.72}$ | $70.4_{\pm 0.37}$ | $43.73_{\pm 0.75}$ | $71.23_{\pm 2.27}$ | **$82.18_{\pm 2.38}$** | $72.21_{\pm 0.82}$ | $78.56_{\pm 1.99}$ | $43.8_{\pm 10.49}$ | $56.17_{\pm 22.46}$ | $75.62_{\pm 0.52}$ | $74.81_{\pm 0.61}$ | $66.15_{\pm 0.65}$ |
| census | **$72.26_{\pm 0.29}$** | $72.1_{\pm 0.4}$ | $58.46_{\pm 1.06}$ | $62.55_{\pm 2.38}$ | $69.62_{\pm 0.91}$ | $70.56_{\pm 0.35}$ | $70.15_{\pm 0.23}$ | $35.24_{\pm 4.19}$ | $54.16_{\pm 4.33}$ | $63.30_{\pm 3.62}$ | $60.54_{\pm 4.96}$ | $68.56_{\pm 1.52}$ |
| cover | $97.54_{\pm 0.15}$ | $97.73_{\pm 0.55}$ | $96.97_{\pm 0.24}$ | $86.31_{\pm 2.07}$ | $97.76_{\pm 1.28}$ | $89.34_{\pm 4.02}$ | $98.35_{\pm 0.66}$ | $13.83_{\pm 13.34}$ | $49.12_{\pm 14.74}$ | $63.82_{\pm 4.34}$ | $99.12_{\pm 0.29}$ | **$99.39_{\pm 0.12}$** |
| donors | $99.49_{\pm 0.06}$ | $99.26_{\pm 0.29}$ | $47.42_{\pm 0.0}$ | $89.44_{\pm 2.22}$ | $98.15_{\pm 0.37}$ | $99.9_{\pm 0.05}$ | $82.5_{\pm 1.83}$ | $33.57_{\pm 16.0}$ | $72.95_{\pm 17.81}$ | $95.19_{\pm 0.92}$ | **$99.91_{\pm 0.02}$** | $98.82_{\pm 0.07}$ |
| fault | $58.73_{\pm 0.0}$ | $58.64_{\pm 0.65}$ | **$94.35_{\pm 1.41}$** | $55.86_{\pm 2.03}$ | $59.46_{\pm 1.45}$ | $60.63_{\pm 0.5}$ | $61.09_{\pm 1.16}$ | $58.89_{\pm 0.61}$ | $54.31_{\pm 1.62}$ | $56.68_{\pm 0.52}$ | $61.82_{\pm 0.00}$ | $58.36_{\pm 0.00}$ |
| fraud | $95.43_{\pm 1.04}$ | **$95.64_{\pm 1.05}$** | $88.82_{\pm 1.98}$ | $81.09_{\pm 2.56}$ | $92.42_{\pm 2.3}$ | $92.78_{\pm 1.46}$ | $93.65_{\pm 0.92}$ | $69.75_{\pm 21.3}$ | $83.13_{\pm 6.6}$ | $91.00_{\pm 1.98}$ | $93.91_{\pm 2.81}$ | $95.46_{\pm 0.88}$ |
| glass | $92.04_{\pm 1.12}$ | $89.64_{\pm 3.54}$ | $66.92_{\pm 7.01}$ | $82.69_{\pm 2.75}$ | $92.42_{\pm 2.3}$ | **$99.44_{\pm 0.58}$** | $66.67_{\pm 13.63}$ | $59.03_{\pm 12.64}$ | $83.67_{\pm 16.2}$ | $98.98_{\pm 0.57}$ | $98.30_{\pm 0.53}$ | $94.50_{\pm 2.89}$ |
| hepatitis | $96.46_{\pm 1.46}$ | $93.22_{\pm 3.9}$ | **$99.98_{\pm 0.03}$** | $82.69_{\pm 2.75}$ | $98.78_{\pm 0.88}$ | $99.94_{\pm 0.13}$ | $97.74_{\pm 1.18}$ | $84.5_{\pm 3.25}$ | $99.57_{\pm 0.24}$ | $99.96_{\pm 0.09}$ | $99.88_{\pm 0.24}$ | $99.89_{\pm 0.22}$ |
| http | $100.0_{\pm 0.0}$ | $99.98_{\pm 0.03}$ | $49.57_{\pm 0.0}$ | $99.35_{\pm 0.29}$ | $99.45_{\pm 0.1}$ | $98.24_{\pm 3.45}$ | $100.0_{\pm 0.0}$ | $99.68_{\pm 0.13}$ | $61.31_{\pm 51.49}$ | $99.83_{\pm 0.08}$ | $99.92_{\pm 0.05}$ | **$100.00_{\pm 0.00}$** |
| imdb | $50.08_{\pm 0.0}$ | $50.43_{\pm 0.0}$ | **$71.72_{\pm 0.0}$** | $49.53_{\pm 0.78}$ | $48.05_{\pm 2.22}$ | $52.34_{\pm 0.49}$ | $47.91_{\pm 0.1}$ | $48.46_{\pm 0.65}$ | $49.97_{\pm 5.71}$ | $51.06_{\pm 1.58}$ | $50.84_{\pm 6.24}$ | $50.11_{\pm 0.66}$ |
| internetads | $68.08_{\pm 0.0}$ | $69.96_{\pm 2.22}$ | **$94.29_{\pm 2.2}$** | $47.87_{\pm 2.11}$ | $77.57_{\pm 1.54}$ | $72.2_{\pm 0.55}$ | $65.76_{\pm 0.06}$ | $65.65_{\pm 0.21}$ | $72.96_{\pm 3.24}$ | $60.69_{\pm 1.22}$ | $55.64_{\pm 7.12}$ | $76.98_{\pm 1.14}$ |
| ionosphere | $97.44_{\pm 0.98}$ | $97.77_{\pm 1.39}$ | $66.58_{\pm 0.0}$ | $91.21_{\pm 1.37}$ | $95.42_{\pm 0.58}$ | $98.98_{\pm 0.32}$ | $94.6_{\pm 0.85}$ | $91.54_{\pm 3.05}$ | $97.2_{\pm 1.26}$ | **$99.04_{\pm 0.24}$** | $96.59_{\pm 1.25}$ | $97.95_{\pm 0.86}$ |
| landsat | $68.25_{\pm 0.0}$ | $68.2_{\pm 1.75}$ | $44.83_{\pm 0.0}$ | $58.8_{\pm 2.21}$ | $52.79_{\pm 1.63}$ | $65.13_{\pm 0.44}$ | $51.37_{\pm 1.0}$ | $40.52_{\pm 2.32}$ | $59.44_{\pm 1.41}$ | $59.05_{\pm 0.14}$ | **$69.53_{\pm 0.00}$** | $52.09_{\pm 0.11}$ |
| letter | $35.43_{\pm 0.0}$ | $34.38_{\pm 0.98}$ | **$98.21_{\pm 0.75}$** | $32.04_{\pm 1.64}$ | $36.72_{\pm 0.95}$ | $42.68_{\pm 1.17}$ | $38.05_{\pm 1.12}$ | $31.08_{\pm 0.55}$ | $36.4_{\pm 3.05}$ | $42.17_{\pm 0.17}$ | $40.90_{\pm 0.00}$ | $38.33_{\pm 0.00}$ |
| lymphography | $99.93_{\pm 0.08}$ | $99.93_{\pm 0.09}$ | $83.4_{\pm 0.0}$ | $99.45_{\pm 0.32}$ | $98.99_{\pm 0.4}$ | **$100.0_{\pm 0.0}$** | $99.94_{\pm 0.09}$ | $99.89_{\pm 0.08}$ | $99.73_{\pm 0.3}$ | $99.95_{\pm 0.11}$ | $99.93_{\pm 0.11}$ | $99.98_{\pm 0.04}$ |
| magic.gamma | $83.27_{\pm 0.0}$ | $83.57_{\pm 0.76}$ | $85.52_{\pm 0.0}$ | $77.09_{\pm 1.29}$ | $87.5_{\pm 0.0}$ | $75.56_{\pm 0.42}$ | $85.97_{\pm 1.08}$ | $69.46_{\pm 2.39}$ | $62.97_{\pm 1.07}$ | $85.67_{\pm 0.05}$ | $84.77_{\pm 0.14}$ | **$88.41_{\pm 0.08}$** |
| mammography | $87.58_{\pm 0.0}$ | $87.62_{\pm 0.09}$ | **$92.93_{\pm 0.0}$** | $88.02_{\pm 0.3}$ | $86.42_{\pm 1.72}$ | $71.87_{\pm 9.11}$ | $81.01_{\pm 2.04}$ | $69.94_{\pm 8.59}$ | $71.5_{\pm 7.4}$ | $89.24_{\pm 0.13}$ | $69.11_{\pm 1.21}$ | $82.24_{\pm 0.41}$ |
| mnist | $93.85_{\pm 0.0}$ | $94.02_{\pm 0.42}$ | **$100.0_{\pm 0.0}$** | $86.6_{\pm 1.99}$ | $87.43_{\pm 2.48}$ | $90.11_{\pm 1.13}$ | $38.05_{\pm 1.12}$ | $90.07_{\pm 0.35}$ | $66.37_{\pm 11.03}$ | $94.21_{\pm 0.60}$ | $91.27_{\pm 0.00}$ | $87.97_{\pm 0.17}$ |
| musk | $100.0_{\pm 0.0}$ | $100.0_{\pm 0.0}$ | $96.65_{\pm 0.0}$ | $100.0_{\pm 0.0}$ | $100.0_{\pm 0.0}$ | $100.0_{\pm 0.0}$ | $100.0_{\pm 0.0}$ | $100.0_{\pm 0.0}$ | $99.99_{\pm 0.01}$ | **$100.00_{\pm 0.00}$** | $100.00_{\pm 0.00}$ | $100.00_{\pm 0.00}$ |
| optdigits | $93.72_{\pm 0.0}$ | $94.28_{\pm 1.67}$ | $91.3_{\pm 0.0}$ | $81.07_{\pm 3.28}$ | $82.38_{\pm 2.68}$ | **$97.18_{\pm 0.8}$** | $90.76_{\pm 2.12}$ | $67.46_{\pm 4.94}$ | $39.45_{\pm 18.65}$ | $91.89_{\pm 0.79}$ | $85.28_{\pm 0.00}$ | $85.57_{\pm 0.10}$ |
| pageblocks | $89.65_{\pm 0.0}$ | $89.32_{\pm 0.26}$ | **$99.05_{\pm 0.0}$** | $82.64_{\pm 0.89}$ | $89.89_{\pm 0.55}$ | $88.39_{\pm 0.77}$ | $86.93_{\pm 0.42}$ | $88.05_{\pm 1.19}$ | $78.39_{\pm 1.53}$ | $92.11_{\pm 0.13}$ | $86.58_{\pm 0.00}$ | $90.88_{\pm 0.17}$ |
| pendigits | **$99.87_{\pm 0.0}$** | $99.61_{\pm 0.17}$ | $70.53_{\pm 2.19}$ | $77.22_{\pm 0.48}$ | $97.79_{\pm 0.68}$ | $96.71_{\pm 0.83}$ | $98.11_{\pm 0.23}$ | $89.97_{\pm 2.03}$ | $46.29_{\pm 11.35}$ | $97.13_{\pm 0.14}$ | $98.11_{\pm 0.00}$ | $99.31_{\pm 0.03}$ |
| pima | $76.94_{\pm 1.87}$ | $81.5_{\pm 2.57}$ | $80.3_{\pm 0.0}$ | $74.26_{\pm 1.63}$ | $69.88_{\pm 1.95}$ | **$85.15_{\pm 0.48}$** | $70.27_{\pm 2.28}$ | $62.31_{\pm 13.7}$ | $57.99_{\pm 2.85}$ | $82.81_{\pm 1.11}$ | $80.46_{\pm 1.48}$ | $81.30_{\pm 1.15}$ |
| satellite | $82.24_{\pm 0.0}$ | $82.11_{\pm 0.67}$ | $99.38_{\pm 0.0}$ | $77.46_{\pm 1.48}$ | $78.61_{\pm 0.7}$ | **$99.48_{\pm 0.22}$** | $77.7_{\pm 0.53}$ | $68.76_{\pm 0.92}$ | $76.19_{\pm 2.66}$ | $80.02_{\pm 0.03}$ | $84.98_{\pm 0.00}$ | $73.77_{\pm 0.05}$ |
| satimage-2 | $99.71_{\pm 0.0}$ | $99.67_{\pm 0.0}$ | **$99.98_{\pm 0.0}$** | $99.12_{\pm 0.21}$ | $99.34_{\pm 0.07}$ | $99.92_{\pm 0.04}$ | $99.62_{\pm 0.16}$ | $98.99_{\pm 0.06}$ | $92.94_{\pm 2.84}$ | $96.68_{\pm 0.15}$ | $98.40_{\pm 0.00}$ | $99.75_{\pm 0.01}$ |
| shuttle | $99.91_{\pm 0.0}$ | **$99.93_{\pm 0.02}$** | $86.34_{\pm 1.77}$ | $99.65_{\pm 0.07}$ | $99.75_{\pm 0.0}$ | $6.58_{\pm 0.63}$ | $99.91_{\pm 0.01}$ | $70.44_{\pm 16.28}$ | $99.79_{\pm 0.07}$ | $99.93_{\pm 0.00}$ | $99.82_{\pm 0.09}$ | $99.92_{\pm 0.01}$ |
| skin | **$99.49_{\pm 0.08}$** | $98.86_{\pm 0.46}$ | $93.42_{\pm 2.48}$ | $89.42_{\pm 0.58}$ | $91.77_{\pm 0.22}$ | $68.68_{\pm 3.66}$ | $88.74_{\pm 4.4}$ | $64.95_{\pm 2.25}$ | $59.95_{\pm 4.47}$ | $90.57_{\pm 3.42}$ | $70.60_{\pm 0.88}$ | $97.01_{\pm 0.15}$ |
| smtp | $92.43_{\pm 2.7}$ | $92.98_{\pm 2.91}$ | $73.23_{\pm 0.0}$ | $90.36_{\pm 2.13}$ | $95.27_{\pm 1.28}$ | $74.36_{\pm 7.07}$ | **$95.43_{\pm 1.26}$** | $78.78_{\pm 13.12}$ | $85.24_{\pm 6.6}$ | $84.11_{\pm 1.17}$ | $83.04_{\pm 4.69}$ | $89.67_{\pm 3.21}$ |
| spambase | $83.36_{\pm 0.0}$ | $83.74_{\pm 0.69}$ | $37.53_{\pm 0.0}$ | $85.18_{\pm 1.69}$ | $83.01_{\pm 0.41}$ | $83.53_{\pm 0.45}$ | $64.54_{\pm 0.8}$ | $81.78_{\pm 0.36}$ | $70.24_{\pm 5.02}$ | **$85.54_{\pm 0.08}$** | $78.21_{\pm 0.00}$ | $79.73_{\pm 0.09}$ |
| speech | $36.36_{\pm 0.0}$ | $41.37_{\pm 0.0}$ | **$93.74_{\pm 2.36}$** | $37.7_{\pm 1.72}$ | $38.17_{\pm 0.57}$ | $48.86_{\pm 2.72}$ | $36.96_{\pm 0.86}$ | $36.63_{\pm 1.14}$ | $48.88_{\pm 2.88}$ | $37.18_{\pm 1.22}$ | $40.10_{\pm 1.75}$ | $37.39_{\pm 0.24}$ |
| stamps | $95.89_{\pm 1.44}$ | $97.87_{\pm 0.37}$ | $92.72_{\pm 0.0}$ | $93.47_{\pm 1.42}$ | $91.6_{\pm 2.04}$ | $96.68_{\pm 1.09}$ | $91.84_{\pm 4.15}$ | $81.46_{\pm 15.36}$ | $71.09_{\pm 3.68}$ | $97.73_{\pm 0.59}$ | $97.52_{\pm 0.39}$ | **$98.59_{\pm 0.57}$** |
| thyroid | $98.68_{\pm 0.0}$ | $98.63_{\pm 0.04}$ | $64.3_{\pm 1.31}$ | **$98.96_{\pm 0.2}$** | $98.74_{\pm 0.14}$ | $95.4_{\pm 0.98}$ | $97.95_{\pm 0.22}$ | $95.15_{\pm 0.85}$ | $88.77_{\pm 3.69}$ | $97.61_{\pm 0.06}$ | $96.84_{\pm 0.00}$ | $97.34_{\pm 0.05}$ |
| vertebral | $57.67_{\pm 3.58}$ | $54.3_{\pm 15.47}$ | $86.3_{\pm 0.0}$ | $45.64_{\pm 3.83}$ | $66.41_{\pm 1.9}$ | $79.19_{\pm 5.06}$ | $70.67_{\pm 6.28}$ | $46.73_{\pm 8.74}$ | $44.79_{\pm 1.71}$ | **$95.30_{\pm 1.82}$** | $87.20_{\pm 2.58}$ | $84.77_{\pm 2.56}$ |
| vowels | $82.21_{\pm 0.0}$ | $81.42_{\pm 1.44}$ | $76.0_{\pm 0.0}$ | $61.83_{\pm 0.62}$ | $86.93_{\pm 2.25}$ | $85.1_{\pm 2.13}$ | $86.38_{\pm 1.94}$ | $68.49_{\pm 3.71}$ | $55.73_{\pm 4.43}$ | $76.10_{\pm 0.58}$ | $81.29_{\pm 0.00}$ | **$87.14_{\pm 0.00}$** |
| waveform | $75.21_{\pm 0.0}$ | $74.48_{\pm 0.0}$ | **$80.52_{\pm 4.47}$** | $72.29_{\pm 1.52}$ | $65.21_{\pm 1.05}$ | $68.68_{\pm 3.66}$ | $62.17_{\pm 2.64}$ | $64.99_{\pm 3.1}$ | $59.94_{\pm 2.58}$ | $71.52_{\pm 0.10}$ | $71.02_{\pm 0.00}$ | $72.63_{\pm 0.07}$ |
| wbc | $99.12_{\pm 0.19}$ | $99.54_{\pm 0.28}$ | $99.62_{\pm 0.21}$ | $99.41_{\pm 0.37}$ | $80.53_{\pm 7.15}$ | **$99.66_{\pm 0.36}$** | $99.2_{\pm 0.39}$ | $99.14_{\pm 0.2}$ | $91.44_{\pm 3.64}$ | $99.65_{\pm 0.37}$ | $99.61_{\pm 0.31}$ | $99.55_{\pm 0.50}$ |
| wdbc | $99.05_{\pm 0.27}$ | $99.52_{\pm 0.38}$ | $68.81_{\pm 0.0}$ | $98.73_{\pm 0.64}$ | $68.53_{\pm 1.59}$ | **$99.78_{\pm 0.27}$** | $99.3_{\pm 0.17}$ | $98.96_{\pm 0.25}$ | $99.31_{\pm 0.4}$ | $99.35_{\pm 0.37}$ | $99.77_{\pm 0.25}$ | $99.64_{\pm 0.26}$ |
| wilt | $63.66_{\pm 0.0}$ | $62.91_{\pm 5.59}$ | **$98.36_{\pm 0.38}$** | $47.97_{\pm 3.12}$ | $85.1_{\pm 1.13}$ | $76.42_{\pm 3.46}$ | $71.66_{\pm 0.81}$ | $51.38_{\pm 5.33}$ | $34.41_{\pm 1.7}$ | $77.86_{\pm 0.06}$ | $97.13_{\pm 0.00}$ | $92.74_{\pm 0.07}$ |
| wine | $99.19_{\pm 0.22}$ | $99.44_{\pm 0.97}$ | $57.36_{\pm 2.01}$ | $93.92_{\pm 1.79}$ | $99.97_{\pm 0.04}$ | $99.87_{\pm 0.29}$ | $99.61_{\pm 0.09}$ | $94.11_{\pm 1.86}$ | $92.16_{\pm 4.34}$ | **$99.97_{\pm 0.05}$** | $99.94_{\pm 0.12}$ | $99.95_{\pm 0.10}$ |
| wpbc | $63.67_{\pm 2.34}$ | $83.16_{\pm 13.46}$ | $45.79_{\pm 0.0}$ | $56.33_{\pm 2.72}$ | $68.88_{\pm 2.7}$ | **$96.61_{\pm 1.17}$** | $66.49_{\pm 3.17}$ | $51.39_{\pm 5.74}$ | $82.67_{\pm 5.47}$ | $96.50_{\pm 1.53}$ | $76.10_{\pm 2.11}$ | $90.73_{\pm 1.61}$ |
| yeast | $44.74_{\pm 0.0}$ | $44.58_{\pm 0.33}$ | **$67.2_{\pm 0.0}$** | $41.8_{\pm 0.75}$ | $47.08_{\pm 1.1}$ | $48.98_{\pm 2.39}$ | $49.13_{\pm 2.85}$ | $52.53_{\pm 3.88}$ | $47.62_{\pm 5.96}$ | $48.92_{\pm 0.11}$ | $49.91_{\pm 0.00}$ | $51.15_{\pm 0.00}$ |
| yelp | $68.07_{\pm 0.0}$ | $68.66_{\pm 0.0}$ | **$87.21_{\pm 0.0}$** | $61.07_{\pm 0.53}$ | $59.94_{\pm 5.32}$ | $55.79_{\pm 0.5}$ | $59.3_{\pm 0.09}$ | $61.07_{\pm 0.98}$ | $49.9_{\pm 3.3}$ | $68.97_{\pm 0.61}$ | $60.29_{\pm 5.14}$ | $67.15_{\pm 0.25}$ |
| MNIST-C | $84.11_{\pm 0.0}$ | $84.74_{\pm 0.0}$ | **$91.6_{\pm 0.0}$** | $76.75_{\pm 1.45}$ | $86.1_{\pm 0.84}$ | $85.62_{\pm 0.52}$ | $80.12_{\pm 0.14}$ | $79.34_{\pm 0.41}$ | $64.7_{\pm 4.6}$ | $86.62_{\pm 0.60}$ | $77.28_{\pm 1.69}$ | $85.65_{\pm 0.21}$ |
| FashionMNIST | $89.87_{\pm 0.0}$ | $90.14_{\pm 0.0}$ | $70.3_{\pm 0.0}$ | $84.15_{\pm 1.07}$ | $90.21_{\pm 0.55}$ | $90.56_{\pm 0.27}$ | $88.52_{\pm 0.07}$ | $88.03_{\pm 0.24}$ | $75.45_{\pm 2.22}$ | $89.81_{\pm 0.62}$ | $86.66_{\pm 0.66}$ | **$91.21_{\pm 0.03}$** |
| CIFAR10 | $67.53_{\pm 0.0}$ | $67.82_{\pm 0.0}$ | $63.82_{\pm 0.0}$ | $64.04_{\pm 0.93}$ | $68.53_{\pm 1.59}$ | $63.6_{\pm 0.83}$ | $67.91_{\pm 0.13}$ | $67.53_{\pm 0.72}$ | $56.12_{\pm 2.2}$ | $65.54_{\pm 0.86}$ | $64.53_{\pm 0.62}$ | **$70.98_{\pm 0.20}$** |
| SVHN | $61.69_{\pm 0.0}$ | $62.13_{\pm 0.0}$ | **$80.36_{\pm 2.14}$** | $58.99_{\pm 0.88}$ | $62.91_{\pm 1.1}$ | $61.7_{\pm 0.5}$ | $61.37_{\pm 0.08}$ | $60.75_{\pm 0.49}$ | $53.92_{\pm 3.05}$ | $62.44_{\pm 0.38}$ | $59.93_{\pm 0.64}$ | $64.45_{\pm 0.04}$ |
| MVTec-AD | $81.51_{\pm 1.82}$ | $89.66_{\pm 1.51}$ | $60.19_{\pm 0.62}$ | $77.38_{\pm 1.95}$ | $89.39_{\pm 2.0}$ | **$94.75_{\pm 0.84}$** | $78.0_{\pm 1.99}$ | $77.06_{\pm 2.1}$ | $89.55_{\pm 2.19}$ | $94.22_{\pm 0.46}$ | $75.31_{\pm 1.26}$ | $86.30_{\pm 0.29}$ |
| 20news | $57.36_{\pm 0.66}$ | $59.97_{\pm 0.74}$ | **$74.58_{\pm 0.0}$** | $54.91_{\pm 1.17}$ | $64.3_{\pm 3.17}$ | $61.3_{\pm 1.05}$ | $54.88_{\pm 0.52}$ | $54.92_{\pm 1.04}$ | $55.64_{\pm 4.17}$ | $67.38_{\pm 0.39}$ | $56.93_{\pm 1.46}$ | $65.43_{\pm 0.81}$ |
| agnews | $67.05_{\pm 0.0}$ | $67.98_{\pm 0.0}$ |  | $58.43_{\pm 1.09}$ | $68.16_{\pm 3.22}$ | $62.55_{\pm 0.47}$ | $57.82_{\pm 0.12}$ | $59.87_{\pm 0.86}$ | $49.83_{\pm 5.63}$ | **$74.43_{\pm 0.78}$** | $60.96_{\pm 3.69}$ | $72.20_{\pm 0.26}$ |

*Table 39* Individual dataset performances on ADBench w.r.t. F1 (± standard dev. over five seeds). Best and Second-best results are highlighted.

| Dataset | KNN | DTE-NP | LOF | IForest | DTE-C | ICL | DDPM | GOAD | DeepSVDD | Tabpfn-OD | FoMo-OD | OUTFORMER |
|---|---|---|---|---|---|---|---|---|---|---|---|---|
| aloi | $5.9_{\pm 0.0}$ | $5.82_{\pm 0.07}$ | $8.16_{\pm 0.0}$ | $4.2_{\pm 0.26}$ | $4.2_{\pm 0.2}$ | $4.91_{\pm 0.56}$ | $6.76_{\pm 0.19}$ | $5.73_{\pm 1.45}$ | $5.17_{\pm 0.92}$ | $6.37_{\pm 0.14}$ | $7.82_{\pm 0.41}$ | $6.06_{\pm 0.10}$ |
| amazon | $11.4_{\pm 0.0}$ | $10.8_{\pm 0.0}$ | $10.0_{\pm 0.0}$ | $11.28_{\pm 0.64}$ | $11.8_{\pm 1.53}$ | $9.4_{\pm 0.32}$ | $11.08_{\pm 0.11}$ | $11.56_{\pm 0.26}$ | $11.76_{\pm 2.19}$ | $11.48_{\pm 0.84}$ | $11.72_{\pm 3.24}$ | $9.72_{\pm 0.53}$ |
| annthyroid | $61.99_{\pm 0.0}$ | $61.84_{\pm 1.59}$ | $49.63_{\pm 0.0}$ | $55.02_{\pm 4.22}$ | $77.72_{\pm 0.5}$ | $49.44_{\pm 3.88}$ | $57.23_{\pm 2.96}$ | $55.77_{\pm 4.62}$ | $23.33_{\pm 5.12}$ | $63.41_{\pm 0.15}$ | $49.25_{\pm 0.00}$ | $62.43_{\pm 0.54}$ |
| backdoor | $52.01_{\pm 1.92}$ | $51.48_{\pm 17.26}$ | $72.42_{\pm 2.18}$ | $4.07_{\pm 2.4}$ | $82.58_{\pm 2.44}$ | $87.15_{\pm 1.1}$ | $9.6_{\pm 0.62}$ | $4.79_{\pm 3.47}$ | $82.96_{\pm 3.28}$ | $23.41_{\pm 6.12}$ | $45.12_{\pm 11.91}$ | $82.17_{\pm 1.83}$ |
| breastw | $95.77_{\pm 0.19}$ | $96.66_{\pm 0.71}$ | $85.4_{\pm 5.93}$ | $96.94_{\pm 0.46}$ | $88.18_{\pm 2.86}$ | $95.91_{\pm 0.68}$ | $95.04_{\pm 0.75}$ | $95.66_{\pm 0.34}$ | $91.85_{\pm 0.77}$ | $97.07_{\pm 0.70}$ | $95.92_{\pm 0.79}$ | $97.11_{\pm 0.75}$ |
| campaign | $50.37_{\pm 0.0}$ | $50.98_{\pm 0.61}$ | $42.24_{\pm 0.0}$ | $43.7_{\pm 0.91}$ | $52.12_{\pm 0.62}$ | $51.03_{\pm 0.73}$ | $50.4_{\pm 0.68}$ | $22.62_{\pm 9.05}$ | $37.89_{\pm 12.9}$ | $44.18_{\pm 0.52}$ | $39.72_{\pm 0.60}$ | $48.96_{\pm 0.20}$ |
| cardio | $61.93_{\pm 0.0}$ | $63.07_{\pm 0.0}$ | $62.5_{\pm 0.0}$ | $67.5_{\pm 3.32}$ | $58.3_{\pm 0.76}$ | $52.16_{\pm 5.49}$ | $61.7_{\pm 1.73}$ | $74.89_{\pm 0.93}$ | $38.4_{\pm 4.19}$ | $57.95_{\pm 0.62}$ | $72.16_{\pm 0.00}$ | $66.48_{\pm 0.00}$ |
| cardiotocography | $46.35_{\pm 0.0}$ | $46.78_{\pm 1.44}$ | $48.28_{\pm 0.0}$ | $56.14_{\pm 2.75}$ | $38.37_{\pm 2.23}$ | $38.93_{\pm 2.89}$ | $38.84_{\pm 2.75}$ | $59.96_{\pm 1.08}$ | $37.08_{\pm 5.46}$ | $32.83_{\pm 0.24}$ | $56.22_{\pm 0.00}$ | $46.78_{\pm 0.00}$ |
| celeba | $17.19_{\pm 0.83}$ | $15.81_{\pm 0.69}$ | $1.91_{\pm 0.47}$ | $17.33_{\pm 2.29}$ | $17.35_{\pm 3.48}$ | $12.69_{\pm 1.83}$ | $26.02_{\pm 2.89}$ | $3.98_{\pm 2.98}$ | $8.43_{\pm 5.5}$ | $19.78_{\pm 0.80}$ | $8.41_{\pm 0.88}$ | $6.39_{\pm 0.66}$ |
| census | $22.52_{\pm 0.51}$ | $22.21_{\pm 0.6}$ | $13.09_{\pm 0.42}$ | $10.54_{\pm 1.4}$ | $17.43_{\pm 2.38}$ | $23.96_{\pm 0.54}$ | $20.27_{\pm 0.48}$ | $4.96_{\pm 2.13}$ | $19.28_{\pm 1.44}$ | $19.11_{\pm 3.26}$ | $17.40_{\pm 5.36}$ | $19.39_{\pm 1.59}$ |
| cover | $65.1_{\pm 2.15}$ | $66.84_{\pm 7.4}$ | $82.4_{\pm 2.16}$ | $11.61_{\pm 1.24}$ | $71.04_{\pm 9.46}$ | $39.96_{\pm 12.71}$ | $76.86_{\pm 1.33}$ | $0.0_{\pm 0.0}$ | $3.43_{\pm 3.44}$ | $2.46_{\pm 2.11}$ | $69.92_{\pm 7.24}$ | $86.35_{\pm 2.10}$ |
| donors | $94.91_{\pm 0.67}$ | $92.9_{\pm 2.83}$ | $74.47_{\pm 2.04}$ | $43.46_{\pm 3.54}$ | $82.17_{\pm 2.54}$ | $97.22_{\pm 1.04}$ | $25.01_{\pm 7.13}$ | $4.29_{\pm 4.13}$ | $41.44_{\pm 0.4}$ | $60.92_{\pm 7.01}$ | $97.85_{\pm 0.79}$ | $86.37_{\pm 0.89}$ |
| fault | $55.57_{\pm 0.0}$ | $56.2_{\pm 0.4}$ | $50.67_{\pm 0.0}$ | $53.64_{\pm 1.34}$ | $56.23_{\pm 1.73}$ | $57.59_{\pm 0.56}$ | $58.45_{\pm 1.15}$ | $55.96_{\pm 0.68}$ | $54.92_{\pm 1.38}$ | $53.52_{\pm 0.57}$ | $57.50_{\pm 0.00}$ | $54.98_{\pm 0.00}$ |
| fraud | $45.22_{\pm 4.87}$ | $48.36_{\pm 5.41}$ | $59.47_{\pm 4.5}$ | $28.03_{\pm 4.09}$ | $68.23_{\pm 13.11}$ | $57.43_{\pm 5.97}$ | $73.16_{\pm 2.29}$ | $37.28_{\pm 25.8}$ | $58.11_{\pm 14.77}$ | $56.13_{\pm 6.24}$ | $61.34_{\pm 11.80}$ | $27.48_{\pm 6.20}$ |
| glass | $25.87_{\pm 13.76}$ | $24.58_{\pm 16.83}$ | $20.46_{\pm 8.79}$ | $16.18_{\pm 7.01}$ | $37.46_{\pm 6.39}$ | $87.83_{\pm 9.1}$ | $32.81_{\pm 13.36}$ | $20.15_{\pm 11.0}$ | $45.39_{\pm 21.87}$ | $82.71_{\pm 10.90}$ | $77.60_{\pm 6.07}$ | $74.64_{\pm 15.70}$ |
| hepatitis | $81.29_{\pm 5.04}$ | $79.03_{\pm 11.95}$ | $41.95_{\pm 10.62}$ | $54.0_{\pm 5.54}$ | $92.8_{\pm 3.32}$ | $99.64_{\pm 0.79}$ | $86.69_{\pm 4.26}$ | $57.86_{\pm 7.77}$ | $93.82_{\pm 1.29}$ | $99.64_{\pm 0.71}$ | $99.64_{\pm 0.71}$ | $99.64_{\pm 0.71}$ |
| http | $100.0_{\pm 0.0}$ | $97.43_{\pm 3.53}$ | $96.78_{\pm 3.13}$ | $25.8_{\pm 22.46}$ | $24.59_{\pm 15.3}$ | $60.7_{\pm 53.56}$ | $99.68_{\pm 0.3}$ | $56.39_{\pm 17.61}$ | $25.0_{\pm 22.46}$ | $78.17_{\pm 9.87}$ | $89.24_{\pm 5.79}$ | $98.16_{\pm 0.82}$ |
| imdb | $5.4_{\pm 0.0}$ | $5.2_{\pm 0.0}$ | $6.4_{\pm 0.0}$ | $6.2_{\pm 0.49}$ | $7.24_{\pm 1.62}$ | $10.56_{\pm 0.54}$ | $5.56_{\pm 0.09}$ | $5.64_{\pm 0.65}$ | $9.96_{\pm 2.71}$ | $6.76_{\pm 0.51}$ | $10.12_{\pm 4.42}$ | $5.44_{\pm 0.23}$ |
| internetads | $51.9_{\pm 0.0}$ | $53.21_{\pm 3.77}$ | $54.62_{\pm 0.0}$ | $26.41_{\pm 4.44}$ | $64.78_{\pm 2.48}$ | $55.87_{\pm 0.91}$ | $45.87_{\pm 0.35}$ | $46.14_{\pm 0.52}$ | $54.29_{\pm 5.92}$ | $40.27_{\pm 3.79}$ | $40.82_{\pm 7.48}$ | $61.63_{\pm 1.84}$ |
| ionosphere | $90.47_{\pm 2.17}$ | $91.51_{\pm 2.09}$ | $87.53_{\pm 2.54}$ | $83.43_{\pm 3.5}$ | $89.58_{\pm 0.9}$ | $94.18_{\pm 1.62}$ | $88.64_{\pm 1.58}$ | $83.42_{\pm 5.88}$ | $93.07_{\pm 1.23}$ | $94.26_{\pm 0.74}$ | $92.31_{\pm 1.33}$ | $94.13_{\pm 1.08}$ |
| landsat | $51.46_{\pm 0.0}$ | $51.22_{\pm 2.55}$ | $53.64_{\pm 0.0}$ | $43.27_{\pm 1.34}$ | $38.29_{\pm 2.94}$ | $53.82_{\pm 0.35}$ | $40.23_{\pm 1.02}$ | $32.96_{\pm 1.19}$ | $42.18_{\pm 2.53}$ | $42.94_{\pm 0.30}$ | $52.66_{\pm 0.00}$ | $41.55_{\pm 0.06}$ |
| letter | $1.0_{\pm 0.0}$ | $1.0_{\pm 0.0}$ | $10.0_{\pm 0.0}$ | $3.8_{\pm 1.1}$ | $2.4_{\pm 1.67}$ | $7.2_{\pm 0.84}$ | $3.6_{\pm 0.89}$ | $1.2_{\pm 0.45}$ | $5.0_{\pm 1.58}$ | $2.80_{\pm 0.40}$ | $4.00_{\pm 0.00}$ | $5.00_{\pm 0.00}$ |
| lymphography | $94.5_{\pm 6.47}$ | $95.78_{\pm 5.81}$ | $74.87_{\pm 7.44}$ | $85.05_{\pm 4.72}$ | $82.01_{\pm 3.83}$ | $100.0_{\pm 0.0}$ | $95.19_{\pm 6.67}$ | $93.13_{\pm 5.46}$ | $89.82_{\pm 9.49}$ | $97.37_{\pm 5.26}$ | $96.39_{\pm 5.14}$ | $97.89_{\pm 4.21}$ |
| magic.gamma | $76.17_{\pm 0.0}$ | $76.46_{\pm 0.81}$ | $76.08_{\pm 0.0}$ | $69.64_{\pm 1.25}$ | $80.78_{\pm 0.8}$ | $69.55_{\pm 0.47}$ | $78.88_{\pm 0.95}$ | $62.72_{\pm 1.48}$ | $59.88_{\pm 0.89}$ | $78.87_{\pm 0.11}$ | $77.89_{\pm 0.22}$ | $80.91_{\pm 0.10}$ |
| mammography | $40.38_{\pm 0.0}$ | $42.38_{\pm 1.03}$ | $38.46_{\pm 0.0}$ | $39.23_{\pm 2.48}$ | $37.31_{\pm 3.91}$ | $17.38_{\pm 3.62}$ | $24.62_{\pm 4.04}$ | $35.62_{\pm 5.7}$ | $31.62_{\pm 10.21}$ | $49.92_{\pm 0.45}$ | $37.00_{\pm 0.62}$ | $38.46_{\pm 0.24}$ |
| mnist | $71.86_{\pm 0.0}$ | $72.6_{\pm 1.21}$ | $71.43_{\pm 0.0}$ | $52.6_{\pm 4.64}$ | $58.46_{\pm 2.88}$ | $64.89_{\pm 1.95}$ | $60.37_{\pm 3.89}$ | $63.91_{\pm 1.13}$ | $43.31_{\pm 11.21}$ | $73.34_{\pm 1.94}$ | $62.00_{\pm 0.00}$ | $57.09_{\pm 0.37}$ |
| musk | $100.0_{\pm 0.0}$ | $100.0_{\pm 0.0}$ | $100.0_{\pm 0.0}$ | $35.88_{\pm 24.78}$ | $100.0_{\pm 0.0}$ | $83.3_{\pm 8.97}$ | $100.0_{\pm 0.0}$ | $100.0_{\pm 0.0}$ | $99.18_{\pm 1.13}$ | $100.00_{\pm 0.00}$ | $93.20_{\pm 3.23}$ | $100.00_{\pm 0.00}$ |
| optdigits | $21.33_{\pm 0.0}$ | $27.2_{\pm 10.73}$ | $53.33_{\pm 0.0}$ | $12.8_{\pm 6.28}$ | $10.93_{\pm 3.39}$ | $57.73_{\pm 8.41}$ | $28.4_{\pm 7.05}$ | $0.4_{\pm 0.37}$ | $0.0_{\pm 0.0}$ | $34.00_{\pm 2.73}$ | $37.33_{\pm 0.00}$ | $8.93_{\pm 0.33}$ |
| pageblocks | $59.02_{\pm 0.0}$ | $59.29_{\pm 0.26}$ | $65.88_{\pm 0.0}$ | $42.63_{\pm 2.29}$ | $62.12_{\pm 0.47}$ | $64.9_{\pm 1.29}$ | $50.31_{\pm 0.73}$ | $50.24_{\pm 1.07}$ | $54.71_{\pm 3.06}$ | $68.47_{\pm 0.91}$ | $61.18_{\pm 0.00}$ | $59.29_{\pm 0.79}$ |
| pendigits | $90.38_{\pm 0.0}$ | $83.46_{\pm 6.02}$ | $76.28_{\pm 0.0}$ | $57.95_{\pm 4.75}$ | $56.03_{\pm 8.28}$ | $61.15_{\pm 5.82}$ | $64.62_{\pm 2.62}$ | $41.54_{\pm 3.49}$ | $12.31_{\pm 12.2}$ | $59.62_{\pm 1.22}$ | $69.23_{\pm 0.00}$ | $79.36_{\pm 0.75}$ |
| pima | $70.56_{\pm 2.49}$ | $74.69_{\pm 2.25}$ | $66.75_{\pm 3.0}$ | $69.57_{\pm 2.68}$ | $65.25_{\pm 3.25}$ | $73.54_{\pm 2.63}$ | $66.62_{\pm 2.63}$ | $59.19_{\pm 11.77}$ | $55.95_{\pm 2.36}$ | $75.53_{\pm 1.61}$ | $74.65_{\pm 1.84}$ | $76.30_{\pm 1.89}$ |
| satellite | $71.81_{\pm 0.0}$ | $71.91_{\pm 0.99}$ | $72.64_{\pm 0.0}$ | $67.12_{\pm 0.64}$ | $72.33_{\pm 0.63}$ | $74.99_{\pm 0.46}$ | $73.74_{\pm 0.27}$ | $63.58_{\pm 0.48}$ | $67.76_{\pm 2.88}$ | $71.29_{\pm 0.14}$ | $77.75_{\pm 0.00}$ | $68.43_{\pm 0.16}$ |
| satimage-2 | $90.14_{\pm 0.0}$ | $90.14_{\pm 0.0}$ | $81.69_{\pm 0.0}$ | $89.58_{\pm 1.61}$ | $75.48_{\pm 0.9}$ | $88.45_{\pm 1.54}$ | $78.59_{\pm 5.12}$ | $90.7_{\pm 0.77}$ | $73.24_{\pm 6.68}$ | $51.55_{\pm 2.61}$ | $88.73_{\pm 0.00}$ | $84.51_{\pm 0.00}$ |
| shuttle | $98.23_{\pm 0.0}$ | $98.3_{\pm 0.1}$ | $98.41_{\pm 0.0}$ | $96.71_{\pm 0.53}$ | $97.99_{\pm 0.02}$ | $98.82_{\pm 0.16}$ | $98.3_{\pm 0.08}$ | $56.26_{\pm 30.2}$ | $98.11_{\pm 0.09}$ | $98.21_{\pm 0.06}$ | $97.72_{\pm 0.20}$ | $98.10_{\pm 0.07}$ |
| skin | $96.35_{\pm 0.62}$ | $95.08_{\pm 1.16}$ | $70.8_{\pm 2.09}$ | $78.06_{\pm 0.72}$ | $82.23_{\pm 0.43}$ | $1.09_{\pm 0.97}$ | $73.42_{\pm 5.35}$ | $52.05_{\pm 1.57}$ | $43.26_{\pm 2.59}$ | $78.23_{\pm 6.75}$ | $49.67_{\pm 1.90}$ | $89.59_{\pm 0.73}$ |
| smtp | $69.5_{\pm 4.43}$ | $69.59_{\pm 4.42}$ | $65.82_{\pm 5.88}$ | $0.0_{\pm 0.0}$ | $69.5_{\pm 4.43}$ | $6.96_{\pm 12.38}$ | $56.19_{\pm 11.77}$ | $48.56_{\pm 12.43}$ | $34.0_{\pm 23.28}$ | $52.67_{\pm 6.11}$ | $36.65_{\pm 2.80}$ | $68.05_{\pm 5.12}$ |
| spambase | $80.52_{\pm 0.0}$ | $80.67_{\pm 4.48}$ | $73.97_{\pm 0.0}$ | $80.49_{\pm 1.34}$ | $80.02_{\pm 0.23}$ | $79.27_{\pm 0.49}$ | $63.55_{\pm 0.95}$ | $78.81_{\pm 0.63}$ | $69.55_{\pm 3.79}$ | $81.64_{\pm 0.08}$ | $74.27_{\pm 0.00}$ | $77.67_{\pm 0.12}$ |
| speech | $3.28_{\pm 0.0}$ | $4.92_{\pm 0.0}$ | $3.28_{\pm 0.0}$ | $3.93_{\pm 2.49}$ | $3.93_{\pm 1.47}$ | $2.62_{\pm 1.87}$ | $2.95_{\pm 1.37}$ | $2.95_{\pm 1.37}$ | $1.31_{\pm 1.37}$ | $1.31_{\pm 0.66}$ | $2.95_{\pm 1.23}$ | $1.64_{\pm 0.00}$ |
| stamps | $75.47_{\pm 9.45}$ | $85.93_{\pm 3.97}$ | $63.52_{\pm 13.2}$ | $63.62_{\pm 8.81}$ | $57.97_{\pm 11.6}$ | $77.17_{\pm 7.83}$ | $62.98_{\pm 12.09}$ | $52.72_{\pm 15.8}$ | $37.07_{\pm 9.74}$ | $81.39_{\pm 3.84}$ | $85.72_{\pm 3.70}$ | $87.47_{\pm 2.54}$ |
| thyroid | $75.27_{\pm 0.0}$ | $74.84_{\pm 0.96}$ | $52.69_{\pm 0.0}$ | $80.43_{\pm 3.08}$ | $75.48_{\pm 0.9}$ | $56.13_{\pm 8.72}$ | $75.48_{\pm 2.07}$ | $74.19_{\pm 1.32}$ | $65.59_{\pm 8.6}$ | $76.34_{\pm 0.00}$ | $59.14_{\pm 0.00}$ | $64.09_{\pm 1.10}$ |
| vertebral | $23.82_{\pm 5.02}$ | $21.56_{\pm 14.78}$ | $41.26_{\pm 4.76}$ | $15.84_{\pm 2.0}$ | $42.13_{\pm 11.49}$ | $63.39_{\pm 5.17}$ | $37.54_{\pm 12.52}$ | $18.25_{\pm 10.64}$ | $16.71_{\pm 5.9}$ | $75.12_{\pm 7.20}$ | $64.44_{\pm 3.04}$ | $63.85_{\pm 2.66}$ |
| vowels | $26.0_{\pm 0.0}$ | $29.6_{\pm 0.89}$ | $34.0_{\pm 0.0}$ | $15.2_{\pm 3.63}$ | $37.2_{\pm 4.15}$ | $24.4_{\pm 8.29}$ | $41.6_{\pm 5.37}$ | $23.6_{\pm 2.19}$ | $20.8_{\pm 4.15}$ | $24.40_{\pm 1.96}$ | $28.00_{\pm 0.00}$ | $38.00_{\pm 0.00}$ |
| waveform | $27.0_{\pm 0.0}$ | $26.0_{\pm 0.0}$ | $28.0_{\pm 0.0}$ | $10.2_{\pm 2.17}$ | $12.2_{\pm 2.77}$ | $26.8_{\pm 5.81}$ | $12.0_{\pm 1.22}$ | $9.8_{\pm 2.39}$ | $14.6_{\pm 3.58}$ | $21.00_{\pm 0.00}$ | $8.00_{\pm 0.00}$ | $18.60_{\pm 0.80}$ |
| wbc | $86.41_{\pm 3.23}$ | $89.35_{\pm 3.45}$ | $20.27_{\pm 9.64}$ | $88.25_{\pm 2.41}$ | $32.49_{\pm 0.28}$ | $92.88_{\pm 4.57}$ | $86.03_{\pm 5.16}$ | $86.45_{\pm 4.97}$ | $54.17_{\pm 11.28}$ | $89.20_{\pm 8.43}$ | $91.15_{\pm 6.41}$ | $91.82_{\pm 9.16}$ |
| wdbc | $78.7_{\pm 2.32}$ | $85.05_{\pm 6.83}$ | $85.63_{\pm 6.59}$ | $70.91_{\pm 11.05}$ | $68.08_{\pm 14.3}$ | $90.52_{\pm 9.27}$ | $79.28_{\pm 7.36}$ | $75.82_{\pm 5.69}$ | $83.34_{\pm 7.3}$ | $81.22_{\pm 8.99}$ | $92.67_{\pm 5.32}$ | $88.82_{\pm 7.59}$ |
| wilt | $2.33_{\pm 0.0}$ | $2.41_{\pm 1.49}$ | $16.73_{\pm 0.0}$ | $2.02_{\pm 0.33}$ | $17.59_{\pm 2.54}$ | $35.18_{\pm 2.59}$ | $20.23_{\pm 0.91}$ | $12.45_{\pm 2.0}$ | $0.62_{\pm 0.35}$ | $31.05_{\pm 0.75}$ | $77.04_{\pm 0.00}$ | $63.27_{\pm 0.31}$ |
| wine | $87.16_{\pm 5.61}$ | $92.13_{\pm 11.85}$ | $80.84_{\pm 3.22}$ | $71.05_{\pm 4.25}$ | $98.29_{\pm 2.42}$ | $99.32_{\pm 1.52}$ | $90.31_{\pm 3.17}$ | $65.52_{\pm 5.99}$ | $69.81_{\pm 9.29}$ | $99.32_{\pm 1.36}$ | $99.32_{\pm 1.36}$ | $99.32_{\pm 1.36}$ |
| wpbc | $49.09_{\pm 2.1}$ | $68.16_{\pm 14.02}$ | $41.26_{\pm 4.76}$ | $36.63_{\pm 1.93}$ | $57.76_{\pm 6.44}$ | $90.55_{\pm 0.97}$ | $50.71_{\pm 4.98}$ | $34.22_{\pm 3.42}$ | $70.22_{\pm 6.11}$ | $89.77_{\pm 3.23}$ | $58.02_{\pm 1.22}$ | $77.60_{\pm 2.93}$ |
| yeast | $46.75_{\pm 0.0}$ | $46.15_{\pm 0.44}$ | $47.73_{\pm 0.0}$ | $44.46_{\pm 0.86}$ | $49.23_{\pm 1.12}$ | $50.26_{\pm 1.38}$ | $50.85_{\pm 2.02}$ | $53.21_{\pm 2.43}$ | $49.47_{\pm 5.66}$ | $50.10_{\pm 0.28}$ | $50.49_{\pm 0.00}$ | $51.08_{\pm 0.00}$ |
| yelp | $18.8_{\pm 0.0}$ | $19.6_{\pm 0.0}$ | $20.6_{\pm 0.0}$ | $15.8_{\pm 0.62}$ | $14.72_{\pm 1.51}$ | $8.88_{\pm 0.73}$ | $16.08_{\pm 0.18}$ | $15.36_{\pm 0.38}$ | $10.4_{\pm 1.77}$ | $20.60_{\pm 0.25}$ | $16.12_{\pm 3.95}$ | $18.96_{\pm 0.75}$ |
| MNIST-C | $46.3_{\pm 0.0}$ | $47.51_{\pm 0.0}$ | $52.96_{\pm 0.0}$ | $34.7_{\pm 2.59}$ | $48.68_{\pm 1.5}$ | $52.07_{\pm 1.25}$ | $42.44_{\pm 0.25}$ | $42.11_{\pm 0.45}$ | $34.45_{\pm 3.0}$ | $51.61_{\pm 1.44}$ | $39.10_{\pm 2.44}$ | $51.70_{\pm 0.40}$ |
| FashionMNIST | $59.05_{\pm 0.0}$ | $59.68_{\pm 0.0}$ | $63.3_{\pm 0.0}$ | $45.33_{\pm 2.4}$ | $59.22_{\pm 0.99}$ | $62.9_{\pm 1.07}$ | $56.67_{\pm 0.27}$ | $56.25_{\pm 0.6}$ | $47.96_{\pm 2.33}$ | $59.09_{\pm 1.59}$ | $53.44_{\pm 2.28}$ | $62.53_{\pm 0.23}$ |
| CIFAR10 | $22.85_{\pm 0.0}$ | $23.19_{\pm 0.0}$ | $27.07_{\pm 0.0}$ | $18.19_{\pm 1.18}$ | $23.79_{\pm 1.09}$ | $20.59_{\pm 1.37}$ | $22.98_{\pm 0.41}$ | $22.88_{\pm 0.89}$ | $16.46_{\pm 1.96}$ | $21.63_{\pm 1.06}$ | $20.03_{\pm 1.05}$ | $25.99_{\pm 0.23}$ |
| SVHN | $18.95_{\pm 0.0}$ | $19.21_{\pm 0.0}$ | $19.23_{\pm 0.0}$ | $16.45_{\pm 1.06}$ | $19.49_{\pm 0.61}$ | $19.55_{\pm 0.99}$ | $18.73_{\pm 0.23}$ | $18.49_{\pm 0.42}$ | $15.2_{\pm 1.66}$ | $18.67_{\pm 0.55}$ | $17.73_{\pm 0.63}$ | $20.01_{\pm 0.10}$ |
| MVTec-AD | $67.37_{\pm 3.1}$ | $75.69_{\pm 2.88}$ | $67.29_{\pm 3.16}$ | $64.6_{\pm 2.65}$ | $78.98_{\pm 3.08}$ | $82.63_{\pm 2.26}$ | $65.02_{\pm 2.84}$ | $63.83_{\pm 3.2}$ | $76.78_{\pm 3.79}$ | $81.83_{\pm 1.02}$ | $64.20_{\pm 1.71}$ | $75.54_{\pm 0.61}$ |
| 20news | $14.61_{\pm 1.48}$ | $17.83_{\pm 1.58}$ | $16.77_{\pm 1.37}$ | $10.49_{\pm 1.66}$ | $18.95_{\pm 4.69}$ | $16.43_{\pm 1.74}$ | $10.55_{\pm 1.27}$ | $10.7_{\pm 1.03}$ | $13.64_{\pm 3.63}$ | $24.84_{\pm 1.79}$ | $14.59_{\pm 2.22}$ | $21.79_{\pm 1.87}$ |
| agnews | $20.1_{\pm 0.0}$ | $20.7_{\pm 0.0}$ | $30.6_{\pm 0.0}$ | $12.48_{\pm 0.65}$ | $23.93_{\pm 3.44}$ | $17.79_{\pm 0.72}$ | $12.52_{\pm 0.18}$ | $13.15_{\pm 0.69}$ | $10.66_{\pm 3.57}$ | $31.04_{\pm 1.04}$ | $17.60_{\pm 4.62}$ | $24.25_{\pm 0.33}$ |

