# OpenReview forum: "From Zero to Hero: Advancing Zero-Shot Foundation Models for Tabular Outlier Detection"
_ICML.cc/2026/Conference — ICML 2026 regular_

### Official Review · Reviewer_TFNm · 2026-03-05

**Soundness:** 3
**Presentation:** 2
**Significance:** 3
**Originality:** 3
**Overall Recommendation:** 4
**Confidence:** 3

**Summary:**

This paper introduces OUTFORMER, a zero-shot foundation model for tabular outlier detection (OD). To address the deployment challenges caused by the lack of labeled outliers in real-world applications, this work builds upon existing foundation models (e.g., FOMO-0D) by innovatively introducing a "mixture of synthetic priors" composed of Gaussian Mixture Models (GMM), Structural Causal Models (SCM), and Copula models for pre-training. Furthermore, to overcome the training gradient conflicts brought by mixing multiple complex priors, the authors propose a Self-Evolving Curriculum (SEC) algorithm based on multi-armed bandits. Experiments demonstrate that OUTFORMER achieves SOTA performance on the existing ADBench and two newly constructed large-scale benchmark datasets (OddBench and OvRBench), while maintaining highly efficient inference.

**Compliance With Llm Reviewing Policy:**

Affirmed.

**Final Justification:**

I appreciate the authors’ detailed response. It clarifies most of the issues I raised, so I will keep my positive rating (4: Weak accept) unchanged.

**Key Questions For Authors:**

I do not have additional questions apart from listed weaknesses.

**Limitations:**

Please refer to the Weaknesses section (mainly focusing on formatting/presentation issues and the absence of certain SEC algorithm hyperparameter ablation studies, specifically the lack of discussion on the granularity of the
 task bins).

**Strengths And Weaknesses:**

### Strengths
* **i. Innovative construction of mixed synthetic priors:** Building upon previous works (e.g., FOMO-0D, TabPFN), the authors significantly expand the types of underlying mathematical models used for synthetic datasets. By going beyond GMMs and incorporating SCMs and Copula models, the approach aims to comprehensively cover and simulate the highly complex and diverse outlier distributions found in the real world.
* **ii. Proposal of an effective Self-Evolving Curriculum (SEC) algorithm:** To help the model better "digest" these diverse and varying-difficulty data priors and avoid gradient conflicts, the paper cleverly proposes the SEC algorithm. This algorithm can dynamically and adaptively sample the synthetic data that holds the highest learning value for the model at any given moment, successfully unlocking the performance potential of the mixed priors.
* **iii. Solid experiments and significant open-source contributions:** The model performs exceptionally well on mainstream OD datasets. Notably, the authors constructed and open-sourced OddBench and OvRBench, containing over 1400 datasets, which greatly fills the gap of large-scale evaluation benchmarks in this field. Additionally, the paper provides highly detailed ablation studies (e.g., convergence analysis of training steps, independent effectiveness proofs for the three data priors, and necessity arguments for the SEC algorithm), rigorously proving the effectiveness of each module.

### Weaknesses
* **i. Suboptimal formatting and visual presentation:** The layout of the paper is somewhat crowded, negatively impacting the reading experience. For instance, placing a large table as early as page two interrupts the flow of the text. Furthermore, the paper is dense with tables but lacks intuitive methodological diagrams (for instance, the visual presentation of the Self-evolving Curriculum Training algorithm is not intuitive and easy to understand). Additionally, the chart layout on page eight is somewhat cluttered. It would be better if the layout could be more concise and reader-friendly.
* **ii. Missing key hyperparameter analysis in ablation studies:** In the analysis of the Self-Evolving Curriculum (SEC) algorithm, there is a lack of ablation experiments regarding the number of "task categories" (which are defined by $P$ distinct priors and $K$ dimensionality bins). The paper fails to discuss the specific impact of the granularity of the dimension bins ($K$) on the resource allocation efficiency of the multi-armed bandit and the ultimate generalization performance of the model.

---

> ### Author Rebuttal · Authors · 2026-03-30
>
> Thank you for your valuable comments.
>
> **W1: Suboptimal formatting and visual presentation.**  We agree that the presentation can be improved.
>
> Regarding the comment that *“placing a large table early interrupts the flow”*, we intentionally position Table 1 early to highlight results on ADBench, a widely adopted OD benchmark. Achieving SOTA on this benchmark is a key contribution, and Figure 1 complements it by demonstrating strong performance with low inference-only latency, emphasizing the rare combination of accuracy and efficiency.  We will consider the format of the table and placement of this content in our later verions.
>
> For the concern that *“the SEC visualization is not intuitive”*, we will improve Figure 2 (e.g., larger, clearer design). The figure is intended to complement Sections 3–4 and serve as a visual aid rather than a standalone explanation.
>
> Finally, we acknowledge that *“the chart layout is cluttered”*. This stems from extensive experiments and ablations supporting our contributions (heterogeneous priors and SEC). While some results are included in the main text to aid understanding (for example, Table in Sec 3 motivates SEC), we agree that readability can be improved. In the final version, we will move selected ablations to the appendix and streamline the main presentation.
>
>
> **W2: Lack of ablation on the number of task categories (priors × dimensionality bins).**
> We conduct an additional ablation by varying the number of bins per prior, and compare against a model without SEC (*No Bins*). Similar to ablation studies listed in the paper, all models are trained with a smaller OutFormer (4 layers), keeping other hyperparameters fixed while only changing the number of bins. All models are trained on 4GPUs for 1,500 epochs until convergence.
>
> **Table 1: Ablation on number of bins, Bold for best and italicize for second-best**
>
> | Setting | Avg. Rank | ELO | Win Rate | rAUC | CΔ |
> |--------|----------:|----:|---------:|-----:|----:|
> | No Bins | 4.14 $\pm$ 1.9 | 939 | 0.36 | 0.948 $\pm$ 0.07 | 0.27 |
> | Bins = 1 (×5 priors) | *2.98 $\pm$ 1.6* | *995* | 0.47 | 0.960 $\pm$ 0.05 | *0.19* |
> | Bins = 5 (×5 priors) | **2.65 $\pm$ 1.7** | 990 | **0.65** | **0.972 $\pm$ 0.05** | **0.13** |
> | Bins = 10 (×5 priors) | 3.03 $\pm$ 1.5 | **1048** | *0.58* | *0.969 $\pm$ 0.06* | 0.20 |
> | Bins = 20 (×5 priors) | 3.75 $\pm$ 1.7 | 957 | 0.42 | 0.950 $\pm$ 0.06 | 0.28 |
> | Bins = 30 (×5 priors) | 3.60 $\pm$ 1.7 | 972 | 0.45 | 0.954 $\pm$ 0.08 | 0.24 |
>
> Table 1 shows the ADBench performances of OutFormer trained on different bin sizes, with five major metrics. Increasing the number of bins (1 $\rightarrow$ 5) **improves generalization performance by enabling finer-grained curriculum control and better separation of task difficulty**. The model trained with 10 bins also shows competitive performance.
>
> However, with a limited number of datasets per epoch (only 1,000 datasets per epoch), large bin counts (e.g., 20 $\times$ 5 priors equals 100 in total) result in only ~10 samples per task category for estimating and updating the rewards. This leads to *noisy* reward estimates and *insufficient* learning signals, degrading stability and generalization. Consequently, performance drops at larger bin sizes (20–30), only slightly better than the no-SEC baseline. We note that this limitation can be mitigated by increasing the number of samples per epoch, though at the cost of longer pretraining time.

---

> > ### Author Rebuttal · Reviewer_TFNm · 2026-04-01
> >
> > I appreciate the authors’ detailed response. It clarifies most of the issues I raised, so I will keep my positive rating (**4: Weak accept**) unchanged.

---

> > > ### Author Response · Authors · 2026-04-03
> > >
> > > We sincerely appreciate the reviewer’s thoughtful evaluation and the positive ratings, as well as the acknowledgment that all concerns have been fully addressed. Your proposed additional experiments, particularly the ablation on bin sizes, have further strengthened the manuscript. We hope the current version reflects these improvements, and we would be grateful if the final assessment could be updated accordingly. We remain fully committed to incorporating all of your valuable suggestions (especially suggestions about the layout) in the future version.

---

### Official Review · Reviewer_Gsaw · 2026-03-09

**Soundness:** 3
**Presentation:** 3
**Significance:** 3
**Originality:** 2
**Overall Recommendation:** 5
**Confidence:** 4

**Summary:**

One key challenge in outlier detection (OD) lies in model selection and hyperparameter tuning. A promising approach is to train a foundation model that can leverage training data as in-context examples to produce predictions without dataset-specific retraining. Prior work such as TabPFN and FOMO-0D has explored this idea using PFNs with promising results. OUTFORMER extends this line of research by improving synthetic data generation and introducing a self-evolving curriculum learning strategy. Empirical results show that OUTFORMER achieves the best performance on the ADBench benchmark.

**Compliance With Llm Reviewing Policy:**

Affirmed.

**Final Justification:**

I like the idea of foundational models for anomaly detection. This work advances the performance by including more data priors and curriculum learning. Extensive evaluations are made.

The rebuttal satisfactorily addressed my concerns.

But I do agree with another reviewer that the organization of the tables and figures should be revised. It is indeed dense and looks weird.

**Key Questions For Authors:**

- I am curious about the efficiency results reported in Figure 1. It is unclear what exactly the reported runtime corresponds to. In particular, PFN-based models are typically pre-trained offline and do not require dataset-specific training, whereas many deep OD baselines need to be trained from scratch on each dataset. It would therefore be helpful to clarify whether the reported runtime includes dataset-specific training for all methods, and whether the inference time is measured per dataset or per test sample.

- From the figure, OutFormer appears to have a larger runtime than FOMO-0D. A more detailed discussion of this difference would be helpful to better understand the computational trade-offs between the two approaches.

- While I appreciate the extensive empirical evaluation, the comparison is largely limited to aggregate rankings and performance metrics. It would be interesting to include a case study comparing OutFormer and FOMO-0D, illustrating how the additional data priors (e.g., SCM and copulas) contribute to improved anomaly detection performance.

**Limitations:**

Yes.

**Strengths And Weaknesses:**

Strengths

- The paper is well written and easy to follow, with a clear and well-organized discussion of the current state-of-the-art methods in tabular outlier detection.

- The core idea of PFNs is to pre-train on synthetically generated datasets drawn from a broad distribution. In this context, combining data generated from different priors intuitively improves the generalizability of the PFN model.

- For heterogeneous datasets, introducing a curriculum learning strategy to mitigate gradient interference appears reasonable and well-motivated.

- Extensive experimental evaluation demonstrates strong performance on multiple benchmark datasets, and the ablation studies provide evidence supporting the effectiveness of the proposed curriculum learning design.

Weakness

- While I do think the paper is solid with strong performance and appreciate the effort of the authors, the contribution appears to be incremental on TabPFN and FOMO-OD, rather than introducing a fundamentally new modeling paradigm.

---

> ### Author Rebuttal · Authors · 2026-03-30
>
> Thank you for your feedback.
>
> **W1 Contribution is incremental on TabPFN architecture instead of introducing a fundamentally new modeling paradigm**: Foundation models typically scale and improve along three axes: **(1) modeling architecture**, **(2) data**, and **(3) training algorithms**. While our work does not introduce a new architectural paradigm beyond TabPFN, we make substantive contributions on the other two fronts: For **data**: We design a *mixture of heterogeneous priors* that expands coverage over diverse outlier archetypes (e.g., structural, measurement, dependence), mitigating the narrow inductive bias of single-prior training like FoMo. For **training**, we introduce a *self-evolving curriculum (SEC)*  that adaptively schedules tasks based on difficulty, improving learning stability and generalization across priors.
>
> Together, these advances substantially enhance zero-shot performance and robustness, demonstrating that scaling along the data and training axes alone can yield meaningful gains even with a fixed architecture, which is novel for tabular foundation models.
>
> **Q1 efficiency results reported in Figure 1 in our paper**:  In Figure 1, we report *training + inference time* per dataset for deep and shallow OD baselines, as they must be retrained from scratch on each dataset. For PFN-based foundation models (TabPFN-OD, FoMo-0D, OutFormer), we *exclude pretraining time* and only include inference time, since it is a one-time cost and does not affect per-dataset deployment, whether on one or a mliion datasets. We will add this clarification in the paper.
>
> All timings are measured *per dataset (not per sample)*, allowing users to estimate end-to-end latency in practice. For OutFormer, we report runtime with 50 context ensembles. We also use a batched/parallelized implementation for both FoMo-0D and OutFormer, which explains why FoMo-0D appears faster than reported in its original paper.
>
> **Q2 Computational trade-off between OutFormer and FoMo-0D**:  OutFormer incurs additional runtime in two aspects: (1) *Ensembling*: multiple inference passes (up to 50) to improve performance, especially on high-dimensional data;  (2) *Model size*: a larger architecture than FoMo-0D (~50M vs. 5M parameters). Despite this, Figure 1 shows that OutFormer remains efficient, with sub-linear latency scaling in the number of ensembles (see response to Reviewer aM69), making the performance–efficiency trade-off favorable in practice.
>
>
> **Q3 Case Study comparing OutFormer and FoMo-0D**: To showcase cases where OutFormer benefits from additional priors beyond FoMo-0D’s GMM, we perform a goodness-of-fit test [1] on ADBench datasets. For each dataset $D_{\text{real}}$, we fit a GMM (up to 5 components, select the best GMM via BIC measure), sample $D_{\text{syn}}$ from fitted GMM, and conduct a two-sample test. A small p-value (≤ 0.05) indicates that the GMM prior fails to capture the data distribution.
>
> When GMM is a poor fit (p ≤ 0.05), OutFormer shows substantial gains (Avg. Rank: **5.13 vs. 6.93**). We show representative cases that show OutFormer greatly improves over FoMo-0D in Table 1 below, demonstrating the benefit of heterogeneous priors over a single GMM prior.
>
> **Table 1: Example datasets with goodness-of-fit p-values vs. AUROC (Rank)**
> | Dataset | FoMo | OutFormer |
> |---|---:|---:|
> | skin (p=0.001) | 0.706 (9) | 0.970 (3) |
> | fraud (p=0.0002) | 0.939 (6) | 0.955 (2) |
> | Satimage-2 (p=0.005) | 0.984 (10) | 0.998 (1) |
> | Mammography (p=0.003) | 0.691 (12) | 0.822 (7) |
> | Pageblocks (p=0.006) | 0.866 (10) | 0.910 (3) |
> | Campaign (p=0.002) | 0.653 (10) | 0.784 (4) |
> | Census (p=0.009) | 0.605 (9) | 0.686 (6) |
> | Backdoor (p=0.015) | 0.786 (10) | 0.922 (5) |
>
> When GMM fits well (p > 0.05), OutFormer still improves (Avg. Rank: **3.56 vs. 4.39**), suggesting complementary gains from curriculum learning and ensembling. Additionally, OutFormer achieves strong improvements on high-dimensional and large-scale datasets (as shown in Table 2), highlighting robustness beyond prior mismatch scenarios.
>
> **Table 2: Example large and high-dimensional datasets (N: sample sizes, D: dimensions)**
> | Dataset | FoMo | OutFormer |
> |--|----:|------:|
> | internetads (N=1,966, D=1555) | 0.556 (11) | 0.770 (2) |
> | CIFAR10 (N=5,263, D=512) | 0.645 (9) | 0.710 (1) |
> | MNIST-C (N=10,000, D=512) | 0.773 (10) | 0.857 (4) |
> | 20news (N=11,905, D=768) | 0.569 (8) | 0.654 (2) |
> | agnews (N=10,000, D=768) | 0.610 (8) | 0.722 (3) |
>
> While improvements arise from multiple factors (priors, training dynamics, ensembling), the results consistently support the effectiveness of expanding the prior space. We will include decision boundary visualizations and attention-based analyses in the later versions of our manuscript.
>
> [1]  Huber-Carol, Catherine, et al., eds. Goodness-of-fit tests and model validity. Springer Science & Business Media, 2012.

---

> > ### Author Rebuttal · Reviewer_Gsaw · 2026-03-31
> >
> > The author solved my concerns. Therefore, I suggest accepting this paper.

---

> > > ### Author Response · Authors · 2026-04-03
> > >
> > > We sincerely thank the reviewer for raising the score to the 4–5 range, and we truly appreciate your thoughtful evaluation and positive ratings. We are also grateful for your acknowledgment that the concerns have been addressed.
> > >
> > > We remain fully committed to incorporating all of your valuable suggestions in the next version of the manuscript. In particular, we will include additional clarifications on inference time and running time in Figure 1, a deeper analysis of why OutFormer is slower than FoMo-0D, and expanded case studies with goodness-of-fit test.

---

### Official Review · Reviewer_UMJv · 2026-03-11

**Soundness:** 3
**Presentation:** 3
**Significance:** 3
**Originality:** 2
**Overall Recommendation:** 4
**Confidence:** 3

**Summary:**

This paper introduces OUTFORMER, a zero-shot foundation model for tabular outlier detection. It is pretrained on a large set of synthetic anomaly-detection tasks and then applied to new datasets without dataset-specific training. Its main contribution is the combination of mixed synthetic priors and a self-evolving curriculum, which together improve generalization across large benchmark suites.

**Compliance With Llm Reviewing Policy:**

Affirmed.

**Final Justification:**

The strong empirical results contributed to my final decision to accept the paper.

**Key Questions For Authors:**

1. Ensemble Efficiency: What is the performance-efficiency trade-off if the ensemble size is reduced from 50 to a smaller number for real-time deployment?

2. Out-of-Distribution Priors: How does the model handle anomalies that violate all three synthetic archetypes, such as temporal drifts or adversarial manipulation?

3. Scalability of Priors: Can the self-evolving curriculum scale to include more complex, non-parametric priors without increasing training instability?

**Limitations:**

yes

**Strengths And Weaknesses:**

**Strengths**
- Zero-shot applicability: It can be deployed on new tabular datasets without requiring labeled anomalies or per-dataset retraining.

- Diverse training priors: It uses GMM, SCM, and Copula-based synthetic priors to expose the model to a broader range of anomaly structures.

- Curriculum design: It goes beyond naive prior mixing by introducing an adaptive curriculum that selects training categories dynamically.

**Weakness**
- Categorical Data Handling: The synthetic priors are primarily designed for numerical distributions, leaving the model’s efficacy on high-cardinality categorical variables under-explored.

- Computational and Inference Overhead: Training requires significant resources (e.g., 6 days on 4x A6000), and the reliance on an ensemble of 50 models may limit its use in real-time, low-latency applications.

- Context Window Constraints: Performance is inherently capped by the Transformer’s sequence length, necessitating sub-optimal feature bagging for very high-dimensional datasets.

---

> ### Author Rebuttal · Authors · 2026-03-30
>
> **W1 Categorical Data Handling**: We have acknowledged the lack of categorical-specific priors as part of our limitations (Appx. B). In addition, we evaluate OutFormer on 7 datasets: IEEE-CIS-Fraud (mixed numeric+categorical) [1] and repurposed TALENT [2] datasets (>15 categorical features only) which treat the majority class as inliers and subsample others as outliers. OutFormer handles categorical data via: (i) one-hot + subsampling (to 100 dims), and (ii) feature projection (Sec 3.2, XStream [3]).
>
> Table1: AUROC on categorical data
> | Dataset | iForest | OutFormer (w/o proj.) | OutFormer (w/ proj.) |
> |--|--:|--:|--:|
> | CustomerSatisfaction | 0.873 | **0.912** | 0.755 |
> | Mfeat_pixel | 0.901 | **0.944** | 0.894 |
> | MIC | 0.789 | **0.789** | 0.765 |
> | PhishingWebsites | 0.701 | **0.887** | 0.770 |
> | StudentDropout | **0.685** | 0.650 | 0.660 |
> | AndroidPermissions | 0.693 | **0.793** | 0.751 |
> | IEEE-CIS-Fraud | **0.741** | 0.738 | 0.739 |
>
> OutFormer (w/o projection) outperforms iForest on **5/7 datasets**, showing that **existing priors generalize to categorical settings**. Projection degrades performance due to information loss. Dedicated categorical priors may further help for future work.
>
> **W2 & Q1 Analysis of Ensemble sizes**: While OutFormer requires non-trivial pretraining, this is **a one-time cost and remains modest compared to LLM-scale training**. For inference, we quantify the latency–ensemble trade-off on ADBench using a size estimate:
> $$
> S = \frac{\text{num context points} \times \text{num features}}{5000 \times 100}
> $$
> Datasets with number of contexts < 5000 and \(d < 100\) require no ensembling; others are grouped as $\textit{Small}\ (1 < S < 5),\ \textit{Medium}\ (5 < S < 30),\ \textit{Large}\ (S > 30)$
>
> Table 2: Avg. inference time (s) vs. ensemble size
> | Size | No Ens. | Ens. (5) | Ens. (10) | Ens. (20) | Ens. (50) | Ens. (100) |
> |--|--:|--:|--:|--:|--:|--:|
> | No Ens. | 0.09 | 0.09 | 0.09 | 0.09 | 0.09 | 0.09 |
> | Small | 0.44 | 0.88 | 0.90 | 1.67 | 3.23 | 6.42 |
> | Medium | 1.09 | 3.26 | 4.03 | 6.08 | 12.23 | 20.89 |
> | Large | 1.23 | 3.92 | 4.43 | 7.35 | 12.87 | 20.50 |
>
> Table 3: Mean rAUC vs. ensemble size
> | Size | No Ens. | Ens. (5) | Ens. (10) | Ens. (20) | Ens. (50) | Ens. (100) |
> |--|--:|--:|--:|--:|--:|--:|
> | Small | 0.963 | **0.965** | 0.965 | 0.964 | 0.965 | 0.964 |
> | Medium | 0.958 | 0.974 | 0.978 | 0.977 | **0.983** | 0.978 |
> | Large | 0.924 | 0.973 | 0.984 | 0.986 | 0.987 | **0.990** |
>
> Tables 2–3 report latency and performance (rescaled AUROC). Latency scales **sub-linearly** with ensemble size due to batching/parallelism. Ensemble improves rAUC via diverse context averaging. Gains saturate on small datasets but continue for larger ones. In addition We measure efficiency:
>
> $$
> \mathrm{Eff}(k) = \frac{(\mathrm{rAUC}(k) - \mathrm{rAUC}(0) )\times 100 }{T(k) - T(0)}
> $$
>
> Table 4: Efficiency vs. ensemble size
> | Size   | Ens. (5) | Ens. (10) | Ens. (20) | Ens. (50) | Ens. (100) |
> |--|--:|--:|--:|--:|--:|
> | Small  | **0.455** | 0.435 | 0.081 | 0.072 | 0.017 |
> | Medium | **0.737** | 0.680 | 0.381 | 0.224 | 0.101 |
> | Large  | 1.822 | **1.875** | 1.013 | 0.541 | 0.343 |
>
> Table 4 shows that **for real-time deployment, small ensembles (5–10) provide the best efficiency**.
>
> **W3 Sub-optimal feature bagging**: Context optimization methods(e.g. ICD [4] and TuneTables [5]) can be combined with our model for feature/context selection, but introduce additional inference-time latency (minutes to hours on large datasets). In contrast, our ensembling approach achieves strong performance gains with negligible overhead, leveraging parallelism and batching because OutFormer utilizes a single forward-pass inference.
>
> **Q2 OOD Data**: Real-world datasets are inherently OOD. While true anomaly archetypes are unobservable, our benchmarks (OddBench, OnevRBench) include diverse shifts (e.g., temporal drift, system faults, adversarial network traffic). We further demonstrate OOD generalization in Table 2 of our paper: models trained on a single prior transfer to unseen archetypes (e.g., SCM-structure $\rightarrow$ SCM-measurement: AUROC 0.962; Copula-dependence $\rightarrow$ Copula-probability: AUROC 0.813).
>
>
> **Q3 Complex, nonparametric priors**: Yes. SEC does not rely on explicit prior parameterization and naturally extends to complex/nonparametric priors. Empirically (Table 7), expanding the prior set consistently improves performance on ADBench, suggesting strong compositional generalization. We would love others to expand our prior catalog.
>
> [1] Howard et al. IEEE-CIS Fraud Detection. https://kaggle.com/competitions/ieee-fraud-detection
>
> [2] Liu et al. TALENT: A tabular analytics and learning toolbox. JMLR. 2025;26(226):1-6.
>
> [3] Manzoor et al. XStream: Outlier Detection in Feature-Evolving Data Streams. KDD 2018.
>
> [4]: Ma et. al.. In-context data distillation with tabpfn. arXiv:2402.06971
>
> [5]: Feuer et al. Tunetables: Context optimization for scalable prior-data fitted networks. NeurIPS (2024)

---

> > ### Author Rebuttal · Reviewer_UMJv · 2026-04-04
> >
> > The strong results you provided have resolved my concerns, and I will increase my score.

---

> > > ### Author Response · Authors · 2026-04-06
> > >
> > > We sincerely thank the reviewer for raising the score from 3 to 4, and for the thoughtful, positive evaluation. We appreciate your acknowledgment that the concerns have been addressed.
> > >
> > > We will incorporate all your valuable suggestions in the next revision, including additional analysis of the performance–latency trade-off across ensemble sizes and expanded comparisons of OutFormer with baselines on high-dimensional categorical datasets.

---

### Official Review · Reviewer_aM69 · 2026-03-12

**Soundness:** 3
**Presentation:** 2
**Significance:** 3
**Originality:** 3
**Overall Recommendation:** 4
**Confidence:** 3

**Summary:**

This paper introduces OUTFORMER, a foundation model for zero-shot tabular outlier detection. The model leverages a diverse mixture of synthetic priors—including GMMs, SCMs, and Copulas—to generate varied inlier and outlier archetypes. A key contribution is the Self-Evolving Curriculum (SEC), a multi-armed bandit framework that addresses gradient interference during pretraining on heterogeneous data. At inference, OUTFORMER performs zero-shot detection via in-context learning with context ensembling. Extensive evaluations across multiple benchmarks demonstrate state-of-the-art performance with fast inference.

**Compliance With Llm Reviewing Policy:**

Affirmed.

**Key Questions For Authors:**

Please see the weakness:
1. Can the authors provide a more detailed breakdown of inference latency when scaling the ensemble size up to 50 contexts?
2. How well do these priors transfer to tabular datasets with categorical variables?

**Limitations:**

yes

**Strengths And Weaknesses:**

# Strengths

1. Soundness: The Self-Evolving Curriculum (SEC) eloquently solves the non-trivial challenge of training across mixed datasets with varying distributions and difficulties.
2. Significance: The empirical evaluation is exceptionally comprehensive. The introduction of OddBench and OvRBench provides a massive and diverse testbed for the OD community, enhancing the paper's contribution.


# Weaknesses

1. Soundness: The framework relies on aggregating over 50 random contexts/subsamples to achieve its peak performance and to handle large datasets. While still faster than training a deep model, this linearly scales the inference latency compared to a pure single forward-pass FM.
2. Significance: The synthetic priors primarily model continuous numerical data. It remains somewhat unclear how well these priors transfer to tabular datasets dominated by high-cardinality categorical variables.

---

> ### Author Rebuttal · Authors · 2026-03-30
>
> Thank you for the valuable comments.
>
> **Q1 & W1, Inference latency vs Ensembles**: we conducted an additional experiment on ADBench to quantify the trade-off between inference latency and ensemble size. We partition datasets by an effective size measure:
>
> $$
> S = \frac{\text{num context points} \times \text{num features}}{5000 \times 100}
> $$
>
> where 5,000 and 100 are the maximum number of inlier context points and dimensions for each forward pass, respectively. Datasets with fewer than 5,000 clean inlier context points and fewer than 100 features do not benefit from ensembling (“No Ens. Required”). The remaining datasets are grouped as: $\textit{Small}\ (1 < S < 5),\ \textit{Medium}\ (5 < S < 30),\ \textit{Large}\ (S > 30)$.  The avg. inference time (seconds) across ensemble sizes is below:
>
> | Size | No Ens. | Ens. (5) | Ens. (10) | Ens. (20) | Ens. (50) | Ens. (100) |
> |----|------:|---------:|----------:|----------:|----------:|-----------:|
> | No Ens. Required | 0.09 | 0.09 | 0.09 | 0.09 | 0.09 | 0.09 |
> | Small | 0.44 | 0.88 | 0.90 | 1.67 | 3.23 | 6.42 |
> | Medium | 1.09 | 3.26 | 4.03 | 6.08 | 12.23 | 20.89 |
> | Large | 1.23 | 3.92 | 4.43 | 7.35 | 12.87 | 20.50 |
> | **Total Avg.** | 0.84 | 1.95 | 2.61 | 3.86 | 5.27 | 12.20 |
>
> In fact, **latency scales sub-linearly with the number of ensemble components**, because the framework is highly parallelizable: (i) batch across ensemble components and (ii) distribute evaluation across multiple GPUs.  Since OutFormer requires only a single forward pass per context without any per-dataset training, it plays us a strong hand for ensembling while maintaining modest and controllable cost.
>
> In addition, we show the mean rescaled AUROC (rAUC):
>
> | Size | No Ens. | Ens. (5) | Ens. (10) | Ens. (20) | Ens. (50) | Ens. (100) |
> |--------------|--------:|---------:|----------:|----------:|----------:|-----------:|
> | Small | 0.963 | **0.965** | 0.965 | 0.964 | 0.965 | 0.964 |
> | Medium | 0.958 | 0.974 | 0.978 | 0.977 | **0.983** | 0.978 |
> | Large | 0.924 | 0.973 | 0.984 | 0.986 | 0.987 | **0.990** |
>
> Ensemble improves performance by averaging over diverse context samplings. Gains saturate on small datasets, but continue to scale on larger datasets due to increased context diversity and improved estimation of the decision boundary. Ensemble offers a favorable performance–latency trade-off; Figure 1 in our paper shows OutFormer remains faster than most competitors.
>
> **Q2 & W2, Analysis of categorical variables**: we have acknowledged the lack of categorical-specific priors (See Limitations, Appendix B). In addition, we evaluate OutFormer on datasets with high-cardinality categorical features.
>
> Most public AD datasets lack high-cardinality categorical features that meaningfully contribute to outliers. One relevant dataset is *IEEE-CIS-Fraud* [1] for Vesta’s e-commerce fraud detection. Following the basic solution, we use EDA-selected features with mean imputation, yielding 186 mixed (one-hot + numeric) features, and use inlier-only train subset as context. We also repurpose classification datasets from *TALENT* [2] with >15 categorical features. We retain only categorical features, treat the majority class as inliers, and downsample other classes (15–20%) as outliers.
>
> OutFormer can be used for categorical AD in two ways: (i) One-hot + subsampling: Treat one-hot features as numeric and subsample to 100 dimensions. (ii) Feature projection: Apply the XStream projection (see Sec. 3.2 of [3]) to construct 100 numeric features. The performance of OutFormer is below:
>
> | Dataset | iForest(default HP) | OutFormer (w/o proj.) | OutFormer (w/ proj.) |
> |--------|--------:|----------------------:|---------------------:|
> | Customer Satisfaction (N=20K, D=88) | 0.873 | **0.912** | 0.755 |
> | Mfeat_pixel (N=128, D=1648) | 0.901 | **0.944** | 0.894 |
> | MIC (N=890, D=247) | 0.789 | **0.789** | 0.765 |
> | PhishingWebsites (N=3941, D=68) | 0.701 | **0.887** | 0.770 |
> | Student Dropout (N=1413, D=366) | **0.685** | 0.650 | 0.660 |
> | Android Permissions (N=9408, D=172) | 0.693 | **0.793** | 0.751 |
> | IEEE-CIS-Fraud (N=455K, D=186) | **0.741** | 0.738 | 0.739 |
>
> OutFormer (w/o. projection) outperforms iForest on **5/7 datasets**, indicating that existing priors **generalize reasonably well to categorical settings**. Projection-based features are less effective, due to information loss. While current priors already show generalization, we agree that designing dedicated categorical priors is an important direction for future work.
>
> [1]Addison Howard, Bernadette Bouchon-Meunier, IEEE CIS, inversion, John Lei, Lynn@Vesta, Marcus2010, and Prof. Hussein Abbass. IEEE-CIS Fraud Detection. https://kaggle.com/competitions/ieee-fraud-detection
>
> [2] Liu SY, Cai HR, Zhou QL, Yin HH, Zhou T, Jiang JP, Ye HJ. TALENT: A tabular analytics and learning toolbox. Journal of Machine Learning Research. 2025;26(226):1-6.
>
> [3] Manzoor et al. (2018). XStream: Outlier Detection in Feature-Evolving Data Streams. KDD.

---

### Decision · Program_Chairs · 2026-04-30

**Decision:**

Accept (regular)

**Comment:**

The authors propose a foundation model for anomaly detection in tabular data. The new method is an in-context learner that builds on the recent tabPFN model. The authors propose several new elements that make the model better suited to the task than the original PFN model. Reviewers were positive about the paper and noted some of its merits. These include the use of diverse synthetic priors, a self-evolving curriculum to stabilize training, and a broad evaluation on 1500+ datasets. I see the evaluation on these additional datasets (beyond Adabench) as a valid contribution. Reviewers noted some limitations regarding inference time and handling categorical variables, but the authors addressed these concerns effectively in the discussion. I find the fact that TabpfnOD performs very similarly to the method as a limitation, but the method shines in runtime compared to the vanilla PFN. Overall, I think this is a solid paper and would be interesting for the community, and I recommend acceptance.